# Model-based analysis of solute transport and potential carbon mineralization in the active layer of a hillslope underlain by permafrost with seasonal variability and climate change

Alexandra Hamm [1,2], Erik Schytt Mannerfelt [3], Aaron A. Mohammed [4,5], Scott L. Painter [6], Ethan T. Coon [6], and Andrew Frampton [1,2]

[1]Department of Physical Geography, Stockholm University, Stockholm ,Sweden
[2]Bolin Centre for Climate Research, Stockholm University, Stockholm, Sweden
[3]Department of Geosciences, University of Oslo, Oslo, Norway
[4]Department of Earth and Environmental Sciences, Syracuse University, New York, 13244, USA
[5]Department of Civil and Environmental Engineering, Syracuse University, New York, 13244, USA
[6]Climate Change Science Institute and Environmental Sciences Division, Oak Ridge National Laboratory, Oak Ridge, Tennessee 37830, USA

**Correspondence:** Andrew Frampton (andrew.frampton@natgeo.su.se)

**Abstract.** Permafrost carbon, stored in frozen organic matter across vast Arctic and sub-Arctic regions, represents a substantial and increasingly vulnerable carbon reservoir. As global temperatures rise, the accelerated thawing of permafrost releases greenhouse gases, exacerbating climate change. However, freshly thawed permafrost carbon may also experience lateral transport by groundwater flow to surface water recipients such as rivers and lakes, increasing the terrestrial-to-aquatic transfer of permafrost carbon. The mobilization and subsurface transport mechanisms are poorly understood and not accounted for in global climate models, leading to high uncertainties in the predictions of the permafrost carbon feedback. Here, we focus on a hillslope in Endalen valley, Svalbard, as a representative example of a high-Arctic hillslope underlain by continuous permafrost. We analyze solute transport in the form of a non-reactive tracer representing dissolved organic carbon (DOC) using a physics-based numerical model with the objective to study governing cryotic and hydrodynamic transport mechanisms relevant for warming permafrost regions. We first analyze transport times for DOC pools at different locations within the active layer under present-day climatic conditions and proceed to study susceptibility for deeper ancient carbon release in the upper permafrost due to thaw under different warming scenarios. Results suggest that DOC in the active layer near the permafrost table experiences rapid lateral transport upon thaw due to saturated conditions and lateral flow, while DOC close to the ground surface experiences slower transport due flow in unsaturated soil. Deeper permafrost carbon release exhibits vastly different transport behaviors depending on warming and thaw rate. Gradual warming leads to small fractions of DOC being mobilized every year, while the majority moves vertically through percolation and cryosuction. Abrupt thaw resulting from a single very warm year leads to faster lateral transport times, similar to active layer DOC released in saturated conditions. Lastly, we analyze the potential susceptibility of DOC to mineralization to $CO_2$ prior to export due to soil moisture and temperature conditions. We find that high liquid saturation during transport coincides with very low mineralization rates and potentially inhibits miner-

alization into $CO_2$ before export. Overall, the results highlight the importance of subsurface hydrologic and thermal conditions on the retention and lateral export of permafrost carbon by subsurface flow.

## 1 Introduction

Permafrost stores vast amounts of soil organic carbon (SOC) currently immobilized in frozen ground (Zimov et al., 2006). This carbon stock has been built up over millennia and is not currently part of the active carbon cycle, but is subject to
remobilization under climate change (Tarnocai et al., 2009). Due to climate warming and permafrost loss, this carbon gets mobilized when thawed out (Miner et al., 2022), which may lead to the possible release of large amounts of terrestrial carbon into the atmosphere in the form of greenhouse gases (GHGs). However, the fate of the mobilized carbon is fraught with uncertainty around the questions of how much of this carbon will ultimately be released to the atmosphere and how much of it experiences lateral waterborne transport. Given that permafrost soils currently contain twice the amount of carbon compared to
the present atmospheric $CO_2$, the potential release of this vast carbon stock into the atmosphere would exert a profound impact on global climate dynamics (e.g., Schuur et al., 2015).

Apart from vertical release as GHGs through microbial mineralization, newly thawed carbon can also be dissolved in groundwater and then be transported as dissolved organic carbon (DOC, Connolly et al., 2020) and lead to a lateral export of permafrost carbon. This lateral transport increases the terrestrial-to-aquatic transfer of SOC, which affects rivers and oceans. Depending
on its biogdegradability, a large part of riverine DOC will be mineralized in the river or delivered to oceans, a small part can be buried within the river sediments (Cole et al., 2007; O'Donnell et al., 2012; Abbott et al., 2014). The fate of permafrost carbon is an essential piece of the global carbon cycle and requires a better understanding of DOC transport mechanics in groundwater (Plaza et al., 2019).

Thawing of permafrost not only releases organic carbon but also has implications for the mobilization of other chemical
species, including contaminants such as mercury or trace metals. As permafrost thaws, the previously sequestered contaminants in frozen soil can be released into the environment, potentially increasing pollutant levels in aquatic systems, threatening human health (O'Donnell et al., 2012; Smith et al., 2024). Additionally, the increased input of DOC from thawed permafrost can influence surface water chemistry. The lateral transport of DOC into oceans may interact with ocean acidification processes, as increased $CO_2$ absorption by oceans decreases pH (Semiletov et al., 2016). This interaction could affect marine biogeochemical
processes and ecosystem health.

Groundwater flow in permafrost regions is for the most part restricted to sub- and supra-permafrost groundwater flow (Walvoord and Kurylyk, 2016). With permafrost acting as a largely impermeable layer between the two, shallow and deep aquifers are mostly disconnected systems in continuous permafrost regions (e.g., Kane et al., 2013). Most of permafrost carbon in the circumpolar permafrost region (46% of SOC in the upper three meters) is stored in the uppermost meter of permafrost soils with deeper layers (1–2 m and 2–3 m depth) containing increasingly less carbon (34% and 20% of total SOC in the upper three meters, respectively Hugelius et al., 2014). Within the active layer, a higher density of carbon abundance can be observed close to the surface in the top organic layer (TOL) as well as cryoturbated material close to the active layer-permafrost boundary (Siewert et al., 2015).

Lateral transport of compounds in permafrost regions is not only important for the fate of organic carbon, but also for the transport of anthropogenic contaminants (Miner et al., 2021), mercury bound to organic material (Schuster et al., 2018), and other chemical species (Wu et al., 2022). Observing transport and groundwater flow in general is inherently difficult; field experiments in which tracers are used in permafrost landscapes (e.g., Wales et al., 2020) give an idea of the velocity of flow as well as the dispersion of solutes but lack the possibility to obtain observations continuous in time. Numerical models allow both simulating current groundwater flow as well as associated heat and solute transport (e.g., McKenzie et al., 2007; Frampton et al., 2011; Harp et al., 2016; Lamontagne-Hallé et al., 2018; Dagenais et al., 2020; Sjöberg et al., 2021), and also enable future predictions and changes in the cryotic-hydrological system (e.g., Bense et al., 2009; Ge et al., 2011; Bense et al., 2012; Frampton et al., 2013; Frampton and Destouni, 2015; Kurylyk et al., 2016; Shojae Ghias et al., 2019; Painter et al., 2023). Therefore, modeling constitutes an important tool for investigating the ultimate fate of permafrost carbon.

Numerical modeling of solute transport under freeze-thaw conditions is computationally challenging and requires well-optimized computer code that can solve the complex interplay between thawing, freezing, groundwater flow, and solute transport (Lapalme et al., 2023; Lemieux et al., 2024). Recent modeling results have highlighted the importance of including freeze-thaw processes when modeling solute transport in permafrost regions (Frampton et al., 2011; Mohammed et al., 2021; Guimond et al., 2021; Huang and Rudolph, 2023; Zastruzny et al., 2024). Jafarov et al. (2022) used the Advanced Terrestrial Simulator (ATS, Coon et al., 2019) to show the difference between including and excluding freeze-thaw dynamics in simulations representing low centered polygons in a polygonal tundra landscape. They use a non-reactive tracer to represent dissolved constituents in the groundwater and found that in the simulations including freeze-thaw, most of the modeled tracer gets mobilized within the freeze-up period and vertical tracer movement is greatly enhanced as compared to the simulations that do not account for freeze-thaw dynamics, where lateral transport is significantly lower. They suggest that capillary forces (cryosuction) might play a substantial role in the movement of solutes during freeze-up.

In this study, we quantify the transport of waterborne solutes in a model representation of a hillslope with a seasonally thawed active layer underlain by permafrost. The model is based on a hillslope in Endalen, Svalbard, which is used as a representative site for the high Arctic region of Svalbard and other similar conditions. We analyze advective transport of a tracer as a proxy for DOC (Fig. 1a), without accounting for carbon-specific reactions, but with a representation of potential mineralization rates. We define multiple tracers to represent carbon pools at different depths in the active layer and within the permafrost (Fig. 1b). Carbon pools in the contemporary active layer are represented by a top organic layer (TOL) close to the surface as seasonal

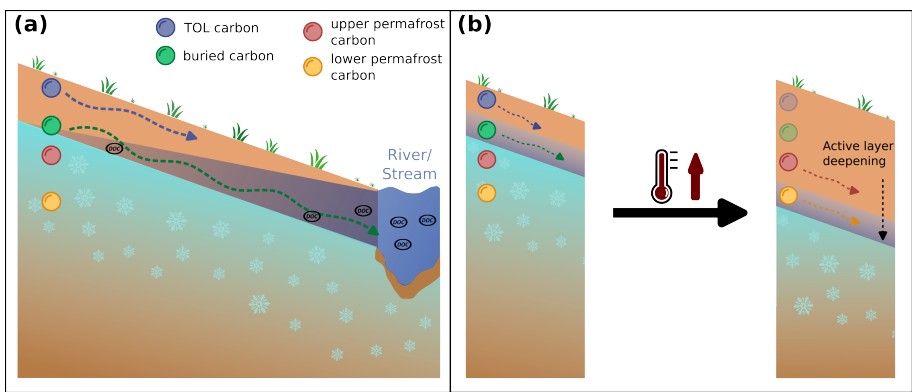

**Figure 1.** Conceptual representation of permafrost carbon transport in the form of dissolved organic carbon in a High Arctic hillslope setting underlain by continuous permafrost. (**a**) Different carbon pools in the active layer (Top organic layer, TOL) carbon, blue sphere, and buried carbon, green sphere) represent present-day active layer carbon release with hypothesized differences in transport velocities indicated by the length of the arrow (short: slow, long: fast). (**b**) Ancient carbon (red and yellow spheres) represent carbon sources within the permafrost, currently immobilized, and their mobilization upon permafrost thaw through increasing air temperatures in the course of climate change and associated active layer deepening.

plant litter, and buried carbon as cryoturbated carbon at the bottom of the active layer. The aim is to understand the relative importance of these sources in terms of their potential for GHG release. This is achieved by comparing transport velocities and residence times in the unfrozen soil, and evaluating their respective availability for microbial mineralization. Furthermore, we explore the effect of a warming climate on the mobilization of ancient, currently frozen, carbon in the upper permafrost zone. Two additional tracers in the model represent carbon sources that are currently immobilized in permafrost but susceptible to release by active layer expansion due to warming. We hypothesize that for a high-Arctic hillslope as represented by our study site in Endalen, Svalbard (i) buried carbon within the active layer will be transported faster than TOL carbon due to higher saturated soil conditions at the bottom of the active layer; and (ii) ancient permafrost carbon will be exposed to similar highly saturated late season conditions leading to rapid transport upon thaw. This insight into transport mechanisms active in permafrost landscapes contributes to understanding the impacts of hydrological flows on permafrost carbon feedback in a changing climate.

## 2 Methods

### 2.1 Study site

The field site considered is located within the Endalen valley on Svalbard (78°11' N, 15°44' E), about 5 km east of the main settlement of Longyearbyen, along the North-West facing slope and extending from the water divide at the top of the adjacent plateau down to a groundwater spring located on the foothill adjacent to Endalen river (Fig. 2). The area has been surveyed

by a drone flyover in summer 2022 to produce a high resolution digital elevation model ($5 \times 5$ cm; Schytt Mannerfelt, 2023, Appendix F). We delineate the photogrammetry-derived DEM using QGIS (version 3.22.4). The landscape in the area is characterized by gentle slopes towards the valley bottom ($\sim 12°$) with gravely soils covered by a shallow organic layer, and

a steep slope ascending towards the plateau ($\sim 37°$) characterized by a mostly gravely and rocky subsurface. Soil profiles in surrounding areas have shown that a large part of the SOC is stored within the near-surface layers of the TOL but that solifluction and cryoturbation have also led to an increase in SOC in the deeper layers of the sites where cryoturbation processes are active (Weiss et al., 2017). However, this is only true for soil profiles close to the lower, gentle part of the slopes where there is at least a thin organic layer (Appendix B). The total area considered is approximately $0.25 \, \text{km}^2$, and the elevation difference

between the groundwater spring and the uppermost edge on the plateau is 376 m. Additional qualitative field observations are described in Appendix B.

## 2.2 Model

We adopt a semi-generic modeling approach, where the model represents the main topographical features of a hillslope site and makes use of weather station data, but where soil properties are based on general observations and literature values (Table

1). The model design aims to capture key physiographic characteristics of the site and is intended to be a synthetic domain that is broadly representative of hillslopes in continuous permafrost regions. Thereby, we represent a realistic setup in terms of topography and weather conditions in a continuous permafrost landscape where observations are sparse, especially observations of groundwater flow. Semi-generic numerical modeling is a commonly used approach for investigating groundwater systems in remote cold regions where data availability is limited (Lemieux et al., 2024; Lamontagne-Hallé et al., 2020; Walvoord and

Kurylyk, 2016).

We use a physics-based numerical model which couples cryotic and hydrological processes (the Advanced Terrestrial Simulator; ATS v1.4.1, Coon et al., 2019). In its configuration for modeling cold regions, ATS couples the intricate interplay of freeze-thaw dynamics and both surface and subsurface energy, and surface and subsurface hydrology (Painter et al., 2016) and has been successfully evaluated against multiple types of field observations (Jan et al., 2020; Painter et al., 2023). ATS employs

an adaptive time-stepping scheme with user-defined minimum and maximum time steps. In this study, we set the minimum time step to $10^{-10}$ days and the maximum to 1 day, allowing for efficient and accurate simulations across different seasons, which require higher or lower resolution time stepping. Solute transport is represented through the advection-dispersion equation (Molins et al., 2022), which in simplified form can be written as

$$\frac{\partial (\phi s_l C)}{\partial t} = -\nabla \cdot (qC) + \nabla \cdot (\phi s_l D_l \nabla C) + Q_s \tag{1}$$

where $\phi$ is soil porosity (-), $s_l$ is liquid saturation (-), $C$ is solute concentration in liquid water (mol solute $\text{mol}^{-1}$ water), $t$ is time (s), $q$ is specific discharge of water ($\text{m s}^{-1}$), $D_l$ is the hydrodynamic dispersion coefficient ($\text{m}^2 \, \text{s}^{-1}$), and $Q_s$ is a solute source/sink term ($\text{mol s}^{-1}$). Solute is excluded from the ice phase by the formulation, which represents solute as moles of solute per moles of liquid water and by a nonlinear solver that enforces solute mass conservation (for details see Molins et al. (2022)).

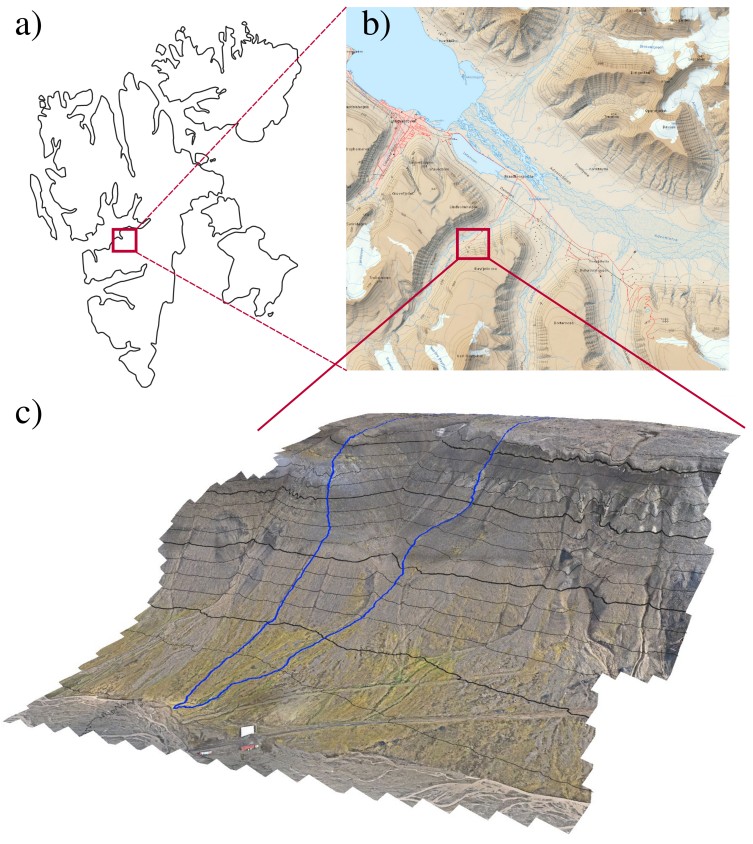

**Figure 2.** Overview over the study area. **(a)** Location of the study area within the Svalbard archipelago with the red square indicating the location of the Adventdalen valley close to the settlement of Longyearbyen. **(b)** Overview over the Adventdalen area with the red square highlighting the location of the Endalen valley system (map data based on Toposvalbard (Norwegian Polar Institue)). **(c)** Drone orthophoto view of the area within Endalen valley which forms the basis for the model setup. The blue outline represents the convergent hillslope area considered and is delineated based on the high resolution digital elevation model using QGIS.

Freezing of pore water or melting of pore ice results in an increase or decrease, respectively, of solute concentration in the liquid
phase. In the simulations performed in this study, dispersion is intentionally omitted by setting $D_l = 0$, focusing exclusively on solutes advected by water flow. This approach allows for the identification and analysis of the transient and seasonally variable flow field exhibited in the active layer, as well as the effects of freeze-thaw, cryosuction, wetting-drying in partially saturated soil, and unfrozen water seepage in permafrost. Although it would be valuable to consider dispersion, it is highly substrate-dependent and challenging to quantify with field observations. Consequently, assuming dispersion coefficients would
increase parameterization uncertainty and potentially obfuscate the advection-specific transport patterns caused by small-scale water movement such as percolation of unfrozen water at sub-zero temperatures or transport through capillary forces (such as cryosuction). Thus, the omission of dispersion ensures a clearer focus on lateral transport driven by advection, simplifying

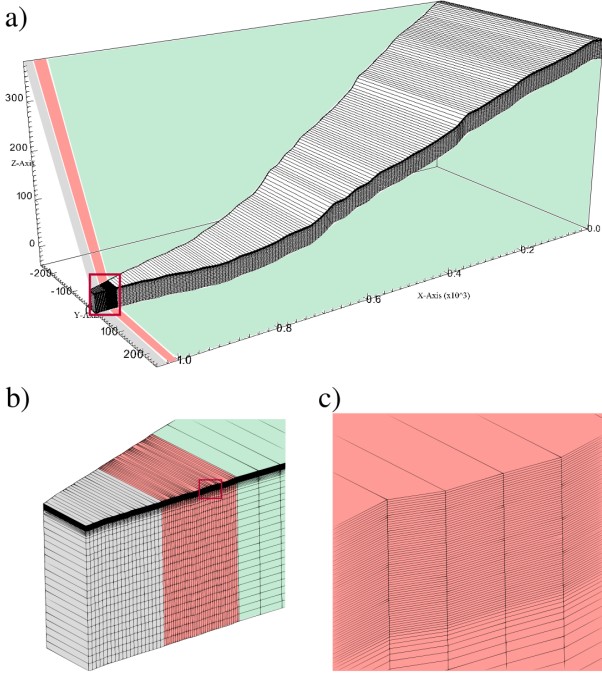

**Figure 3.** The model domain with representation of the computational mesh used for numerical simulations, which corresponds to a convergent hillslope with a wide width at the upper boundary and a small width at the outlet. The model area, length, variable width and elevation is based on the hillslope shown in Fig. 2. Dimensions of mesh elements vary in all three (x, y, z) directions. The domain discretization adopts a variable mesh approach, where the width in the $y$-direction increases uniformly with distance along the $x$-direction. Cell length $\Delta x$ in the $x$-direction is larger in the upper part of the domain ($\Delta x \approx 6\,$m) and smaller in the lowermost $40\,$m of the domain ($\Delta x = 0.5\,$m between $x = 1000\,$m and $1020\,$m and $\Delta x = 1\,$m between $x = 1020\,$m and $1040\,$m). The zoomed panel (b) shows the transport area, which is the main region of interest used for analysis of solute release and transport and has a finer resolution mesh. Throughout the entire domain the resolution in the $z$-direction is finer in regions corresponding to the active layer ($\Delta z = 0.02\,$m in the upper $1.2\,$m; panel c) and coarser in regions corresponding to permafrost (varying with depth with $\Delta z = 1.2\,$m to $2\,$m). The total depth of the domain is $40\,$m and the elevation is defined relative to the surface at the outlet (the groundwater spring), which is set to $0\,$m elevation.

interpretation of the model's results. For a full description of the governing equations used in the permafrost configuration of ATS, please refer to Painter et al. (2016) and Atchley et al. (2015).

## 2.3 Mesh and boundary conditions

The two-dimensional model domain is configured using a variable-width mesh (Fig. 3) with length, area and elevation corresponding to the DEM obtained from the Endalen hillslope (Fig. 2c). In general, a variable width approach is necessary to preserve the contributing area to a stream segment for a given travel distance to the stream, where distance is defined along the flowpath. In our case, it enables capturing the hydrological and thermal processes of the upslope part of the hillslope in order

to obtain reasonable flow accumulation at the lowermost part of the model domain corresponding to the region where transport is studied (Fig. 3b).

The domain extends 1040 m in the direction of highest flow accumulation from the upper boundary to the outlet and is of variable width in the perpendicular direction. This enables representation of the upslope contributing area to the transport section, which is the main area of interest for subsequent analysis. The variable-width mesh approach preserves flow convergence, enables a reasonable representation of the surface-energy balance, and allows hydrological processes to be well represented without the expense of a fully three-dimensional model (e.g., Fan and Bras, 1998; Troch et al., 2003; Hazenberg et al., 2015). In the direction of flow, the mesh is divided into three key sections (Fig. 3):

1. The upper slope (0-1000 m) – this primarily serves as a water source with representation of net precipitation and surface-energy balances feeding the foothill (green-shaded area).

2. The transport section (1000-1020 m) – this is the main area of interest for the model analysis of solute transport and corresponds to the foothill of the hillslope (red-shaded area).

3. The buffer zone (1020-1040 m) – this reduces boundary effects at the model outlet from influencing simulation results (grey-shaded area).

The mesh is coarser in the upper slope area and refined in the lower areas to better capture fine-scale processes in the transport section close to the valley bottom. We repeated the spinup simulations with a refined mesh to confirm that the mesh resolution is adequate (Appendix D). This strategy, together with variable-width elements that preserve flow convergence, provides a suitable representation of hillslope processes (Appendix A and D).

Boundary conditions are prescribed as zero-flux on the vertical sides and the bottom. At the bottom (40 m depth), a constant temperature boundary condition of -6.6°C is applied, following borehole observations (Christiansen et al., 2020). At the surface, a surface energy balance model (Atchley et al., 2015) is used to obtain thermal exchange between the atmosphere and snow, ice, ponded water enabling evapotranspiration, subsurface recharge and ground heat flux. This is advantageous as it allows for standard hydro-meteorological weather station data to be used to drive the model instead of imposing heat and flow (recharge) boundary conditions on the surface (Lamontagne-Hallé et al., 2020).

Soil physical properties are generic and literature values are used based on qualitative field observations and are defined to resemble highly conductive material (Table 1, Appendix B). A more detailed description of the mesh setup, boundary conditions, and soil physical properties is provided in Appendix A.

## 2.4 Model spinup and initial conditions

To initialize the model, a 3-step spinup procedure is required (e.g., Jafarov et al., 2018; Jan et al., 2020; Hamm and Frampton, 2021). First, a single column model extending to the full depth of the final 2D mesh is used to establish an ice-saturated subsurface with an ice table near the surface. This was accomplished by freezing an initially hydrostatic and isothermal water column from below keeping an open top boundary to allow the volume expansion of the phase change to push out excess water.

**Table 1.** Physical soil properties used in all model runs for the entire model domain (Freeze and Cherry, 1979; Fitts, 2013; Schwartz and Zhang, 2024). Porosity and permeability are from e.g. (Freeze and Cherry, 1979; Fitts, 2013; Schwartz and Zhang, 2024). Water retention parameters and thermal properties are from Schuh et al. (2017).

| Parameter | Unit | Value |
|---|---|---|
| Porosity | - | 0.5 |
| Permeability | $m^2$ | $2 \times 10^{-11}$ |
| van Genuchten $\alpha$ | $Pa^{-1}$ | $3 \times 10^{-4}$ |
| van Genuchten n | - | 2 |
| Thermal conductivity (saturated, unfrozen) | $W\,m^{-1}\,K^{-1}$ | 1 |
| Thermal conductivity (dry) | $W\,m^{-1}\,K^{-1}$ | 0.3 |

This sets a temperature of -6.6 °C at the bottom of the column and a permafrost table close to the surface. Second, the column model is run with the full forcing dataset until a cyclic steady-state is reached and year-to-year differences in e.g., daily ground temperature or evaporative flux are negligible. In the third step, the resulting column data is mapped to each of the columns in the transect, and the entire model is run again for ten years of average weather conditions accounting for all lateral processes such as lateral water flow and energy transport. We confirmed that the temperature is in a cyclic steady state at the end of the 10 year spinup period (Fig. D2).

The resulting subsurface state presents the initial conditions for the transient simulations. In total, there are five different transient simulation scenarios. The first one represents present-day weather and active layer development conditions. Two simulations represent gradual climate warming scenarios and two further simulations that mimic an abrupt increase in active layer depth (see Sect. 2.5 for in-depth descriptions of the scenarios). We consider a total of four tracer injection points in the subsurface domain of the model: two tracers within the extent of the active layer and two within the upper permafrost (for exact locations see Fig A2). In the active layer, we define a tracer injection point representing TOL carbon consisting of recently deposited SOC close to the surface, and cryoturbated or buried carbon that has been vertically moved towards the bottom of the active layer and upper permafrost layers. Buried carbon can especially be found in solifluction-affected hillslope system, such as Endalen (see Fig. E1 and Weiss et al., 2017). The TOL carbon is injected close to the surface (0–2 cm below the surface) and the buried carbon is injected close to the bottom of the active layer (90–92 cm below the surface) in the first mesh column of the main area of interest of the mesh (x = 1000–1000.5 m, see Fig. A2). Both tracers are injected during fully frozen conditions from May 16 to 17 (24 hours). This ensures that they stay frozen and in place before the onset of thaw. The injection rate is a constant rate of $0.0012\,mol\,s^{-1}$. This yields a total of 100 mol in the model, which is an arbitrary number to represent 100 units of solute. Since solute transport is conservative and non-reactive, the total mass of solute in the model will not change. The maximum simulated thaw depth (max. depth where the ground temperature > 0 °C) in the present-day model runs extends to 1 m.

Additionally, we define two injection points for tracers within the permafrost (representing ancient carbon): one close to the present-day active layer thickness (ALT; at 1.15 m depth) and one slightly deeper, just below the current permafrost table (at

1.55 m depth, see Fig. A2). These tracers are injected in the same mesh column, so that they have the same x-distance to the groundwater spring, and at the same time and rate as the active layer carbon tracers, yielding a total of 100 moles each.

## 2.5 Model forcing by weather and climate

The model forcing by hydro-meteorological data for the different model scenarios represent different climatic conditions,
depicting current conditions as well as gradually warming air temperatures, and two scenarios representing an abrupt deepening of the active layer.

### Present-day weather conditions

To simulate active layer carbon transport, we use present-day weather conditions to run the model (the same forcing data as used during the spinup). This simulation is run for two years. We analyze transport times by observing the tracer breakthrough
curve (BTC) at multiple distances relative to the injection point (at 10 and 20 m distance). In the model output, a BTC is defined when the concentration reaches a minimum threshold of 0.01 mol at a given observation point. We distinguish between surface and subsurface BTCs, with arrival times marked by the concentration exceeding and subsequently falling below this threshold. For surface transport of the buried tracer, however, the threshold is set to 0.0005 mol due to the small amount of mass transported along the surface. The thawing-out and mobilization of permafrost carbon is simulated in a different set of
experiments described below.

### Gradual warming scenarios

We simulate gradual permafrost carbon mobilization by incrementally increasing near-surface air temperature over a period of 50 years based on local predictions of air temperature trends in Svalbard (Hanssen-Bauer et al., 2018). The reference temperature is based on the current day-of-year average air temperature ($T_{avg}$), which is the only variable that changes over time
in the respective scenarios. We increase air temperature by applying warming rates specific for Svalbard of $0.125\,°C\,year^{-1}$ to represent an RCP8.5 scenario and a warming rate of $0.075\,°C\,year^{-1}$ for an RCP4.5 scenario (Hanssen-Bauer et al., 2018). These warming rates are based on the predictions for air temperature increases by the end of the 21st century and are applied to each day of the year equally (season-specific warming trends are not considered here). This scenario is stylized and only aims to capture the effects of air temperature warming without considering complex, difficult-to-predict, and highly uncertain changes
in precipitation. As we do not change the bottom boundary condition in these simulations, we confirmed with 1D simulations (Fig. D3) that the treatment of the lower boundary has negligible effect on the near-surface region of interest up to year 40 and then only minor effects from year 40-50. Note that the mesh resolution in the upper slope area has been reduced by a factor of four (equivalent to halving the resolution twice, resulting in 23 m per column) to decrease the computational cost for these long-term scenarios. We have tested and confirmed that this reduction in spatial resolution does not significantly impact seasonal
transport patterns, and we therefore do not anticipate long-term effects from the coarser mesh resolution (see Appendix D).

Additionally, the output frequency has been reduced to monthly intervals to accommodate the extended simulation time and manage data output over this period.

**Abrupt active layer deepening**

In two additional scenarios, we simulate abrupt active layer deepening by creating two scenarios with a single exceptionally warm year ($T_{avg} + 3\,°C$ and $T_{avg} + 5\,°C$, respectively). The warming is applied to each day equally. The purpose of this scenario is to create abrupt thaw of permafrost within a single summer, which represents and hyperbolizes the high interannual variation in ALT observed in Svalbard (Strand et al., 2020). The abrupt deepening simulations are run for five years; two years of average present-day conditions to assure equilibrium conditions, one year of increased air temperature, and two more years of present-day conditions. The tracer in these simulations is injected in the first winter before the onset of the first thaw.

## 2.6 Potential mineralization rates

To further evaluate the importance of solute transport in the permafrost carbon cycle, we compute the potential for carbon mineralization to $CO_2$ in the active layer throughout the thawing season based on the local prevailing environmental conditions in the soil in the present day scenarios. In this study, we do not consider mineralization into methane ($CH_4$), as the study site is located on a hillslope with rapid groundwater flow. Therefore, the production of methane is expected to be insignificant compared to those generated by wetlands (Denman et al., 2007). We calculate potential DOC mineralization depending on soil temperature following a $Q_{10}$ formulation, and soil moisture following a threshold behavior function with an optimum liquid saturation of 0.7 (Wen et al., 2020; Rawlins et al., 2021). While this formulation usually only accounts for a binary distinction between air and liquid saturation, we extended it by accounting for the presence of pore ice. For that, we multiply the function for liquid saturation by a function for air saturation, representing a threshold behavior with a maximum at 0.3 air saturation. Together, they represent soil moisture. Furthermore, since we do not work with absolute molar mass in this study, our potential for DOC mineralization solely depends on soil temperature and moisture (no kinetic rate constants or sorption parameters are considered). Hence, the three-phase adjusted equation adapted from Rawlins et al. (2021) to derive mineralization rates in our model scenarios is

$$r_m = k_{decomp} f(T) f(S_l) f(S_a), \tag{2}$$

where $r_m$ is the local potential mineralization rate (day$^{-1}$), $k_{decomp}$ is the DOC mineralization rate coefficient as per Rawlins et al. (2021) (set to $0.83 \times 10^{-2}$ day$^{-1}$) and $f(T)$, $f(S_l)$, and $f(S_a)$ are the temperature, moisture, and air saturation dependencies, respectively. Functions for temperature dependence $f(T)$, moisture dependence $f(S_l)$, and air saturation dependence $f(S_a)$ are described using the $Q_{10}$ coefficient (Wen et al., 2020)

$$f(T) = Q_{10}^{|T-10|/10}, \tag{3}$$

with $T$ being local soil temperature and $Q_{10}$ representing the mineralization rate increase per 10°C increase in temperature (in this case set to 1.7 according to Yurova et al. (2008) and Dusek et al. (2019)) and the threshold behavior functions (Wen et al., 2020) for soil liquid saturation $S_l$ and air saturation $S_a$ described as

$$f(S_l) = \begin{cases} (\dfrac{S_l}{0.7})^{1.5}, & S_l \leq 0.7 \\ (\dfrac{1-S_l}{1-0.7})^{1.5}, & S_l > 0.7 \end{cases}. \tag{4}$$

$$f(S_a) = \begin{cases} (\dfrac{S_a}{0.3})^{1.5}, & S_a \leq 0.3 \\ (\dfrac{1-S_a}{1-0.3})^{1.5}, & S_a > 0.3 \end{cases}. \tag{5}$$

A plot showing the scaling effect of soil temperature and moisture for a range of -20–30°C for temperature and 0–1 for both liquid and air saturation is given in Fig. 4. The resulting mineralization rates are representative of a mineralization rate per day and is highly dependent on soil moisture. In this formulation, mineralization results in 0 at fully saturated ($S_l = 1$ and $S_a = 0$) soil conditions due to oxygen deprivation. We chose this threshold behavior for soil moisture, because a linear behavior would result in most favorable conditions for mineralization under fully saturated conditions, which is not to be expected at the site.

Soil moisture and temperature conditions are extracted as a post-processing step within the present-day scenarios and do not affect the solute mass computed during the model run. Due to the large (monthly) output time-step size in the warming scenarios, we did not calculate mineralization for these scenarios. Further, because the tracer used in this model does not chemically resemble DOC, carbon mass is not included in this formulation, and actual mineralization rates cannot be obtained. However, this approximation allows for a first order estimate of the fate of the modeled DOC upon release accounting for the hydrothermal dynamics in the active layer. Lastly, this formulation does not account for the possibility of methane production as it is not expected to be the dominant mineralization mode at the hillslope site in Endalen.

## 3 Results

We evaluate solute transport by analyzing BTCs and tabulating peak as well as first and last arrival times of active layer carbon and ancient carbon, and by visualizing two-dimensional plume transport for each injected tracer. BTCs are obtained by vertically integrating a surface-subsurface column at 10 m and 20 m relative distance to the injection point for each given time step (for active layer carbon: minutes, for ancient carbon: days). The initial arrival and end of each BTC is defined by setting a threshold value of 0.01 mol. The arrival time is defined as the first day when the concentration exceeds 0.01 mol; the falling limb of the breakthrough curve is defined as the first day after the arrival time when the concentration falls below this threshold. The resulting value is a temporal snapshot of tracer mass at the given time. Hence, the BTC represents the temporal evolution of tracer concentration at a given location, showing how the concentration changes over time, though discretized due to time steps in the model. The plume distribution, on the other hand, represents the spatial distribution of tracer concentration

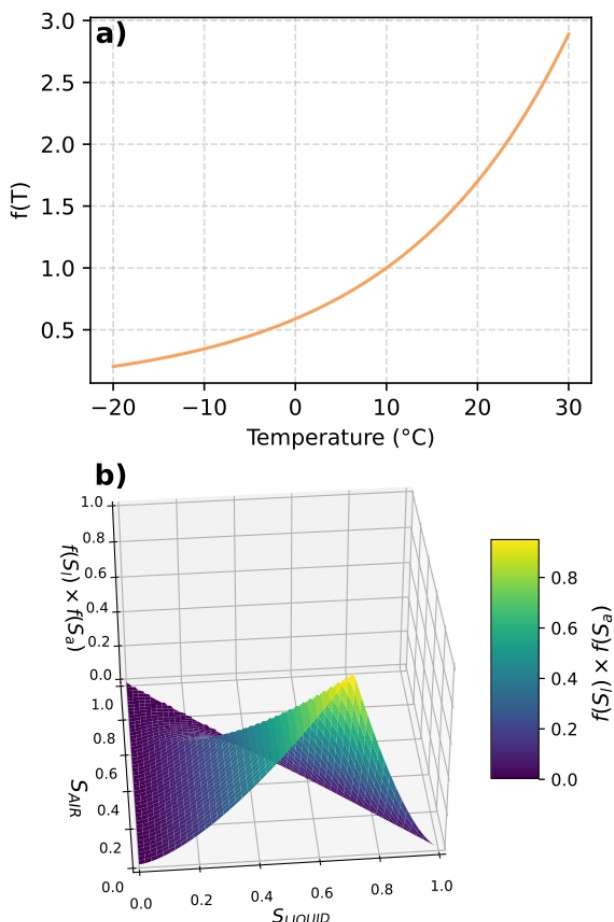

**Figure 4.** Functions **(a)** f(T) and **(b)** f(S$_l$) and f(S$_a$) using Eq. (3), (4), and (5), respectively, for a temperature range of -20–30°C and a liquid and air saturation range of 0–1. The value on the y-axis in **(a)** and z-axis in **(b)** corresponds to the scaling factor that is then used in Eq. 2.

at specific moments in time, showing how the tracer is distributed in space at a particular point in time. Arrival times are expressed as dates within a year that represent present-day weather conditions in the study site.

## 3.1 Active layer carbon

Tracers injected at different depths in the active layer exhibit different breakthrough behaviors (Fig. 5). Most noticeable, tracer breakthrough for both active layer sources occurs at different times throughout the warm season (June to October). Initial mobilization of TOL carbon starts on 3 June during early active layer development and gets released from the initial point of injection quickly (Fig. 6a and Table 2). During the early warm season, the active layer is shallow (thaw depth <10 cm) but snowmelt infiltration as well as rain infiltration cause the thawed soil to be highly water saturated until mid-June. This leads to

ponded water on the surface from a horizontal distance of 1011 m onward allowing for rapid surface runoff and transport (see

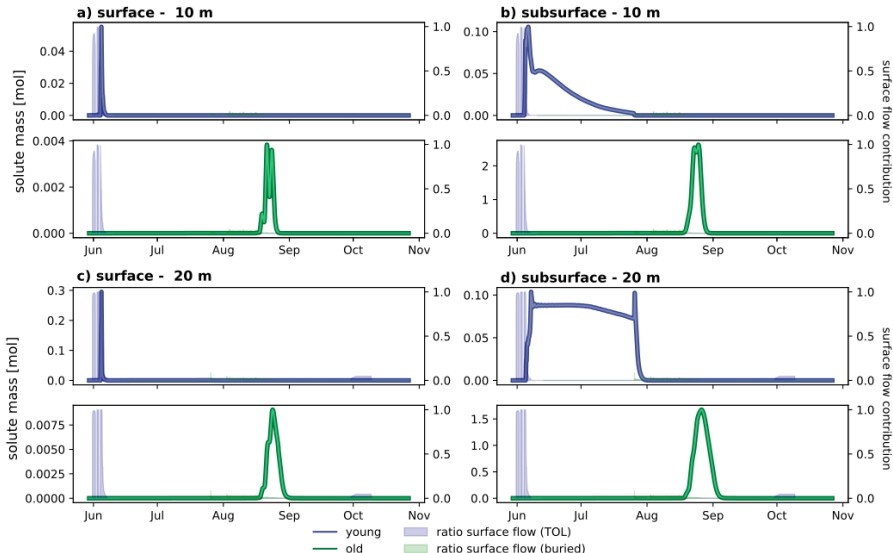

**Figure 5.** Tracer breakthrough curves (BTC) for **top organic layer (TOL)** carbon (blue line) and **buried** (green line) carbon tracer mass transported in surface runoff observed at (**a**) 10 and (**b**) 20 m relative distance to the injection point, and the tracer mass transported in the subsurface flow at (**c**) 10 and (**d**) 20 m distance. The blue and green shaded areas indicate the ratio (0–1) of transport on the surface as compared to the subsurface (0 = subsurface transport only, 1 = surface transport only). Note that the left y-axis scale varies between the four subplots to better visualize small amounts of transported mass.

Fig. 8). Surface transport is therefore responsible for 100% of the tracer mass passing through both observation points in the early thaw-season and only lasts for 1 day at both observation points (Fig. 5a and b and Table 2). From mid-June onward, most of the surface ponded water at the 20 m observation point infiltrates, reducing surface transport to essentially 0%. This aligns with field observations, where snowmelt in early June led to fully inundated conditions across portions of the foothill of the hillslope, followed by drying as thaw depth increased and melt water can infiltrate into the subsurface (see Appendix B).

**Table 2.** Initial mobilization, first, last, and peak arrival of both active layer carbon tracers, including breakthrough times for tracer transported in the subsurface and the surface. Note that a threshold value of 0.01 mol was applied to determine onset and end of the breakthrough curve. This threshold value was set to 0.0005 mol for the buried tracer transported in the surface due to the small values. The arrival times are not directly comparable to the above arrival times and are hence written in italics.

| carbon source | initial mobilization | first arrival | | peak arrival | | last arrival | | breakthrough time | |
| --- | --- | --- | --- | --- | --- | --- | --- | --- | --- |
| | | 10 m | 20 m | 10 m | 20 m | 10 m | 20 m | 10 m | 20 m |
| TOL subsurface | 3 Jun | 4 Jun | 5 Jun | 6 Jun | 7 Jun | 10 Jul | 28 Jul | 35 days | 53 days |
| TOL surface | 3 Jun | 4 Jun | 4 Jun | 4 Jun | 4 Jun | 5 Jun | 5 Jun | 1 day | 1 day |
| buried subsurface | 15 Aug | 17 Aug | 18 Aug | 25 Aug | 26 Aug | 30 Aug | 5 Sep | 12 days | 17 days |
| *buried surface* | *15 Aug* | *18 Aug* | *18 Aug* | *21 Aug* | *24 Aug* | *25 Aug* | *29 Aug* | *6 days* | *10 days* |

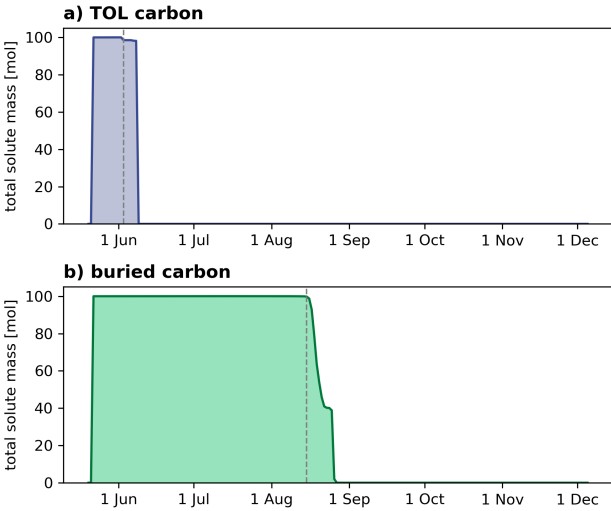

**Figure 6.** Mass release at the injection point of top organic layer (TOL) (a) and buried (b) carbon over time. Grey dashed lines indicate the timing of initial mobilization, which occurs when the thaw front reaches the solute injection depth (tracer mass $< 99.9$ mol, 3 June and 15 August for TOL and buried, respectively).

The remaining tracer mass that has not been transported by surface runoff is transported with groundwater flow or seepage, which starts at both observation points after mid-June, when all runoff is occurring in the subsurface. At the 10 m observation point (Fig. 5c) peak arrival of the subsurface pulse can be detected on 6 June. With active layer deepening throughout the warm season, the near surface layers become progressively less saturated as groundwater is restricted to deeper layers in the subsurface. This leads to unsaturated flow and transport in the vicinity of the TOL tracer, slowly moving the tracer from the 10 m to the 20 m observation point. An initial subsurface pulse peak at 20 m can be seen just after the shift from surface- to subsurface dominated transport on 7 June and can be attributed to the early active layer development, where the shallow active layer is saturated. Most of the transport during this time happens at the surface, but as soon as surface water starts infiltrating in mid-June, it creates the first subsurface BTC peak. A second peak at the 20 m observation point on 26 July indicates the arrival of the remaining mass mobilized in the warm season. The abrupt end of the BTC can be attributed to the increasing liquid saturation in the end of July that enables rapid transport at fully saturated conditions. All in all, the BTCs suggest long travel times in the subsurface upon initial release. The full range of breakthrough times from first arrival at 10 m (4 June) to last arrival at 20 m (28 Jul) amounts to a total of 54 days. Although the initial breakthrough is rapid and dominated by surface transport, the tail of the BTC is prolonged due to the residual mass experiencing unsaturated subsurface transport.

With a substantial amount of TOL carbon tracer mass transported during fully saturated conditions in the early active layer development, potential microbial mineralization rates are simultaneously low or even entirely absent (Fig. 7a). With an increasing and drying active layer in late June and throughout July, mineralization rates are increasing close to the surface to a maximum of $2.9 \times 10^{-5}$ day$^{-1}$. However, due to the transport behavior of the solute in the model, the tracer is mostly

located in the highly saturated zone, where mineralization is very low or absent (e.g., on 14 June, Fig. 7a). This suggests that the majority of carbon in the TOL may experience transport before mineralization becomes significant enough to degrade it. Later in the season, when TOL carbon transport is slowed down due to unsaturated conditions in the topsoil, mineralization will likely have an effect on the remaining solute mass and reduce the amount of potentially exported DOC.

Buried active layer carbon mobilization only starts on 15 August, when thaw depth development has progressed and almost reached its maximum extent (Fig. 6b and Table 2). Due to its location close to the base of the maximum active layer thickness, the dominant mode of transport is almost exclusively through subsurface flow. Initial arrival at the 10 m and 20 m observation point is observed on 17 August and 18 August, respectively. The full range of breakthrough times compared to the TOL carbon transport is significantly faster (12 and 17 days at the 10 m and 20 m point, respectively, Fig. 5c and d; Table 2). Liquid saturation at the bottom of the active layer is high throughout the year, leading to saturated and therefore faster subsurface transport compared to unsaturated conditions.

A small fraction of buried carbon tracer (tracer mass $< 0.005$ mol) experiences surface transport (also visible in Fig. 9) because it gets transported upwards by groundwater upwelling. This is caused by a combination of the terrain unevenness and the undulating impermeable permafrost table, causing local downslope water accumulation and a fully saturated active layer. Arrival and breakthrough times for buried carbon transported in surface runoff in Table 2 are therefore based on a separate threshold value of 0.0005 mol. The fully saturated conditions throughout the active layer further lead to an absence of mineralization during most of the transport period in the model setup presented here (e.g., on 25 Aug, Fig. 7b).

In summary, the timing of initial mobilization due to the depth of each of the carbon pool tracers, and liquid saturation in the soil greatly determines the shape of the BTC and its potential for mineralization for each of the active layer carbon tracers. Unsaturated subsurface conditions lead to a prolonged tail in subsurface TOL tracer breakthrough and non-zero mineralization rates, while saturated conditions lead to a rapid, mostly symmetrical breakthrough of the buried carbon tracer and no potential for mineralization.

The spatial distribution and spreading of the tracers within the transect are analyzed by visualizing temporal snapshots of the plume in 2D subsurface cross plots. Solute movement of the TOL carbon tracer is strongly limited to the surface and the uppermost soil layers (up to 10 cm below the surface) and does not significantly spread into deeper soil layers even after deeper active layer development (Fig. 8). Note that in this figure, only subsurface mass transport can be visualized. The subsurface plume spreads slowly and has not reached the 20 m observation point by the last temporal snapshot shown here (20 June). This highlights the significantly slower rate of unsaturated transport as compared to saturated transport, which is predominantly responsible for the rapid transport of buried carbon (Fig. 8). Here, the plume movement is largely uniform in the subsurface. Due to the unevenness of the terrain and highly saturated conditions, some tracer mass is transported upwards towards the surface (e.g., at 1005 m on 23, 25, and 27 August in Fig. 9), but the amount of buried carbon tracer being transported on surface is small (max. 8% on 26 July, Fig. 5b and d, green shaded area).

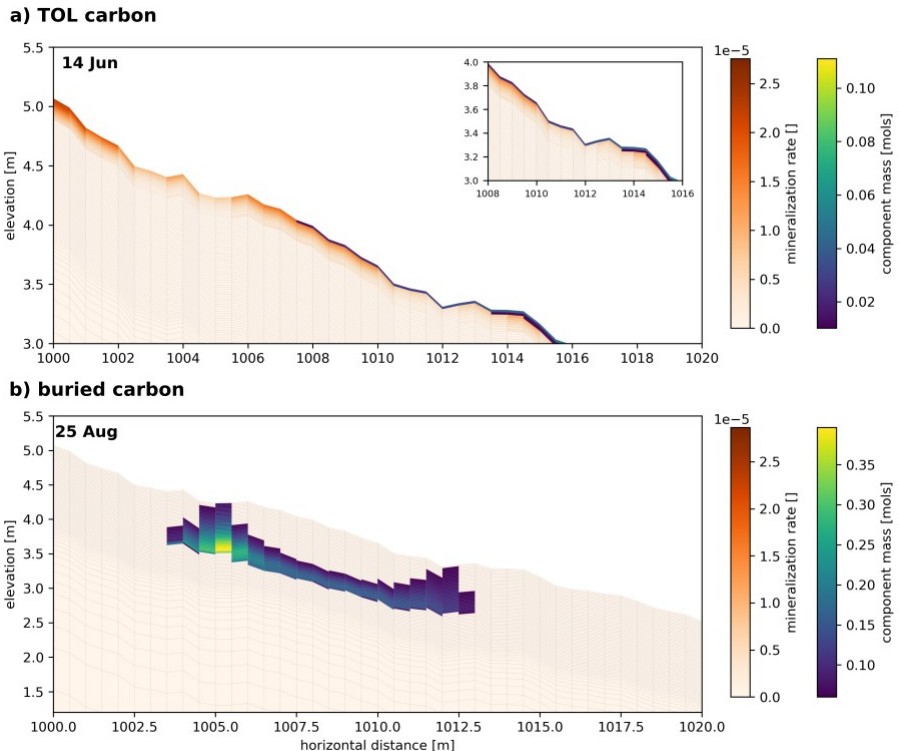

**Figure 7.** Temporal snapshot of potential normalized mineralization rate and component mass plume of (**a**) TOL carbon and (**b**) buried carbon spreading throughout the transect on 14 Jun and 25 Aug, respectively. Light-orange areas mark comparably low potential mineralization rates, dark-orange areas indicate the highest potential for mineralization. Component mass for the selected date (14 Jun and 25 Aug) is added as a blue to yellow overlay. Note that the tracer mass is restricted to the uppermost subsurface cell and masked to only represent values > 0.01 mol (for TOL carbon) and 0.06 mol (for buried carbon). The inset at the top right in panel (**a**) shows a zoomed-in version of the area in which tracer is present.

## 3.2 Ancient carbon

### Gradual warming scenario

Carbon mobilization as simulated in the gradual warming scenarios RCP4.5 and RCP8.5 exhibit distinctly different transport patterns from active layer carbon transport. Within the first year and in both gradual warming simulations, before air temperature warming has even affected the ALT, vertical mass movement of the upper ancient carbon tracer (ancient C at 1.15 m depth) can be observed. A substantial fraction of the initially injected tracer mass ($\sim 40\%$) moves vertically (both upwards and downwards) within the same mesh column in which it was injected (see Fig. 10a and d and Fig. 11). This way, carbon gets vertically distributed over the first four years, before the active layer has deepened sufficiently to allow for lateral transport to move tracer mass out of the initial column of injection. This observation can be attributed to percolation (during thaw) and

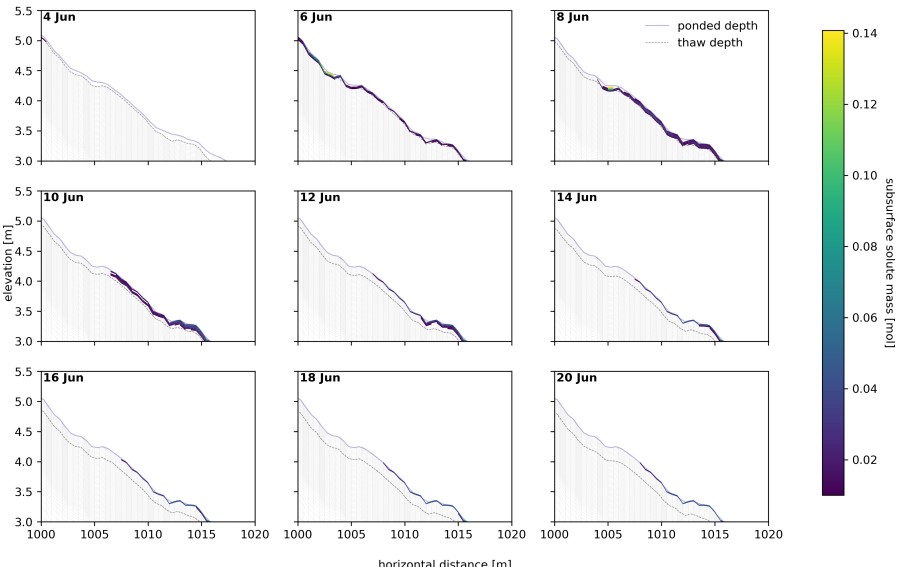

**Figure 8.** 2D representation of the top organic layer (TOL) carbon plume dispersion between the point of tracer injection (x = 1000 m) and the last observation point (x = 1020 m) for selected dates during summer after tracer injection. Elevation is given in relative elevation to the conceptual valley bottom (surface of the valley bottom = 0 m). Tracer mass is given in mol and is masked to only represent values > 0.01 mol. The blue solid line marks the water table above the surface (ponded depth), the black dashed lines indicates the thaw depth at each date.

cryosuction (during freeze-up). When the active layer is developing in the late thawing season, some tracer moves vertically downwards into partially frozen layers while the opposite occurs during the freeze-up when the freezing front from above draws water towards it due to capillary forces referred to as cryosuction.

From the fourth year onward, a small fraction (0.02–0.08 mol in the RCP4.5, Fig.10b and c, and 0.025–0.125 mol in RCP8.5, Fig.10e and f, respectively) of the ancient carbon source injected at 1.15 m depth arrives at the 10 and 20 m distance observation

points. At the 10 m observation point, annually recurring breakthrough curves are lasting for a short amount of time (component mass > 0.01 mol from mid-September to mid-October) with all the mass that has been released from the source during the respective summer arriving at 10 m and leaving the observation point within this time.

At 20 m, first tracer arrival starts within the same time after the beginning of the simulation as at the 10 m observation point, but instead of being flushed through the observation column, this tracer accumulates within the observation column with short

periods of release (less than 2 weeks) during the late autumn (mid September to mid October) and becomes immobilized upon freeze-up until it thaws out in the following season.

All mass (transported molar mass ≥ 95%) of the ancient carbon pool at 1.15 m has been laterally transported from the injection column after 34 years in the RCP8.5 scenario. In the RCP4.5 scenario, 50 years of gradual warming have led to a release of 93.5% of the ancient carbon tracer mass (Fig. 10a and d). Tracer arrival at the observation points is relatively even

throughout this period indicating no abrupt release as a result of gradual warming.

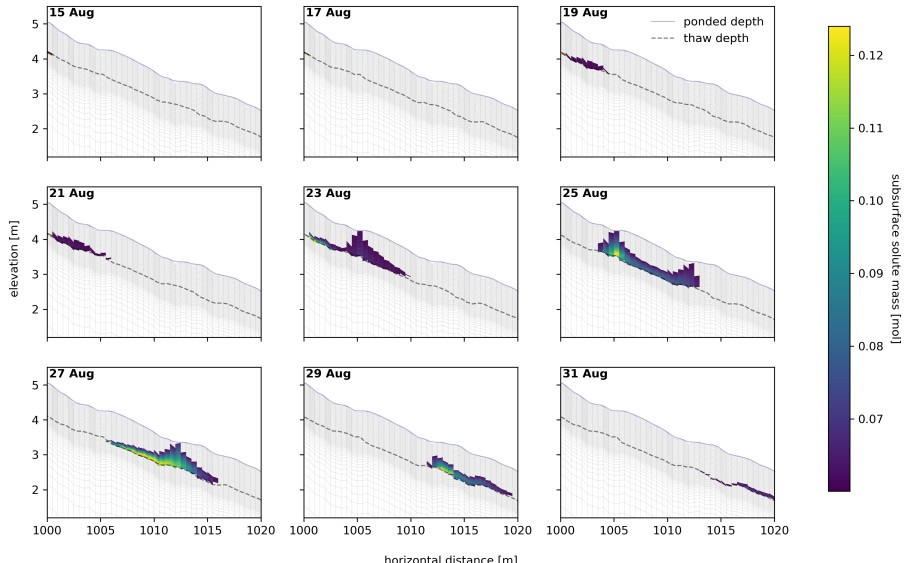

**Figure 9.** 2D representation of the buried carbon plume dispersion between the point of tracer injection (x = 1000 m) and the last observation point (x = 1020 m) for selected dates during summer after tracer injection. Elevation is given in relative elevation to the conceptual valley bottom (surface of the valley bottom = 0 m). Tracer mass is given in mol and values < 0.06 mol are masked for better visualization. The blue solid line marks the water table above the surface (ponded depth), the black dashed line indicates the thaw depth at each date.

A similar behavior can be observed for the ancient carbon pool at 1.55 m depth. Mobilization and lateral transport of this carbon pool (mass in injection column < 99%) starts after 33 years in the RCP8.5 scenario and after 48 years in the RCP4.5 scenario. By the end of the 50 year period, 85% of the mass in the injection column has been released in the RCP8.5 scenario. Towards the end of the simulation time, an abrupt release of the carbon pool tracer at 1.55 m can be seen. This is likely due to the formation of a talik beginning in the winter between years 48 and 49. The initial talik encompasses the ancient carbon injection cell at a depth of 1.15 m, allowing for extended transport periods, including during winter.

In the RCP4.5 scenario, only 1% of the tracer mass at 1.55 m depth has been released by the end of the simulation. Due to the post-processing procedure of the model output in which we integrate molar mass vertically throughout the injection column, rounding errors cause the total molar mass in the injection column to slightly exceed the initial mass of 100 mol (see Appendix C). This does not represent a physical increase in mass. It can therefore not conclusively be determined when exactly lateral movement of the tracer in 1.55 m depth was initiated. However, vertical mobilization of this tracer can be observed starting after 27 years, causing a vertical redistribution of 65% of the initially injected mass by the end of the simulation period. In summary, these results suggest that gradual warming leads to gradual release of ancient carbon pools with significant redistribution in the vertical directions.

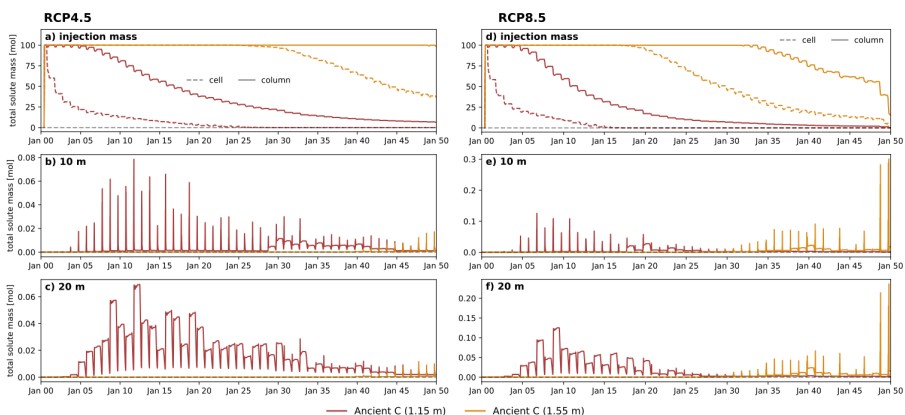

**Figure 10.** Ancient carbon mobilization upon gradual warming in the RCP4.5 (**a-d**) and RCP8.5 (**d-f**) scenario simulations. Tracer injection mass (**a** and **d**) indicate when the tracer gets mobilized at the initial injection point in the respective cell (dashed line) and integrated vertically over the entire injection column (solid line). Red and orange lines represent ancient carbon in 1.15 m and 1.55 m depth, respectively. When molar mass reaches 0, all tracer has been laterally transported away from the injection point. Tracer BTCs at 10 m (**b** and **e**) and 20 m (**c** and **f**) show the arrival times of tracer mass released at the injection column. Note that rounding errors of small amounts of mass have led to a tracer mass > 100 mol when integrated over the entire column. Hence, values > 100 mol were truncated to 100 mol. For details see Appendix C.

## Abrupt active layer deepening

In the case of the abrupt active layer deepening scenarios, tracer mobilization of ancient carbon is more similar to the behavior of buried active layer carbon released by seasonal thaw (Section 3.1). In the year of abrupt active layer deepening, ALT reaches 1.35 m (+35 cm compared to present-day ALT) and 1.6 m (+60 cm compared to present-day ALT) in the +3 and +5 °C scenarios, respectively. Such variability is not unprecedented. Strand et al. (2020) showed that variations in active layer depth in Adventdalen between cold and warm years can reach up to 30 cm. Our simulations fall within this range, but intentionally amplify the effects of an exceptionally warm year on active layer dynamics, resulting in an exaggerated active layer deepening by a factor of two. The abrupt deepening causes the entire BTC of the ancient carbon tracer at 1.15 m depth to start and finish within the same year and initiates the release of the tracer at 1.55 m in the +5 °C scenario (Fig. 12). Tracer molar mass of 0.05 mol (ancient C at 1.15 m depth) is exceeded on 13 August and falls below 0.05 mol on 27 August (duration: 14 days) at both the 10 and 20 m observation points in the +5 °C scenario. In the +3 °C scenario, these dates are 24 August and 14 September (duration: 21 days). For the ancient carbon tracer at 1.55 m depth, a small breakthrough of max 0.07 mol at the 10 m observation point and max 0.04 mol at the 20 m observation point can be observed in the +5 °C scenario. In the +3 °C scenario, this tracer remains frozen in the year of abrupt active layer deepening.

Observations of molar mass in the injection points in both simulations show that all mass of the ancient carbon at 1.15 m depth is released from the injection column within year two. In the +5 °C scenario, the ancient carbon at 1.55 m depth indicates partial lateral release of ∼ 8% of the injected mass by the end of year two. After full freeze-up in year three, the mass stays

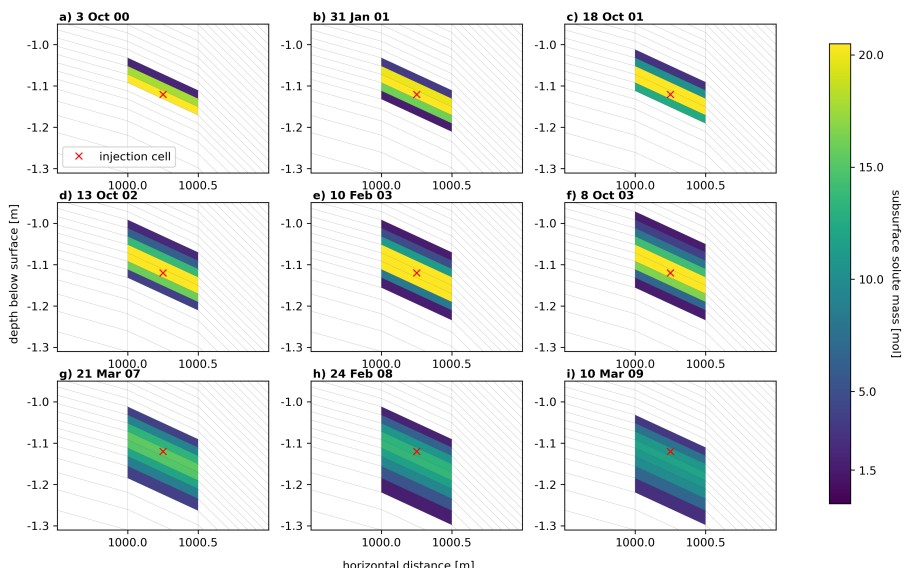

**Figure 11.** Representation of the vertical redistribution of tracer from the initial injection cell (red cross) throughout the depth profile of the injection column. (**a-c**) indicate the distribution throughout a full freeze-thaw cycle, (**d-f**) represent vertical redistribution from the third to the fourth freeze-up including the mid-winter vertical movement of the second freeze-up, and (**g-i**) represent vertical distribution by the end of the first modeled decade. Mass is depicted in molar mass and only values larger than 1.5 mol are shown, the remaining mass is masked for better visualization.

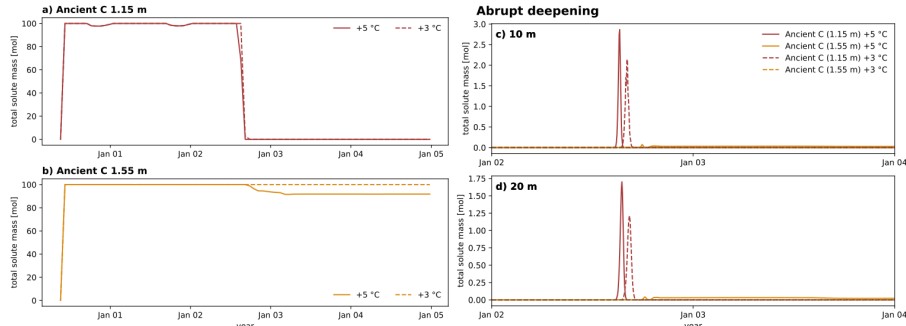

**Figure 12.** Mobilization of ancient carbon upon abrupt active layer deepening. The injected tracer mass (**a** and **b**) indicates when the tracer gets mobilized at the initial injection point in the respective vertically integrated injection column. Solid and dashed lines represent injection mass mobilization under the +5 °C and the +3 °C scenario, respectively. Tracer BTCs at 10 (**c**) and 20 m (**d**) indicate when and how fast the tracer passes through the observation column. Red and orange lines represent the tracer representing ancient carbon at 1.15 m and 1.55 m depth, respectively. Note that rounding errors of small amounts of mass have lead to a tracer mass > 100 mol when integrated over the entire column. Hence, values > 100 mol were truncated to 100 mol. For details see Appendix C.

constant throughout the following two summers, indicating no further release of tracer after the isolated abrupt active layer deepening event.

## 4 Discussion

Through our model experiments, we identified different transport patterns for different carbon pool representations within the active layer and in the permafrost. We find that TOL carbon pools may experience fast initial transport during early active layer development, but also show prolonged residence times due to unsaturated conditions in the subsurface. Buried active layer carbon, on the other hand, moves fast upon release in the saturated part of the active layer (confirming hypothesis (i)). Slower transport indicates longer residence time in the unfrozen soil upon thaw, while fast transport decreases the residence time.

The transport patterns observed in the buried carbon tracer do not carry over to the ancient permafrost carbon that is initially frozen and only gets mobilized upon ALT increase through gradual air temperature warming (confounding hypothesis (ii)). Instead of being flushed out when exposed to groundwater flow, the tracers that represent ancient carbon experience gradual release at increasing rates with increasing permafrost degradation. With rising temperatures near the thaw front, a small fraction of unfrozen water gets mobilized at sub-zero soil temperatures, enabling minor vertically downward migration in the soil profile

without experiencing significant lateral movement. During annual freeze-up, cryosuction (capillary forces causing water to be drawn to the freezing front) causes some fraction of the water and the tracer in it to be vertically migrated upward, moving it closer to the depth of maximum ALT. Here, it is more likely to get thawed out during the next summer, contributing to the small, but recurring, annual releases. Cryosuction has previously been observed to be of importance in a polygonal tundra model with analysis of solute transport (Jafarov et al., 2022), and our results here suggest the importance of cryosuction

not only applies to predominantly flat terrain with microtopography, but also within hillslope systems where the thaw front progressively increases. This indicates the potential for cryosyction-induced dispersion of solutes, highlighting the importance of representing this process in climate warming scenarios. Note that explicit dispersivity was omitted in the model (see Section 2.2) and is therefore not responsible for this observation. We argue that, had dispersivity been included, this result might have been obscured, potentially leading to a misinterpretation of cryosuction's role in solute transport as substrate-dependent

dispersion.

  Under the assumption of an abrupt increase in ALT due to one very warm year ($T_{avg}$ +3, +5), the transport of ancient, currently frozen carbon tracer becomes more similar to the transport of buried carbon. Due to the short-term but significant increase in ground temperatures, the entire injected tracer mass is mobilized as soon as the active layer thaw front reaches the depth of the carbon source. The rapid export of this carbon source suggests that the residence time is short leading to decreased

potential for microbial mineralization.

  A thought experiment in which we calculated potential microbial mineralization as part of the post-processing of environmental soil temperature and moisture shows that the high liquid saturation associated with rapid transport also leads to reduced or absent mineralization throughout most of the active layer. Under generally or future drier conditions (Painter et al., 2023), mineralization rates in the active layer may be higher, especially in the upper part, where fully saturated conditions appear

mostly in the beginning of the season. Translating this result to the transport patterns in the gradual and abrupt warming scenarios, this may imply that permafrost carbon can experience export in the form of DOC before it gets a chance to be mineralized into $CO_2$. However, these results rely on a threshold assumption for the effect of soil moisture on mineralization (Wen et al., 2020), which is uncertain and requires increased attention in future research to better constrain microbial mineralization in permafrost affected landscapes.

The simulations conducted in this study are based on environmental and general soil physical conditions observed in the Endalen valley, Svalbard. However, the model is limited by the availability of observational data for e.g., soil properties and subsurface conditions such as preferential flow paths. Furthermore, it does not account for a full 3D representation of the area as there is no additional discretization in the transverse direction. We partly address this by representing a converging slope model setup, where the cell width in transverse direction varies depending on the distance in longitudinal direction. This way, 455 the surface area of the hillslope is preserved, and it is possible to accurately represent water and energy balances as well as infiltration and evaporation rates. This approach has previously been applied by Gao and Coon (2022). The subsurface material used in our simulations represents a highly conductive soil that allows for fast groundwater seepage from up- to downhill. At the field site, this is a realistic assumption as the ground is dominated by gravely soil overlain by a thin organic layer, but may be different for hillslope systems with finer material such as silts and clays.

While the Endalen valley represents a single case within the context of high Arctic hillslope systems underlain by continuous permafrost (Brown et al., 2002), similar systems are widespread across the Arctic, making this site a valuable example for studying permafrost carbon transport. The slope of approximately 12° in our study area is representative of gently sloped terrain in permafrost landscapes, which accounts for about 20% of the total permafrost region and is commonly found in areas such as Greenland and parts of Canada in addition to Svalbard (Hamm and Frampton, 2021). However, our findings may not be directly 465 transferable to other permafrost settings, such as flat terrain, where differences in groundwater flow velocities, saturation levels, and small-scale processes like cryosuction could significantly alter transport dynamics, especially in landscapes influenced by microtopography, such as polygonal tundra. Additional studies across a variety of terrains, supported by reliable field observations, are necessary to gain a more comprehensive understanding of lateral carbon transport in permafrost environments.

Enhancing our understanding about potential lateral export of carbon is crucial for reducing the uncertainty surrounding 470 the permafrost carbon feedback and the fate of permafrost carbon (Intergovernmental Panel On Climate Change (IPCC), 2022). Advances in research on the permafrost carbon feedback have been made by simulating production, transport, and transformation of carbon in the contemporary carbon cycle across landscapes using land surface models (Bowring et al., 2019, 2020) or by characterizing fluvial derived organic carbon in rivers through radio carbon dating (Wild et al., 2019). However, a mechanistic representation of in-situ mobilization of permafrost carbon in models is generally lacking, and have 475 only begun to be developed (Rawlins et al., 2021; Mohammed et al., 2022). In this study, we address this process by simulating the thaw out and mobilization of permafrost carbon with a 2D hillslope model, only accounting for non-reactive transport of a generic tracer. While our model gives us valuable insight into the mechanisms causing transport to be slower or faster, a full depiction of carbon is not included since representing carbon specifically would require extensive in-situ soil carbon

characterization, and representation of biogeochemical reactions and processes such as microbial mineralization, which are currently not part of the model setup.

Increasing the terrestrial-to-aquatic transport of permafrost carbon can influence the further processing of such carbon and its effects on the oceans. Generally, at least half of the carbon input from terrestrial sources into the aquatic systems is expected to be released to the atmosphere within the riverine system (Cole et al., 2007). However, field sampling of dissolved organic matter (DOM) in Alaska has shown that the organic matter available for export may have low biolability (Mutschlecner et al., 2018). This suggests that the more stable ancient permafrost carbon may indeed experience export into rivers and the ocean without getting entirely mineralized beforehand.

While this study mostly focuses on the potential of carbon transport, the results can also provide valuable insights into contaminant transport, in order to evaluate the importance of hydrology in contaminated sites in permafrost regions and how contaminants may move within the subsurface. Contaminated sites with toxic substances are likely to impact livelihoods and ecosystems in the Arctic and require increased attention (Langer et al., 2023). The modeling approach presented here offers a basis for further development for both carbon transport as well as contaminant transport.

To address these problems more comprehensively, the use of a reactive transport model will help to disentangle the aforementioned complex interactions (e.g. Mohammed et al., 2022). For example, biogeochemical carbon dynamics can be incorporated into the model by assigning chemical properties to the tracer as well as describing microbial mineralization processes based on environmental variables such as soil moisture and temperature. However, the complexity of these processes alone adds significantly to the existing complexity of representing freeze-thaw dynamics in permafrost regions and poses a considerable computational challenge.

Permafrost carbon stock databases such as the one by Hugelius et al. (2014) can further be used to model site-specific conditions based on the presence of permafrost carbon in different depths. In the same way, chemical properties can be assigned to contaminant species. Integrating transport as well as biogeochemical reactions of the dissolved compounds in permafrost hydrological models will then help to further constrain global climate models and reduce the uncertainty surrounding the permafrost carbon feedback.

## 5    Conclusions

The fate of permafrost carbon in a changing climate is an essential part of the permafrost carbon feedback loop. Considerable uncertainty in predicting the effects of permafrost thaw evolve around the uncertainty in how much of the carbon currently immobilized in permafrost gets released vertically to the atmosphere as green house gases and which fraction of it might get transported laterally towards surface water and eventually to the ocean. In this study, we addressed the transport component of this question. For solute transport in the seasonal active layer, we find that:

1.  A tracer representing active layer carbon experiences different transport velocities depending on its location within the active layer.

2. Active layer carbon close to the surface experiences both rapid surface transport but also prolonged subsurface transport due to unsaturated conditions close to the ground surface as the thaw front propagates deeper in the ground.

3. Active layer carbon close to the bottom of the active layer experiences faster subsurface transport and no surface transport due to saturated conditions at depth in the active layer.

4. Carbon released near the bottom of the active layer may experience limited mineralization prior to export due to the combination of relatively rapid transport with groundwater and limited microbial mineralization rates due to saturated conditions.

In a warming climate, solutes such as carbon released from the permafrost are likely to experience different transport patterns. Our main findings for simulations of a warming climate are:

1. Under the simulated environmental and soil physical conditions in this study, a gradual warming of air temperature with climate change may lead to a gradual release of currently frozen carbon pools as the active layer deepens. While moving vertically downward over time, small amounts of carbon can get released laterally every year, but a rapid release of large amounts of carbon at once are not expected. This may favor the the in-situ mineralization of the carbon prior to export.

2. Abrupt active layer deepening, on the other hand, may lead to the rapid release and transport of soil organic carbon (SOC) in the form of dissolved organic carbon (DOC), leaving little time for microbial mineralization when the soil is unfrozen.

3. Seasonal thaw and freeze-up enhances vertical movement of solutes by cryosuction, which can enable an earlier than expected onset of release near the permafrost table.

*Code and data availability.* The Advanced Terrestrial Simulator (ATS) (https://doi.org/10.11578/DC.20190911.1; Coon et al., 2019) is open source under the BSD 3-clause license and is publicly available at https://github.com/amanzi/ats (last access: May 2023) (Coon et al., 2016). Simulations were conducted using version 1.4. Forcing datasets and input files are available at https://github.com/a-hamm/hamm-et-al-2024-permafrost-transport. Weather data to create the forcing dataset were downloaded from https://www.unis.no/resources/weather-stations/ (The University Centre in Svalbard, 2023) and from https://seklima.met.no/observations/ (Norwegian Climate Service Centre, 2023). The drone survey images for the hillslope in Endalen is available at https://doi.org/10.5281/zenodo.8279263 (Schytt Mannerfelt, 2023).

## Appendix A: Mesh and Boundary Conditions

To represent the main topographical characteristics of the hillslope site, a 2D variable-width domain is used, extending 1040 m along the highest flow accumulation path from the upper boundary (376 m relative elevation) to a groundwater spring at the outlet (0 m relative elevation) (Fig. A1). An important consideration in the model design is ensuring that the area is represented because it allows for net precipitation and resulting runoff to be obtained using readily available hydro-meteorological data from nearby weather stations. The surface energy balance model in ATS (Atchley et al., 2015) allows for calculation of evaporation based on weather forcing (incoming shortwave radiation, vapor pressure, wind speed, and air temperature) yielding runoff which can occur as surface overland flow or infiltration and groundwater recharge depending on if the ground is frozen or not. Thereby both surface and subsurface water accumulates and converges to the transport section of the model, consistent with what would be expected for a convergent hillslope.

The domain is numerically meshed following a variable size approach, where the width of mesh elements vary along the $y$-axis (Fig. A1b; cf. also Fig. 3). This allows for the hillslope area to be captured, which in turn allows for calculating the surface energy balance and enables hydrological and thermal processes to be represented across the hillslope without the computational expense of a fully discretized 3D model, preserving surface and subsurface flow convergence. The plan shape and profile curvature of the hillslope, which are dominant topographic controls on flow (Dunne and Black, 1970; Anderson and Burt, 1978), are preserved by the variable-width mesh approach. The use of a variable-width representative-hillslope mesh is a well-established modeling approach in hillslope hydrology. The sides of the hillslope are determined from topography by gradient descent, aligning with surface flow and allowing for no-flow boundary conditions on the sides. This approach has a long history in hillslope hydrology (e.g. Fan and Bras, 1998; Troch et al., 2003; Hazenberg et al., 2015) and has been successfully compared to site data (e.g. Paniconi et al., 2003; Hazenberg et al., 2015, 2016; Loritz et al., 2017)

The domain mesh is divided into three sections (cf. Fig. 3):

1. **Upper Slope** (0-1000 meters in the $x$-direction): The horizontal resolution of cells in this area are $\Delta x = 5.8$ m with varying cell width in the $y$-direction. This section acts as the main water source representing convergent flow to the Transport Section below. The coarser resolution compared to the other sections helps reduce the computational costs for these process-rich simulations. Note that horizontal resolution in the upper slope area is reduced by a factor of four ($\Delta x = 23$ m) for the climate change scenarios due to the high computational costs involved in those simulations which span a longer time period. The upper slope, due to its low SOC abundance (Weiss et al., 2017), is not considered for solute transport in this study.

2. **Transport Section** ($x = 1000$-1020 meters): This is the main area of interest for the model (Fig. A2). The horizontal resolution of cells is increases to $\Delta x = 0.5$ m to account for the more detailed hydrological processes occurring near the foothill, particularly for representing solute transport. This area in the Endalen Valley also contains more SOC (Weiss et al., 2017), and is the part of the model domain where the simulated tracer is injected (see Fig. A2) and breakthrough is observed. The finer resolution allows for more accurate modeling of transport processes.

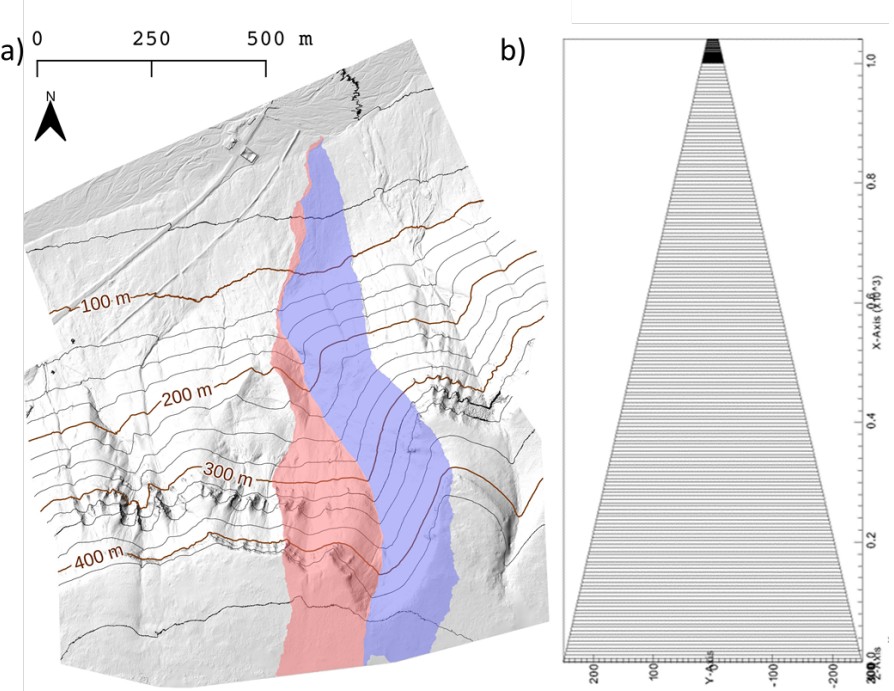

**Figure A1.** (a) Elevation contours showing the hillslope in Endalen, Svalbard which is generally topographically convergent with on average North-East (red) and North-West (blue) slope inclinations. (b) Model domain view from above with numerical mesh, which is based on the main geometric characteristics of the hillslope, including its length, area and slope elevation. The transport section has a much finer mesh resolution at its northernmost tip (cf. Fig 3).

3. **Buffer Zone** ($x = $ 1020-1040 meters): The horizontal resolution of cells is increased to $\Delta x = 1$ m in this area, which acts as a buffer zone where water can accumulate in the subsurface and on the surface as ponded water and overland flow. An outflow boundary condition is prescribed on the surface cell of the last column of the model, which allows ponded surface water to leave the system. This zone is included to reduce boundary effects and is not central to the analysis of the transport study in the transport section. It is implemented based on approaches established by Jan et al. (2020) and Hamm and Frampton (2021).

In the $y$-direction, the model domain width extends from 20 m at the lower end of the hillslope to 500 m at the upper boundary, resulting in a triangular shaped domain with variable width cells when viewed from above (Fig. A1b). In the $z$-direction, the domain follows a topographic profile based on the path of greatest flow accumulation in the hillslope extracted from a high-resolution DEM. This path corresponds to the demarcation between the North-East and North-West slope inclinations (Fig. A1a). The total vertical depth extent of the domain is 40 m, discretized into cells with variable vertical resolution. Near the surface, the resolution is set to $\Delta z = 0.02$ m (Fig. A2) while deeper layers have a progressively coarser resolution up to

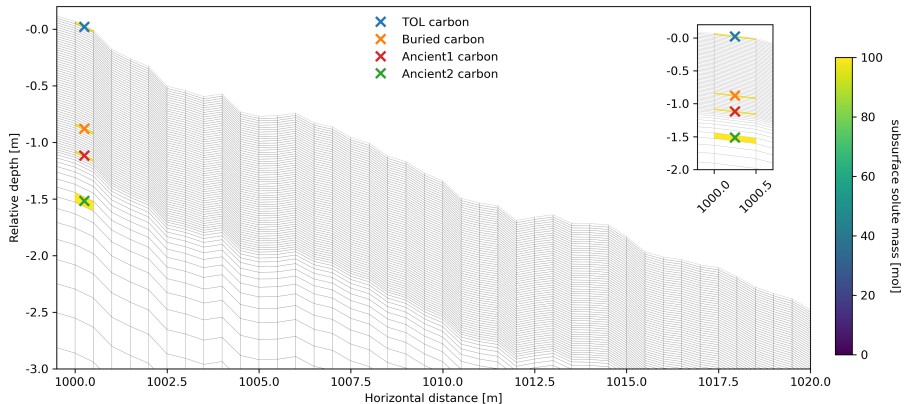

**Figure A2.** Subsurface cross-sectional view of the injection cell depth locations in the Transport Section of the model domain.

$\Delta z = 2\,\text{m}$. The fine near-surface vertical cell resolution is useful because it helps resolve the seasonal dynamics of the active layer and freeze-thaw processes. The deeper coarser cells are generally frozen within the permafrost and are less dynamic.

Hydrological and thermal boundary conditions are prescribed on all external model domain boundaries. Vertical sides are assigned zero-flux for both water and heat. At the base of the model domain, a no-flow boundary condition is imposed, and a constant temperature of -6.6°C, following borehole observations (Christiansen et al., 2020). On the top boundary, a surface energy balance model is used (Atchley et al., 2015), which serves as a source and sink for water and heat to the subsurface, and is applied using available hydro-meteorological data from a nearby weather station in Adventdalen, approximately 3 km northeast of the study site. The data used is air temperature, vapor pressure, incoming shortwave radiation, wind speed, and precipitation. Precipitation data is supplemented by the Longyearbyen airport weather station, 10 km northwest of the study site. Surface runoff occurs whenever the subsurface is fully saturated, causing water to pond and flow on the surface. Thereby, water discharges from the model whenever surface ponding occurs on the downslope boundary of the domain, representing a groundwater spring or overland flow.

The weather forcing datasets contain daily values for all variables. Except for precipitation, a day-of-year average from the years 2013 to 2020 is used to describe a typical yearly cycle representative of present-day weather conditions. Precipitation is split into rain and snow based on air temperature (rain for T > 0°C, snow for T $\leq$ 0°C) and distributed throughout the year using a frequency-density function to match the statistical characteristics of observed rainfall (Magnússon et al., 2022; Hamm et al., 2023). This method avoids distributing the total precipitation evenly over the respective rain and snowfall periods, which would result in small daily precipitation rates that promote evaporation and limit infiltration, reducing soil moisture, and no days with absolute zero precipitation. The total sum of precipitation equals the average sum recorded between 2013 and 2020, with 84 mm of rain and 115 mm of snow. By using a frequency-density function, the resulting distribution more closely resembles natural rainfall variability, ensuring a realistic representation of precipitation events.

The soil physical properties (cf. Table 1) were defined to resemble highly water-conductive material, representative of either moss or a gravely soil. We used representative values for water retention parameters using the van Genuchten model, while

assigning literature values for porous media with high permeability and porosity. This allows the model to capture the key hydrological processes without overly specifying for either moss or gravely soil, thus providing flexibility in representing both substrate types.

## Appendix B: Field observations

Qualitative observations in the field further include the seasonality of changes in saturation, visual interpretation of groundwater flow velocities using a dye tracer in the late thawing season in September 2022, and description of the soil material.

Due to snow melt at the onset of the thawing season, the hillslope landscape is largely inundated as the active layer is still frozen or has only just begun developing, thereby inhibiting infiltration. As the thaw season progresses and the active layer deepens allowing for more significant infiltration, the surface becomes largely dry with a few highly saturated areas. Roughly in the middle of the hillslope there is a small ephemeral stream. There is also a perennial snow-patch near the upper water divide contributing to snow melt.

The surface of the hillslope is largely covered by mosses and short grasses, or by exposed gravel. The organic layer is not extensive and usually only reaches 2–10 cm deep based on spot samples and visual interpretation. Below, the gravely layer begins with an unknown depth and the stratification below these two layers cannot be determined.

Tracer tests with liquid Rhodamine and Fluorescein conducted in September 2022 were used to qualitatively assess the order of magnitude of subsurface flow velocities. Tracers were injected at various depth between 0.5 and 1.2 m below the surface and at various distances (1, 5, 10, and 20 m) from an ephemeral groundwater spring (see Fig. B1). While the exact transport velocities and full BTC could not be determined, we observed a rapid initial breakthrough of the order of several minutes to a few hours. This leads us to believe that the subsurface material is coarse (gravely) and allows for fast transport.

## Appendix C: Climate change scenario post-processing

To quantify mass transported from the injection location in the climate change scenario simulations, the visual output was used to extract the total solute mass from (1) the injection cell and (2) the entire vertical column in which the tracer was injected. Visual output from the model includes values for all cells throughout the entire mesh for each time step of the simulation. We analyze the tracer mass integrated over the entire column due to the vertical tracer movements observed in Fig. 11. Visual output is generated every 20 days throughout the simulation period. Despite a high output precision, rounding and vertical integration leads to values slightly higher than the initial injected mass (100 mol). Maximum values were 100.35 mol, 104.27 mol, and 100.65 mol in the columns for the RCP4.5, the RCP8.5, and the abrupt warming scenarios, respectively. For better visualization, solute mass values have been truncated to 100 in Fig. 10a and d, and Fig. 12a and b during the post-processing procedure.

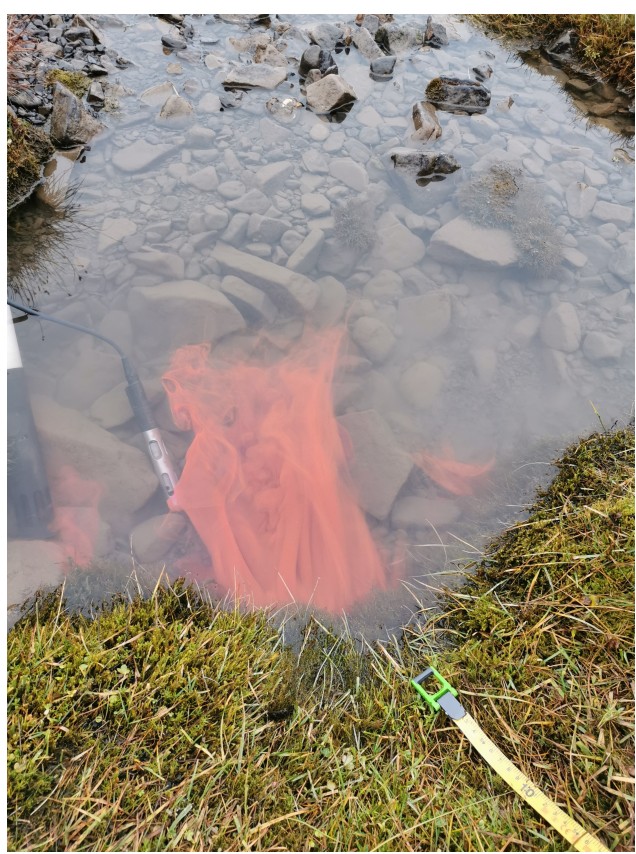

**Figure B1.** Rhodamine tracer emerging at the groundwater spring on the downslope side of the hillslope area in Endalen valley during tracer experiments conducted in September 2022. The liquid tracer was injected 2 m uphill in groundwater at approximately 0.5 m depth. First arrivals where observed only after a few minutes after injection.

## Appendix D: Evaluation of Model Configuration

To ensure that our model results are not influenced by arbitrary choices regarding boundary conditions, parameters, or mesh discretization, we conducted a thorough assessment to evaluate the adequacy of these modeling decisions.

A domain depth of 40 m was used to ensure that the depth of zero annual amplitude remains unaffected by the bottom boundary condition. As shown in Figure D1, the depth of zero annual amplitude is approximately 15 m. This demonstrates that the surface signal penetrates to this depth, beyond which interannual variations in surface forcing have negligible influence on ground temperatures.

Furthermore, we demonstrate that a spinup time of ten years is sufficient by analyzing the day-of-year surface temperature

differences between two consecutive years throughout the spinup period. As shown in Fig. D2, these differences are initially large (up to 4°C) but decrease steadily, becoming negligible by year five. After five years, the average difference throughout the year is less than 0.1°C, indicating that the system has reached a cyclic steady state.

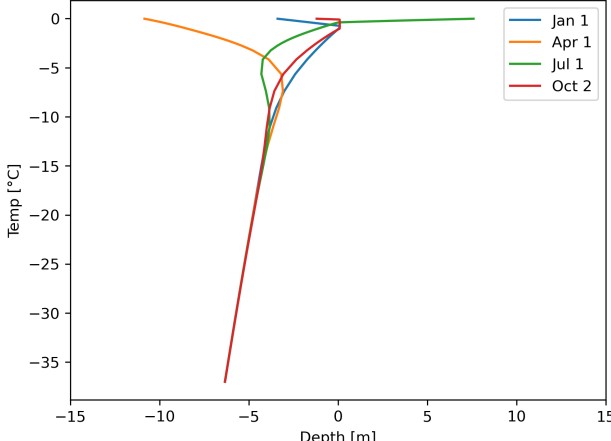

**Figure D1.** "Trumpet curves" showing the seasonal development of the soil temperature profile with depth for four given times. The profiles indicate that the depth of zero annual amplitude is located at approximately 15 m.

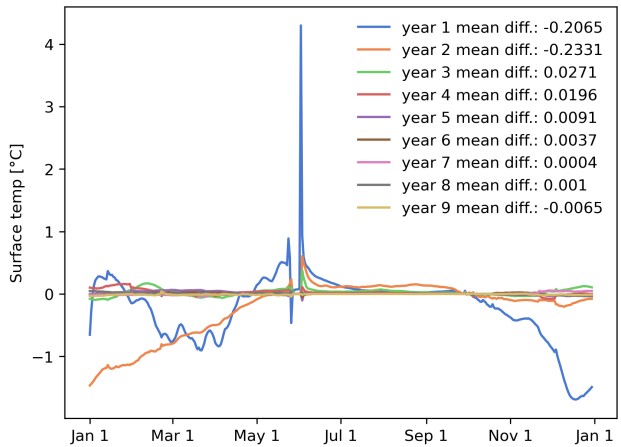

**Figure D2.** Surface temperature evolution in the region of interest throughout the spinup. The legend indicates the mean temperature difference throughout the year and suggests that after four years of spinup, the temperature differences are minimal with slight variations in daily absolute maximum differences. Overall, the temperature evolution shows that ten years of spinup are sufficient for this model setup.

For the climate change scenarios (Fig. D3), air temperature was the only variable that changed and increased over time, while the bottom boundary temperature remained constant at -6°C throughout the 50-year simulation. To verify that this does not impact transport processes in the active layer, we conducted 1D column simulations using the same forcing dataset but with two different bottom boundary conditions: (1) a constant temperature boundary condition and (2) a constant heat flux boundary condition. The results show that the choice of bottom boundary condition has minimal influence on the temperature and associated processes in the active layer during years 1–40 and only minor effects in years 40–50 (never exceeding a

difference of one mesh cell). Therefore, we conclude that the constant temperature bottom boundary condition does not affect
the results presented in Section 3.2.

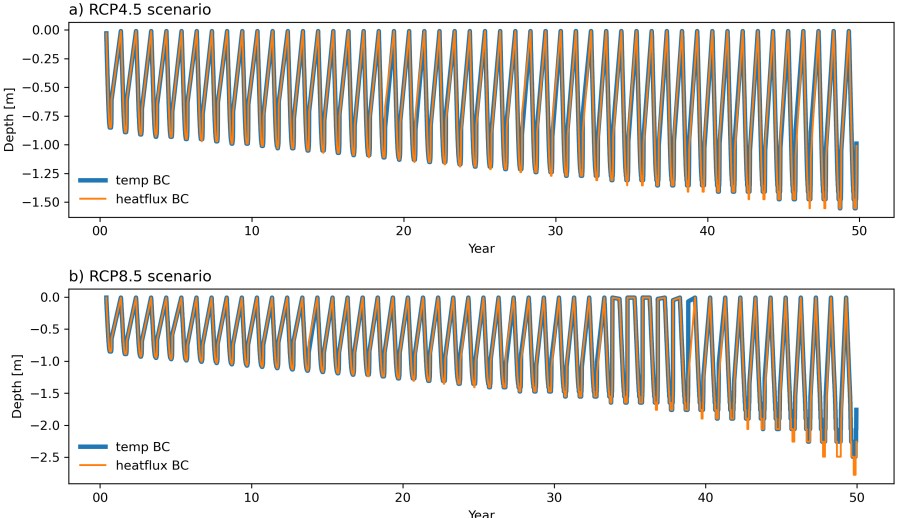

**Figure D3.** Thaw depth progression in simplified 1D column models using the same forcing as the full 2D transect in the main text for the (**a**)
RCP 4.5 and (**b**) RCP 8.5 scenario, with each a constant temperature boundary condition (blue) and a heat flux boundary condition (orange)
prescribed at the bottom of the domain.

As described in Appendix A, the horizontal resolution of the mesh in the upper slope area is coarser than the resolution in
the transport section in the down slope area. This approach is used because the upper slope area is not central to our study
due to its low carbon content, and is only needed for representing the hillslope hydrology. To evaluate possible effects of the
mesh resolution of the model in the upper slope area, we ran three simulations with progressively increasing column widths:
the finest resolution used 5.8 m per column, the medium resolution had 12 m per column, and the coarsest resolution used 23 m
per column.

The differences between these three cases is shown in Fig D4, where it can be seen that the mesh resolution in the upper
slope area slightly influences the tracer breakthrough curves. We attribute these differences to how mesh resolution affects
moisture transport and, consequently, the thaw rate. Faster thawing in the higher resolution cases, likely driven by differences
in soil moisture, leads to earlier tracer release compared to the coarsest scenario. However, the relative differences between
travel times (short for buried carbon and prolonged for TOL carbon) are similar across all cases. For seasonal simulations of
carbon transport, we opted for the finest resolution (5.8 m) in the upslope area, as it may most adequately capture the seasonal
dynamics.

A finer resolution leads to more responsive freeze-thaw and moisture release dynamics, causing slight differences in the
active layer thaw rate, which leads to earlier tracer release compared to the coarsest resolution case. However, the relative
differences in travel time behavior between TOL and buried carbon are similar across the three cases. For site-specific studies

aiming to reproduce field measurements, accurate replication of field-measured thaw rates would be important. However, as the purpose of this study is to investigate relative effects of solute release in different depths in a semi-generic model setup representative for arctic hillslopes, the precise timing of thaw rates are not critical. The coarsest mesh resolution is used for the climate change simulations which are computationally more demanding due to their long simulation times, encompassing 50 years.

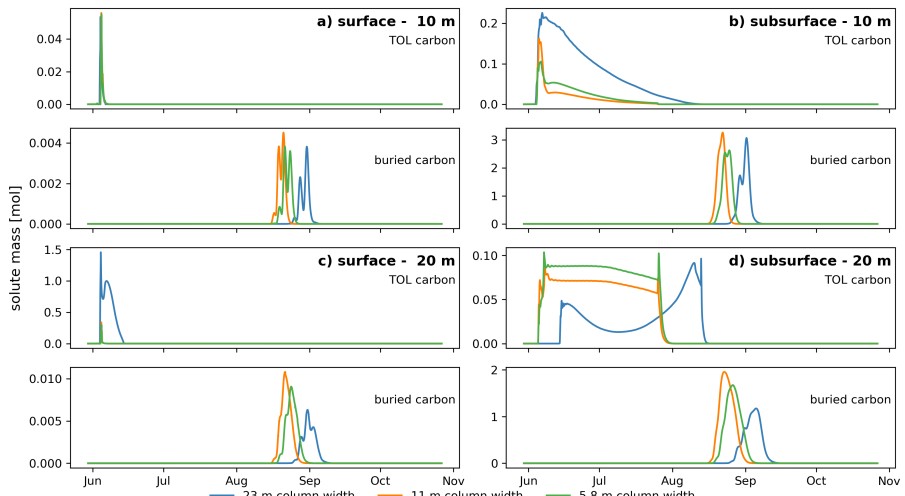

**Figure D4.** Breakthrough curves similar to those shown in Fig. 5 without the surface transport contribution but including breakthrough curves for all tested mesh resolutions for the upper slope area. The different panels show breakthrough curves in the surface (a and c) and the subsurface (b and d) for each, the TOL carbon (top panel) and buried carbon (lower panel) for each of the three cases and at both observation point distances.

To verify that the fine discretization in the transport section was sufficient to resolve the flow dynamics, we repeated the spinup flow simulation with a "superfine" mesh that started with the fine mesh of the seasonal simulations and halved the cell size in the transport section and in the downstream buffer zone, resulting in $x$-spacings of $\Delta x = 0.25$ m in the transport section and $\Delta x = 0.5$ m in the buffer zone. Simulated temperature and $x$-velocity using the fine and superfine meshes are nearly indistinguishable (Fig. D5 and D6).

**Appendix E: Carbon distribution in Adventdalen, Svalbard**

Slopes in the greater Adventdalen area are generally often characterized by the occurrence of solifluction processes. Solifluction can lead to the burial of organic matter into deeper soil layers, enriching the organic carbon content in samples taken from different tributaries of the Adventdalen valley (such as the Endalen valley). Sampling soil cores in Adventdalen by Weiss et al. (2017) has revealed that SOC 0–100 cm stocks in solifluction affected areas are on average $20.7\,\mathrm{kg\,C\,m^{-2}}$, which is much higher than the landscape-level mean of $4.2\,\mathrm{kg\,C\,m^{-2}}$ based on landform upscaling.

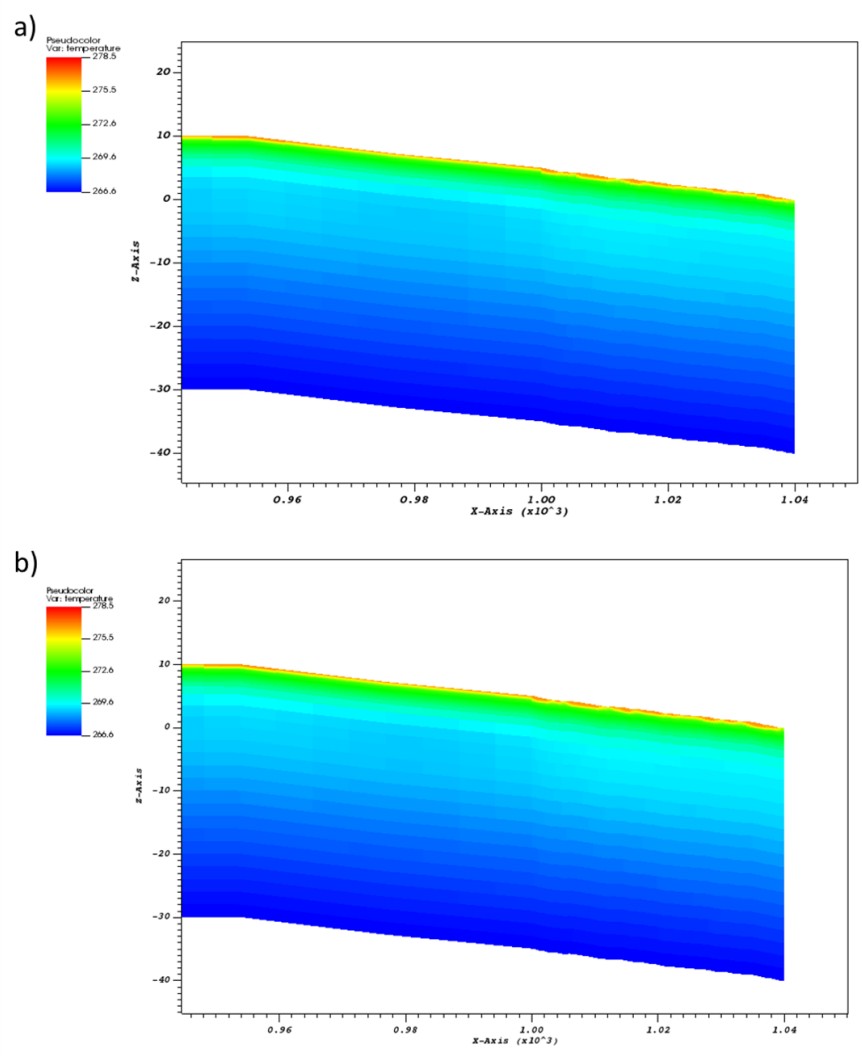

**Figure D5.** Temperature in the transport section obtained from numerical simulations with a (a) fine resolution mesh $\Delta x = 0.5\,\text{m}$ and (b) superfine resolution mesh $\Delta x = 0.25\,\text{m}$. Differences are nearly indistinguishable.

Fig. E1 shows examples of four profiles from two solifluction areas in Adventdalen. The mean SOC 0–100 cm stock of these profiles is $23.4\,\text{kg}\,\text{C}\,\text{m}^{-2}$, slightly higher than the reported mean stock for solifluction areas reported in Weiss et al. (2017). Profiles are generally deep before reaching bedrock or regolith ($\geq 70\,\text{cm}$). They have a thin top organic layer ($\%C > 10$) between 4.5–9 cm (mean 6.4 cm), which stores only 3.0–9.9% (mean 5.5%) of the total SOC stock in the top 100 cm of the profiles. The remainder is stored in the mineral active layer, with negligible amounts in the permafrost layer. Only in profile T3-1, the frost table was reached at 70 cm on the sampling date of July 1, 2013 but the active layer depth is certainly much

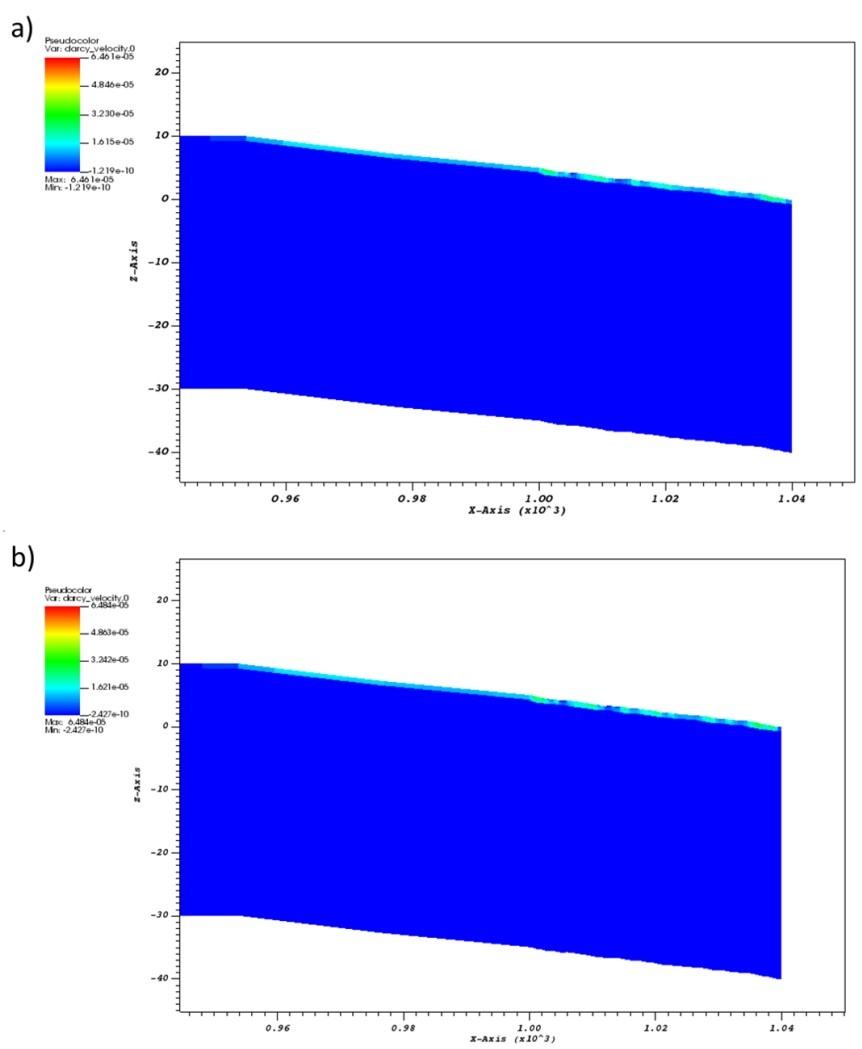

**Figure D6.** Specific discharge (Darcy velocity) in the transport section obtained from numerical simulations with a (a) fine resolution mesh $\Delta x = 0.5$ m and (b) superfine resolution mesh $\Delta x = 0.25$ m. Differences are nearly indistinguishable.

deeper. Two profiles show some C-enrichment at greater depth, likely the result of burial by solifluction processes (T3-1 at
36–60 cm and T5-8 at 18–30 cm). T5 profiles have a larger volume of large stones in their profiles (up to 50%), which reduces total SOC 0–100 cm stocks compared to T3 profiles (no large stones, not shown).

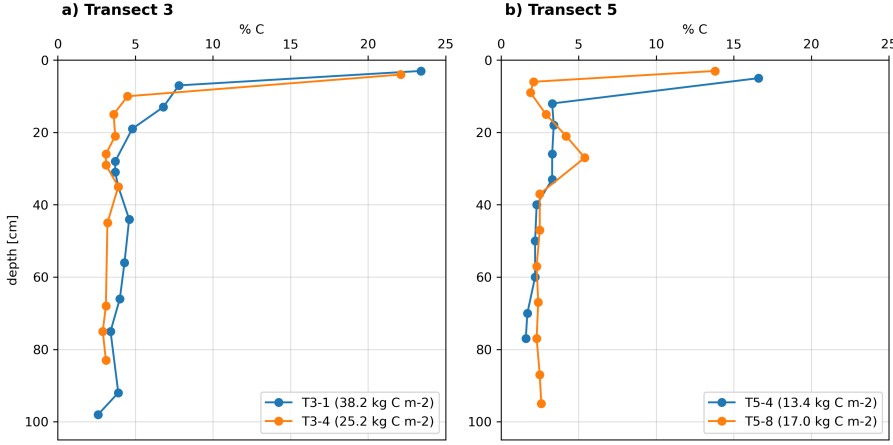

**Figure E1.** Organic carbon content (%) vs depth (cm) in four examples of soil profiles from the Adventdalen area. Transect 3 (**a**) is close to the Endalen valley, which represents the study site in this paper. Transect 5 (**b**) is located close to the Bolterdalen valley, ∼6.5 km southeast of the Endalen valley. Sampling depths vary depending on the sampling method and quality of the soil profiles retrieved. The total soil organic carbon storage in each profile is denoted in the legend entry next to the profile ID.

## Appendix F: Photogrammetry processing

After surveying the hillslope in Endalen valley using a drone, we process the survey in the photogrammetrical suite Agisoft Metashape version 1.8.4 using the default processing settings. We register the survey data to the 2009 Digital Elevation Model
(DEM) made by well-georeferenced aerial images by the Norwegian Polar Institute (Norwegian Polar Institute, 2014), using Iterative Closest Point (ICP) registration in CloudCompare version 2.12, leading to a mean vertical bias of 0.03 m, and a mean absolute point-to-point distance (which we consider the registration uncertainty) of 0.74 m. We assume minimal internal distortion in the survey due to the inclusion of oblique images (James et al., 2017) and therefore consider the affine transformation of the ICP registration sufficient for the accuracy that is required for the study. With a survey diameter of 990 m, this location
uncertainty is approximately equivalent to an angle uncertainty of 0.04° ($\tan^{-1} \frac{0.74}{990}$).

*Author contributions.* AH and AF designed the study with help from AAM, SLP, and EC. ESM surveyed the study site by drone and processed the digital elevation model. AH performed the model runs and analysis with help from AF, SLP, and EC. AH and AAM performed the post-processing of the model results to evaluate potential microbial mineralization rates. AH and AF wrote the manuscript with contribution from all co-authors.

*Competing interests.* The authors declare that they have no conflict of interest.

*Acknowledgements.* Financial support for A.H. and A.F. for this research has been supported by the Svenska Forsknings-rådet Formas (grant no. 2017-00736) and by the Bolin Centre for Climate Research. Computations were enabled by resources provided by the National Academic Infrastructure for Supercomputing in Sweden (NAISS), partially funded by the Swedish Research Council through grant agreement no. 2022-06725. S.L.P and E.C. acknowledge support by the Next-Generation Ecosystem Experiment–Arctic (NGEE Arctic) project. The NGEE
Arctic project is supported by the Office of Biological and Environmental Research in the U.S. Department of Energy Office of Science. The authors also thank Peter Kuhry for providing us with field-based observations and measurements of carbon distributions in Adventdalen.

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
