# Peer review of "Model-based analysis of solute transport and potential carbon mineralization in the active layer of a hillslope underlain by permafrost with seasonal variability and climate change"

_EGUsphere, 2024_

## Author Comment (AC1)

**REVIEWER 1**

Comments by the reviewer are in black text

Our responses are in blue text

*Where applicable, proposed changes to the revised manuscript are in blue and italics*

GENERAL COMMENTS:

The manuscript deals with an important and debated problem, the permafrost carbon feedback. More specifically, it uses a numerical simulation approach for assessing solute transport in a permafrost catchment, in order to study the impacts of climate change on DOC export from the soil of the considered watershed. The mechanistic modelling of dissolved species in permafrost affected areas is at the forefront of the current research in cryohydrogeology, and the used tool is a state of the art one. Thus this work is both novel and of great interest for cryospheric sciences.

Meanwhile, critical weaknesses affect its reliability. First of all, the study focuses on a specific site, while the claimed aim of the work is to draw general conclusions about lateral transfers in permafrost regions. This is contradictory, and if the authors want to produce a general study then the problems of upscaling and transposing to different biogeoclimatic permafrost contexts must be carefully dealt with. I have also a number of technical concerns about the set up of the performed numerical simulations, for instance the poorly justified pseudo-3D approach, the absence of any convergence study, and some problems in the definition of boundary conditions. Finally the discussion of the obtained simulation results is rather weak and should be strengthened. All these points are detailed in the specific comments below.

Overall I think that the manuscript cannot be published in its present form. A thorough effort is needed for making it more solid and meaningful, including possibly new computations and results depending on the answers to some of my concerns (e.g.: numerical convergence, relevance of top and bottom boundary conditions). So I recommend that a major revision of this work should be undertaken prior to reconsider whether or not it should be published in The Cryosphere.

We thank the reviewer for their thoughtful review. In response, we plan to make several modifications to enhance the manuscript. Our primary focus will be on providing a thorough justification of the meshing strategy, which was not adequately addressed in the original version. Additionally, we will clarify the applicability of our results to a broader pan-Arctic context, improve the clarity of the figures, and include additional descriptions where necessary. We are confident that these revisions will significantly strengthen the manuscript.

SPECIFIC COMMENTS:

1. - Title and l 8-9: The title refers to 'a permafrost catchment', while abstract draws conclusions for 'permafrost regions'. There is no *a priori* reason that conclusions made from the study of a peculiar catchment should be relevant for permafrost regions as a whole.

We agree that the conclusion that our results apply on a larger scale may be misleading as it is currently written. While it is not possible to generalize our results for the entire northern hemisphere permafrost region, the results from this study can be extended to other permafrost hillslope catchments with similar climatic, hydrological and geological conditions. This site considered in our study is representative of gravely footslopes with a gentle inclination (5-15°, Hamm and Frampton 2021). We will adjust the phrasing in the abstract and will improve the discussion on how representative these results may be in a broader context of similar permafrost hillslope regions.

2. - l 17: Why considering mineralization of organic matter to $CO_2$ only, and not also to $CH_4$? May be due to the peculiarity of the study site. This should be explained.

We included this information in the methods section, rather than in the abstract. To better highlight our decision, we will add more explanation as to why we do not consider mineralization into CH4. The main reason is that the study site is a hillslope and compared to wetlands and peatlands is unlikely to contribute significantly to overall permafrost CH4 release. We will add a reference to this statement in the methods section.

3. - l 27-41: Other questions than PCF maybe related to lateral transfer of dissolved chemical species from the active layer under climate change (e.g.: impacts on river / ocean ecosystems).

We will add a sentence on potential impacts on river ecosystems and the ocean, such as rusting rivers, mercury release, and ocean acidification.

4. - l 71-72: The assumption that DOC may be considered as a non reactive tracer should be discussed here, with relevant references.

A passive tracer is used to numerically represent advective solute transport in the simulations; the conceptualisation used is that a given quantity of DOC is transported by the advective flow field but can undergo mineralisation which is dependent on soil thermal and saturation dynamics. Although this does not account for the full complexity of carbon reactions it embodies an important advancement in coupling soil and groundwater dynamics with potential DOC transport and is a novel contribution of our study. The implications are mentioned in the introduction but further discussed in the discussion section. We refrain from adding it to the introduction section.

5. - l 80-82: Most likely this behavior will be site dependent! For instance not the same for a tundra hill slope with low evapotranspiration or a boreal forest hill slope with high evapotranspiration, depending on the slope itself, etc.

The emphasis on the site-specific hypothesis is missing. We will add it.

6. - l 111: Is Qs sink/source term accounting for ionic exclusion? To say, does the used model take into account the specificity of solute transport with freeze/thaw?

Yes, exclusion of solute from the ice is included, although it is enforced by the nonlinear solver and the formulation, and not in the Qs term. The solute in ATS is represented as a moles of solute per moles of liquid water. This way, when the water in the soil freezes, the solute remains in the unfrozen water fraction of the soil and solute concentration increases to honor mass conservation, which is enforced by the nonlinear solver. Upon melting of the pore ice, the opposite occurs. We propose to add the following text in the methods section to clarify this:

*"Note that solute is excluded from the ice phase by the formulation which represents solute as moles of solute per moles of liquid water and by the nonlinear solver which enforces solute mass conservation. Thus freezing of pore water results in an increase of solute concentration and the opposite is true for melting of pore ice."*

7. - l 113: Why neglecting dispersivity? The errors associated with this simplifying assumption should be discussed.

This study deliberately focuses on lateral transport by advection. Although it would be interesting to also consider dispersion, it is highly substrate-dependent and can not easily be quantified with field observations. Hence, any assumptions of dispersion coefficients would add to parameterisation uncertainty and obfuscate the advection-specific results discovered here. For example, the observation of vertical tracer movement by wetting/drying and cryosuction could have been misinterpreted as dispersivity had it been included. We will add an explanation to the text better explaining what the consequences of this decision are.

8. - l 118-119, « path of highest flow accumulation »: What about lateral transfers from the slopes to the thalweg? For studying the drainage of active layer waters and the associated solute fluxes the choice of focusing on the thalweg line does not seems obvious to me. This should be discussed.

We will discuss these choices more thoroughly in an expanded description of the meshing process (please see next comment)

9. - l 121, « pseudo-3D approach »: I cannot understand the usefulness of varying y-axis width of cells along the x direction. According to Fig.2, there is only one mesh along y-axis at every x position. So this seems to me a 2D mesh (as said in l 169). The Figure 3 and the associated explanations are hard to follow and should be thoroughly improved (e.g.: a large blue area appear on Fig.2, but only green and red area are mentioned in the text.) Whatever the pseudo-3D approach really is, the statement made by the authors that it « allows us to account for thermal and hydrological balances across the entire catchment area without the need for a complex and computationally intensive full 3D mesh »

should be justified. How does the y-axis width varies along the thalweg? Has any comparative study between results of a full 3D approach and results of this pseudo-3D approach been done? It should have been, prior to use the pseudo 3D approach, or at least if it is not practicable due to computation time the approximation should be discussed, as well as the associated errors.

We acknowledge that the justifications as to why we chose the particular meshing approach were not adequately explained. We will revise that section to better provide the justification. The key point here is that the use of a  variable-width representative hillslope is a well-established modeling approach in catchment and hillslope hydrology. In this approach, the sides of the hillslope are determined from topography by gradient descent, thus aligning the sides with the surface flow and allowing for no-flow boundary conditions on the sides. The fact that the width is now variable results directly from the 3D topography and preserves flow convergence/divergence, which is well-established to be important. We propose to change the corresponding paragraph in the revised manuscript is as follows:

"*To capture the main topographical characteristics of the Endalen sub-catchment without the computational expense of a fully three-dimensional model, we created a variable width representative-hillslope mesh (Fig. 2) extending 1040 m along the direction of highest flow accumulation from the catchment divide (376m relative elevation) to a groundwater spring close to the valley bottom (0m relative elevation). The sides of the delineated hillslope are aligned with the topographic gradient and thus the surface flow, which results in variable-width mesh elements. Such hillslope-based approaches to approximating a three-dimensional landscape with a lower-dimensional model have a long history in catchment hydrology (e.g. Fan and Bras, 1998; Troch et al. 2003; Hazenberg et al. 2015) and have been successfully compared to three-dimensional models (e.g. Hazenberg et al. 2015; Paniconi et al. 2003). Importantly, the plan shape and profile curvature of the catchment, the dominant topographic controls on flow (Dunne and Black, 1970; Anderson and Burt, 1978; Freeze, 1971), are preserved by the variable width mesh.*"

We will also expand on the explanation of the mesh by moving mesh-specific information to the Appendix to be able to elaborate more thoroughly without increasing the already lengthy main text significantly. We will add details to the representation of the catchment hydrology, the boundary conditions, and the importance of accounting for all the water in the form of precipitation within the catchment rather than only accounting for a short transect and the precipitation falling onto its surface. We will also add a sentence on the complexity of the processes represented in the model and how this does not allow for a feasible representation in 3D due to computational costs.

10. - l 126-127, « main area of interest »: Why focusing on such a tiny area of 20 m length, while simulations are done on a 1040 m large domain? This important choice should justified and discussed.

As described in the text, the upper parts of the transect are mainly in place to provide the adequate amount of water to the lower parts of the slope, where organic matter can be

expected to exist in the permafrost. We will add to this explanation by highlighting that the rocky nature of the upslope part leads to very low carbon abundance and that carbon transport in this section is not of interest. Only in the lower parts of the slope, where the inclination is not as steep anymore, carbon may be present. We will further add that the computational costs for these simulations were very high and due to the mesh discretization, a longer transect with a higher lateral resolution to observe transport would significantly increase these costs.

11. - l 129-130, « By directing precipitation from the upper slope to the lower areas, we ensure realistic hydrological conditions with flow accumulation towards the valley bottom. »: What is meant here? The physical equations solved by the numerical model do ensure that gravity exerts a vertical descendant driving force on water, so that water flows from top to bottom when gravity dominates. This sentence seems pointless, I recommend to delete it.

This will be part of the reworked mesh-section described in comment #9 that will be added to the Appendix. The sentence will either be deleted or rephrased for clarity.

12. - l 130-131,« This division of the mesh allows for accurate modeling of the thermal-hydrological processes in the catchment. » : Such a statement should be justified.

This will be part of the reworked mesh-section described in comment #9 that will be added to the Appendix.

13. - l 132, « Each column in this mesh area [...] varies in width in the y-direction. »: Why and how? See above the point on the pseudo-3D mesh approach.

This will be part of the reworked mesh-section described in comment #9 that will be added to the Appendix. We will add information on how the mesh has been created and that its shape has been chosen to represent the approximate catchment area derived from the digital elevation model.

14. - l 132 – 141: Here mesh cells dimensions are described, but not justified. In a modelling study relying on PDE spatio-temporal discretizations (e.g.: finite differences method, finite volume methods, finite elements methods, etc), it is mandatory to assess the truncation errors by dedicated convergence studies, designed for the simulation case under concern. The results of such a convergence study should be given here, by means of a upper bound of the truncation errors for the outputs of interest. Only in this way one can be sure that the variations discussed in the numerical results are not simply due to truncation errors. Please include here the results of such convergence study for the case under consideration.

Were the goals of this study to make a precise prediction of an effect of interest or to demonstrate a new discretization method or other algorithmic improvement, it would absolutely be necessary to demonstrate mesh convergence. However, the goals of this

study are to improve our qualitative understanding of how likely previously frozen carbon is to be exported or mineralized and it is clear the broad stroke conclusions will not be sensitive to details of mesh refinement. The mesh used is already highly resolved, especially in the region of interest. Outside the region of interest coarser mesh elements are adequate as those regions are only acting to establish correct flow boundary conditions on the highly refined region. Based on community experience with these types of models and our own experience with dozens of permafrost hillslope simulations, we are confident the mesh is adequately refined for the purpose of the study. To confirm this, we will prepare an alternative mesh with additional refinement and study the effects of mesh resolution on our results. We will add results of those simulations to the Appendix and add a sentence on mesh refinement and sensitivity to the manuscript accordingly.

It's also important to note that mesh size does not affect mass and energy conservation. ATS allows for user defined convergence criteria and error tolerance and enforces conservation of both mass and energy to those tolerances. The relative error tolerance in the present simulations is set to 1e-6 for all variables.

15. - l 134-137: I do not understand why a buffer zone is needed. This should be explained.

This will be part of the reworked mesh-section described in comment #9 that will be added to the Appendix. The buffer zone is used to avoid effects of the outlet boundary condition; please see response to comment #17 below.

16. - l 142-146: All of these sentences look like unjustified statements. Moreover what is exactly stated is not completely clear. For instance what means « preserving the subsurface volume representation of the catchment » and « a natural equilibrium without artificially imposed boundary conditions »? This paragraph should be either deleted or rewritten in a clearer and justified way.

This will be part of the reworked mesh-section described in comment #9 that will be added to the Appendix. The confusing paragraph will be removed and the text will be clarified.

17. - l 147-148: « The vertical sides of the model are assigned zero-flux boundaries for water » this is questionable, especially for the outlet vertical boundary. Are there any field observation that can be used to justify this choice? This should be discussed.

It is important to note that the outlet boundary is not closed to flow. As noted in the original manuscript, water expressing on the surface is removed. Thus, by closing the subsurface on the outlet, water flow is forced to the surface where it is removed. This can create an artifact in the immediate vicinity of the outlet, which is why we use a buffer region and do not analyze results in that region. By previous experience (Jan and Painter, 2021; Hamm and Frampton, 2021) and physical arguments we know this

boundary condition produces physically correct behavior away from the immediate vicinity of the boundary. We will improve the description in the main text accordingly.

18. - l 151, « in line with borehole observation in Svalbard »: This is important, since it is likely the reason why a 40 m thickness as been chosen for the modeling domain. Please add a figure with the mentioned soil temperature profile evolution, as well as a discussion for explaining in which way these data where used for choosing not only the bottom thermal boundary conditions but also the domain thickness.

We propose to add a plot with trumpet curves showing that the seasonal surface temperature signal does not penetrate the ground deep enough to affect the bottom boundary condition. The maximum annual temperature variations at depth -15 m are 0.1°C, indicating this is a reasonable approximation of the depth of zero annual amplitude (DZAA). Also, see our reply to comment #23 on why we are confident that a 40 m deep domain is sufficient.

[Figure]

*Fig. 1: Trumpet curve plot indicating that the 0 annual amplitude depth is located at ~15 meters depth.*

19. - l 159-165: I have serious concerns about the chosen methodology for building 'present day-like' precipitation. Why not using a day-of-the-year average like for other forcings? This is not justified. Stating that « the resulting rainfall distribution resembles the variability of natural rainfall throughout the year» is not enough for making the arbitrary, artificial precipitation forcing data set relevant. The use of arbitrary precipitation data may impair the interpretability of the simulation results, so either the 'resemblence' between the artificial data set and the observation

should be quantitatively demonstrated, either the observation data set itself should be used as forcing data.

This approach was previously presented in Magnússon et al., 2022. For precipitation, a day-of-year-average is not representative of the physical system because it will always create very small values for each day in a predominantly dry place like the Adventdalen area, as opposed to many days with no precipitation and a few days with significant precipitation. These small precipitation values would lead to an unrealistically high fraction being evaporated due to the surface energy balance and the overall windy conditions in the valley. Additionally, there would be no days with absolute 0 precipitation, which is not realistic. As described in Magnússon et al. 2022, the daily rainfall amounts are not arbitrary, but represent the frequency-intensity distribution of the natural rainfall variability. Details can be found in the Supplementary Methods VI in Magnússon et al. 2022 and in the included repository with an annotated script that was used to create this precipitation time-series.

20. - l 165-166, « Soil physical properties are defined to resemble highly conductive material. »: This should be justified. Why not moderately conductive material?

We will add a justification to this statement. We assume a highly conductive material in the model because observations in the field site suggest that most of the subsurface material is very gravely or mossy. There are no field observations that describe the subsurface composition in detail.

21. - l 169-170, 'to establish a water table at target depth »: I do not understand. What is the 'target depth'? Are there observational evidences of a water table 'at target depth'?

The spinup process needs to establish an ice saturated subsurface below the active layer. A multistep procedure for doing this with ATS was developed more than a decade ago and is the standard spinup procedure for ATS simulations of continuous permafrost. The first step in that procedure is to establish a hydrostatic water column with the water table close to the surface. We then freeze that water in place with an open boundary condition on top to allow water to be pushed out by the liquid-to-ice volume expansion. That establishes an ice table as close to the surface as possible. Subsequent spinup steps with seasonally varying top boundary conditions then remove any excess ice/water and place the system in the desired ice-saturated state in cyclic equilibrium with the atmosphere. The final step enables the active layer to develop together with a seasonally perched water table at a depth consistent with present day conditions. We propose to replace the confusing sentence with the following text:

"*First, a single column model extending to the full depth of the final 2D mesh is used to establish an ice-saturated subsurface with an ice table near the surface. This was accomplished by freezing an initially hydrostatic and isothermal water column from below keeping an open top boundary to allow the volume expansion of the phase change to push out excess water.*"

22. - l 173-174: Why 10 years? Please provide the criterium used for this choice.

10 annual cycles for the second spinup step is a commonly used length for this initialization step in a 2D problem (e.g. Gao and Coon 2022, Jafarov et al. 2022) to reach an annual steady/cyclic state. In the plot below, it can be seen how the mean annual difference in surface temperature at the surface of the first mesh column in the area of interest between 2 adjacent years during the spinup decreases and reaches a minimal temperature variation after 5 years. We propose to include this plot in the Appendix and include the following sentence.

*"We confirmed that the temperature is in a cyclic steady state at the end of the 10 year spinup period (see Appendix)."*

[Figure]

*Fig 2: Plot of difference in surface temperature at the surface of the first mesh column in the area of interest. Mean differences are denoted as mean annual differences between the current year and the prior year. Despite some variability in the temperature difference, the mean difference in the last year is below 0.01°C with an absolute maximum daily difference of 0.2°C. The system is therefore considered to be in a cyclic steady state.*

23. - l 176-178: The targets of this study are simulations under climate change, while the use of a thermal boundary conditions of constant temperature equal to present day temperature at the bottom of the domain (l 150-151) is not compatible with the simulation of climate change scenarios. Or at least, it implies the assumption that the temperature at 40 m depth is not impacted by surface temperature variations at the considered time scale of 50 years. This is a major concern for the validity of the produced simulation results. In order to demonstrate that this strong approximation does not impair the discussed results, the time variations of temperature in depth, close to the bottom boundary, must

be shown. If they are not negligible, then the simulations should be re-ran with an appropriate bottom boundary condition (e.g.: geothermal heat flux).

A geothermal heat flux boundary condition us problematic for these simulations because at 40 m depth the temperature profile is likely to contain a memory of past climates. Rather than deal with spinup in a reconstructed historical climate and all the uncertainties that this would create, it is better to use deep borehole temperature in regions where it is available.We used two simplified 1D column simulations with the same surface forcing as the full 2D transect, and with 1) constant temperature boundary condition and 2) geothermal heat flux boundary condition (which is set to 0) to show that the bottom boundary condition does not significantly affect the temperature in the active layer. We propose to include the following figure in the Appendix and add the following text to the body of the manuscript:

"*We confirmed with 1D simulations (see Appendix) that the treatment of the lower boundary has negligible effect on the near-surface region of interest up to year 40 and then only minor effects from year 40-50.*"

[Figure]

*Fig. 3: Thaw depth progression in simplified 1D column models using the same forcing as the full 2D transect in the main text for the (a) RCP 4.5 and (b) RCP 8.5 scenario, with each a constant temperature boundary condition (blue) and a heat flux boundary condition (orange) prescribed at the bottom of the domain.The simulations show that the choice of bottom boundary condition has little effect on the processes in the active layer throughout years 1-40 and only small effects (never more than 1 mesh cell difference between the two versions) in the years 40-50. Hence, we are confident that the bottom boundary condition is adequate and the domain depth is deep enough.*

24.    - l 184-187: Additional explanations should be added here: why this rate and total amount of injection, why this moment?

We will add more information on the injection rate. The moment of injection is, as described, meant to represent a day at which the domain is at fully frozen conditions before the onset of thaw. The rate of injection is determined by the target length of injection (1 daily cycle) and the total amount of tracer injected is arbitrary since the transport is conservative. We set it to  100 mol simply to represent 100 units of solute.

25. - l 204-205, «air temperature (T avg ), which is the only variable that changes over time in the respective scenarios. »: Why? Precipitation does also change along time in IPCC climate change scenario. This choice should be discussed.

Precipitation changes are difficult to predict and model predictions for high latitudes often diverge. Given that precipitation changes are not the focus of the study, we believe it is reasonable to adopt a stylized scenario that neglects precipitation changes to focus on temperature effects. We will add the following sentence to explain our motivation for this scenario:

"*This scenario is stylized and only aims to capture the effects of air temperature warming without considering complex, difficult-to-predict, and highly uncertain changes in precipitatio*n."

26. - Figure 4: Why the BTCs are multimodal? For instance, 2 modes for TOL carbon in Fig4.b and 2 modes for buried carbon in Fig4.c?

The release of tracer in the model is determined by the thawing rate of the active layer, which in turn is determined by air temperature changes. In our model, we use site-specific average air temperature and precipitation, which can vary from day to day. In a more idealized setting (e.g., annual temperature representation in the shape of a sine-curve or constant precipitation/infiltration rates) may cause a more homogenized image. In our case, daily variations are causing non-linear thaw rates and hence non-linear BTCs as tracer gets mobilized according to the thaw rates and groundwater flow through infiltration. We will extend the description of Figure 4 accordingly.

27. - l 271-272: Is this temporal partitioning between surface and subsurface transport in agreement with field observation? At least has the ponded water before mid-June been observed in the field? Whenever it is possible numerical results should be discussed at the light of field observations.

We will add a reference to field observations when possible in the main text (for example for the subsurface material as well as the ponded water observations argued in this comment). The catchment is indeed characterized by inundated conditions in the beginning of the thaw season (mid-June) due to snowmelt and shallow thaw depths.

28. - l 279-281: In Figure 4.d the peak of the 13 August seems higher than the one of the 14 June, although the opposite is stated in the text. This bimodality should be better explained.

The bimodality is caused by two main modes of transport at two different times of the active layer development. The first peak in the subsurface at 20m can be explained by the initial surface overland flow which causes the spike in Fig 4b as it transports solute mass with it, which then infiltrates into the subsurface and produces a subsurface BTC peak. Later, tracer that has not been transported on the surface initially, will be transported in the subsurface by slower groundwater flow. However, this happens later in the season, leading to the second peak. We agree that the "less pronounced" peak is

misleading and will delete it, while adding some more explanation about the bimodality to the description of the figure.

29. - Figure 8: Vertical peak on 30th of August, dual peaks in the 1st of September … I think these features are strange. May be due to convergence problems? The convergence study must be done for assessing it.

These features are not caused by convergence issues, rather by the fact that we are representing a realistic topography instead of an idealized flat transect. Changes in topography combined with the undulating permafrost table can lead to vertical and sub-vertical groundwater flow and cause the observed vertical increase in solute mass. Further, we mask values smaller than 0.05 mol for better visualization, which does not imply that there is no tracer in the remaining cells, but the quantity is below the defined threshold. This has been described briefly in the text (Line 315). We will extend this explanation and adjust the figure legend to reflect the lower solute mass limit.

30. - l 323-324: « A substantial fraction of the initially injected tracer mass (~ 40%) moves vertically (both upwards and downwards) within the same mesh column in which it was injected (see Fig. 9a and d and Fig. 10). » What phenomena are responsible for this vertical redistribution ? Diffusion, freeze/thaw cycles related effects? This should be explicated.

This vertical movement is a combination of drying/wetting and cryosuction. This is explained using Figure 10. The upwards movement (cryosuction) is so far only discussed in the discussion section, but we will also add clarifications on this to the results section.

31. - Figure 10.h: Numerical instability ? Should be corrected, or explained if it is not an artifact.

The blue cell downslope of the injection point in Fig. 10h is not an artifact nor numerical instability, but rather indicates that some tracer has moved downslope already. It only appears in this specific time instance plot due to the high temporal (daily) variability in output. In the original figure, we masked values smaller than 1 to improve visibility. Since this is not essential to the interpretation of the results and may confuse the reader, we will increase the threshold to 1.5, which will further improve clarity by removing the distracting downslope tracer movement. We will include the information about the masking in the figure caption.

32. - l 346 : « This observation highlights the importance of mesh resolution in lateral transport simulations. ». I fully agree. A convergence study must be done.

Convergence issues are not causing this observation. The mesh resolution in the present-day climate simulations is highest (1 cm per cell) in the contemporary active layer and decreases slowly with depth. However, energy- and mass conservation apply to these cells in the same way that it does in the contemporary active layer cells. The mesh resolution is therefore adequate.

33. - l 352, « vertical mobilization »: once again the involved mechanism must be explicated.

This will now be mentioned earlier on (see comment #30) and is also part of the discussion.

34. - l 407-411 : « We partly address this by representing a converging slope model setup, where the cell width in transverse direction varies depending on the distance in longitudinal direction. This way, the surface area of the catchment is preserved, and it is possible to accurately represent water and energy balances as well as infiltration and evaporation rates throughout the catchment. This approach has previously been applied by Gao and Coon (2022). » I do not think that using a one cell-thick discretization in the transversal direction may allow to simulate the effect of the watershed geometry, either convergent or divergent (using the terminology of Gao and Coon 2022). Nor in the present manuscript or in Gao and Coon 2022 are presented arguments for supporting the validity/usefulness of such a 'pseudo-3D' meshing methodology. A proper comparative study should be done for this, between results obtained with« pseudo-3D » meshes and with full 3D meshes. Of course with only one cell no lateral fluxes may be computed.The only interest I would see would be to weight the inward fluxes through the top cell face according to the area of the cell, but then why not simply apply a spatial weighting on the incoming fluxes prescribed by the boundary conditions? Including this in the meshing seems to me inappropriate and confusing. Anyway in this case the methodology used for computing the cells widths must be explicated.

See our reply to comment #9 and the proposed changes to the text. We aligned the mesh with the topographic gradient and thus the flow; by construction there is no tangential lateral flow, only in the vertical and along the topographic gradient. Flow-aligned meshes are a standard approach for reducing the computational complexity of hillslope simulations, as noted in our reply to comment #9.

35. - l 465, « Under the simulated environmental and soil physical conditions in this study, »: This precision hold true not only for this point, but for all the listed conclusions. The writing of the manuscript should better reflects the fact that this is a numerical study of transport in a specific site, with no possibility of automatic generalization for permafrost regions as a whole.

This very important point will be discussed more thoroughly in the discussion. While the results of this study may apply for foothill systems, they may unfold very differently for permafrost locations in e.g., plains with micro-topography.

TECHNICAL CORRECTIONS :

36. - l 25: Missing ) at the end of the line.

Noted

37. - l 29, l30, and elsewhere: I would recommend to systematically use 'organic carbon' instead of just 'carbon' for naming the C part of the organic matter stored in permafrost.

We prefer to keep using the word "carbon" as it is mentioned nearly 180 times in the current manuscript. We will clarify in the introduction that we refer to organic carbon throughout the manuscript.

38. - l 32: Missing s at 'question'.

Noted

39. - l71-84: Part of this paragraph should be in the Methods section (e.g.: choice of distinguishing four carbon pools and using labelled tracer for identifying them).

This final paragraph in the introduction section summarizes our approach to address the questions that arise from the literature summary in the rest of the introduction and hence is essential to the formulation of the hypothesis. A thorough explanation of the carbon pools and labels follows in the methods section.

40. - l 151, «bottom horizontal boundary »: According to Figure 3, the bottom boundary is not horizontal.

We will remove "horizontal"

41. - l 251-260: These information should be included in the Methods section, along with a figure for quantitative locations of the injection points and of the measurement points within the modelling domain.

We agree that a zoomed-in figure with the injection points would be beneficial for the interpretation of the results and will prepare such an illustration. We prefer to keep the introductory sentences in the results section as they currently are to preserve the flow of reading and introduce the way in which the results are presented.

42. - l 258, « explicit », « continuous in space » : Odd vocabulary. A BTC represents the temporal evolution (not exactly continuous, since there is a time discretization) of concentration at a given location in space, while a plume vizualisation represents the spatial distribution of concentration at a given moment.

We will clarify this sentence.

43. - Legend of table 2: The use of different concentration thresholds for TOL and buried carbon should be mentioned in the body of the text, in the Methods section.

We will add this to section 2.5 "Model forcing" under "Present-day weather conditions".

44. - l 273: « when all runoff is occurring in the subsurface »: Then it is not run off, but groundwater flow.

We prefer to keep our original phrasing as it encompasses both percolation and groundwater flow, as well as highlighting the surface-subsurface nature of runoff and flow interactions. Subsurface runoff is a defined hydrological term (e.g., Pilgrim et al. 1978)

45. - l 285: Do not refer specifically to a Figure in Supplementary material (Fig. S3) in the body of the text. Instead, refer to the supplementary material as a whole.

Supplementary information will be moved to the Appendix

46. - l 286 – 287 : « However, given the specific solute transport patterns in this model, » What is meant here ? Unclear.

We will reword this.

47. - l 299, « vertically transported upward »: Oddly said. Strictly speaking, the topographical effect mentioned here does not generate ascendant flow.

In this part of the domain, the local topography combined with the undulating permafrost table, which acts as an impermeable boundary for flow, causes the active layer to be locally saturated with liquid water at the time of tracer arrival, which in turn enables the observed vertical/sub-vertical tracer movement. The manuscript text will be revised as follows:

 "*A small fraction of buried carbon tracer (tracer mass < 0.005 mol) experiences surface transport (also visible in Fig. 8) because it gets transported upwards by groundwater upwelling. This is caused by a combination of the terrain unevenness and the undulating impermeable permafrost table, causing local downslope water accumulation and a fully saturated active layer.*"

48. - l 302: Do not refer specifically to a Figure in Supplementary material (Fig. S4) in the body of the text.

Supplementary information will be moved to the Appendix

49. - Caption of Fig. 6, « Note that the tracer mass is restricted to the uppermost subsurface cell in this snapshot and is difficult to visualize in this illustration » : True, then this figure has to be improved. May be that plotting the two variables

separately would be an option? Besides, Fig. 6 is mentioned only very briefly once in the text. Either it should be deleted or more extensively commented.

We will add an inset to the figure, zooming into the area where there is tracer and will also change the color scheme so that it becomes more visible (see a first draft of these changes in figure below).

[Figure]

*Fig 4.: New proposed visualization of Figure 6 in the main text*

50. - l 315, « some tracer is moved upwards »: see comment on l 299.

Noted

51. - l 317: 12 % does not obviously look negligible.

Will be changed to "small"

52. - Figure 7: I can't see anything regarding the solute mass. This Figure should be thoroughly reworked so that it becomes informative. Besides, what is the « ponded depth » ?

We will add the information about the ponded depth and also the thaw depth into the figure caption. Apologies for the oversight. However, we argue that the tracer mass is as visible as possible while still showing the entire transect of interest. For better visualization, we will decrease the opaqueness of the mesh and the thaw depth line.

53. - Figure 9: The annual cycles/peaks should be discussed. « The visual increase in mass above 100 mol in the injection columns in (a) and (d) is not a physical phenomenon but a result of aggregating and rounding errors during post-processing of the model output. » Then the post-processing should be improved.

Please see response to comment #54 below.

54. - l 350-355: If the post-processing method does not allow to conclude, then it should be improved.

The issue here is that a limited precision is used in output files causing very slight round-off errors. A limited precision is used due the large quantity of output data. To improve this, an increased precision in the output files could be implemented, but then the simulations would need to be re-run, which is not practical as the runtime is over 3 months wall clock time. As this is a minor issue which has no bearing on the results of the study we prefer to keep the data processing as is, but propose to improve the figures (Fig 9 and Fig 11) by truncating to the 100 molar upper limit where applicable. This correction should help clarify the presentation of results without distracting from cumbersome explanations of data post-processing. The explanation of data post-processing will be provided in the Appendix.

55. - Caption of Figure 11: « The visual increase in mass above 100 mol in the injection column in (b) is not a physical phenomenon but a result of aggregating and rounding errors during post-processing of the model output. » Once again the post-processing method should be improved.

 Please see response to comment #54 above.

56. - l 414-420: this should be part of the introduction, not the discussion.

This part places the work in the context of what has been done on permafrost carbon transport modeling previously and how it can be improved in the future. We therefore think it is important to mention this in the discussion to highlight  future development of this work and prefer to keep it as is.

57. - l 420-425: this should be part of the Methods section, not of the Discussion.

This section is discussing limitations of using a non-reactive representation of the tracer and we feel it is therefore important to elaborate on in the discussion section. We wish to refrain from discussing alternatives in the methods section.

58. - l 432-447: This should be part of a perspective section, not of the Discussion.

We feel it is useful and customary to include perspectives such as this in the discussion section. A dedicated section for this part would likely hinder the flow of the discussion. We therefore prefer to keep this part in the discussion.

59. - Section 4 Discussion: Given the parts that should not be included in the discussion (see the three comments above), the Discussion section is rather short (a bit more than one page), and lack of in depth analyses of the produced results. I recommend to strengthen this section, including for instance elements for linking the simulated behavior and the site specificity.

We will extend the discussion by a discussion about the site specificity, its role in a wider context in the Arctic, and add a discussion point on wetness conditions. We believe this will strengthen the discussion.

60. - l 452, « mechanical transport »: Sounds weird. Passive transport would be more relevant.

Will remove "mechanical"

**References**

Gao, B. and Coon, E.T., 2022. Evaluating simplifications of subsurface process representations for field-scale permafrost hydrology models. *The Cryosphere*, *16*(10), pp.4141-4162.

Hamm, A. and Frampton, A., 2021. Impact of lateral groundwater flow on hydrothermal conditions of the active layer in a high-Arctic hillslope setting. *The Cryosphere*, *15*(10), pp.4853-4871.

Jan, A. and Painter, S.L., 2020. Permafrost thermal conditions are sensitive to shifts in snow timing. *Environmental Research Letters*, *15*(8), p.084026.

Magnússon, R.Í., Hamm, A., Karsanaev, S.V., Limpens, J., Kleijn, D., Frampton, A., Maximov, T.C. and Heijmans, M.M., 2022. Extremely wet summer events enhance permafrost thaw for multiple years in Siberian tundra. *Nature Communications*, *13*(1), p.1556.

Pilgrim, D.H., Huff, D.D. and Steele, T.D., 1978. A field evaluation of subsurface and surface runoff: II. Runoff processes. *Journal of Hydrology*, *38*(3-4), pp.319-341.

---

## Author Comment (AC2)

**REVIEWER 2**

Comments by the reviewer are in black text

Our responses are in blue text

*Where applicable, proposed changes to the revised manuscript are in blue and italics*

**Major comments**

The manuscript describe a study using a numerical model to investigate the factors that influence mobilization and transport of carbon in soils in permafrost regions. Warming and permafrost thaw are likely to cause the transport of carbon from soils through aquatic systems, so a better understanding of physical mechanisms is crucial to the development of more accurate models and climate simulations.

We thank the reviewer for their thoughtful review. In response, we propose several modifications to address their comments and improve the manuscript. Our primary focus will be on providing a thorough justification of the meshing strategy, which was previously overlooked. Although it is not feasible to simulate multiple sites with this model, we will better clarify how our results can be generalized within a broader context. Additionally, we have added several clarifications throughout the text to enhance readability. We are confident that these changes will significantly improve the manuscript.

1. The study is designed as a highly idealized case for a small area in Svalbard. The grid setup is not standard. The time periods noted in the text are variable and confusing. The short time period of the abrupt that complicates interpretation. This application of a single model in this fashion makes extension of the study findings to broad regions of the Arctic questionable. The results would be more meaningful by applying the model for at least one or two other configurations based on observations of soil texture and organic carbon content, weather data, elevation gradients, which are available at select sites in Europe, Siberia, Canada, and northern Alaska. Doing so would allow the authors to have more confidence that what they are seeing represents a robust response and not an artifact of a unique model setup. The authors should also state the choice of application to the Endalen valley on Svalbard where observations are sparse. Is there a rich history of field study there?

We realize the description of the grid setup was not adequately explained and we will revise the text to better provide the justification. The key point is that the use of a flow-aligned, variable-width representative hillslope is a well-established modeling approach in catchment and hillslope hydrology. Please see our response to comment #9 by Reviewer 1. And while we agree that a study of multiple sites would be of interest in a future project, that's clearly out of scope for this project, which addresses our study site and takes advantage of available field observations. Importantly, we are confident that the understanding gained in this study is transferable to other similar locations and sites,

although of course not generally to any permafrost region. We will clarify and elaborate on the potential applicability to different locations in the revised manuscript. Further, we will address the issue of only modeling this specific site by better highlighting the available field observations, which include groundwater flow velocity observations with the help of tracer experiments. This is in line with comment #27 by Reviewer 1.

2. The 0.25 km2 catchment is relatively small. Why not use a traditional uniform horizontal grid and with vertical soil layers? How many grid cells would be needed at resolution of 1 m? Would the computational expense be prohibitive? How does the setup result in "a natural equilibrium without artificially imposed boundary conditions", and what does that phrase mean? Boundary conditions and zero-flux boundaries in a typical 3-D model setup could be the same as those in this study. In other words, I don't see the advantages of this current grid mesh, nor the implications for the interpretation of results. While the authors refer to computational expense, more justification should be presented.

A variable resolution mesh with high refinement around a region of interest is very standard practice in groundwater and surface/subsurface integrated hydrologic modeling. This is especially true for combined flow and transport models because finer meshes are typically required to resolve transport. In our case, we refined our mesh in the region of our numerical transport experiments and kept a coarse mesh away from that region to set the appropriate flow conditions for transport. We stand by this choice of mesh as it is best practice for these types of simulations. The advantage of the domain setup used in this study is it allows for a feasible computational runtime and simplifies analysis of results in a way which is suitable for the aims of the study. For reference, the climate change scenarios alone required over 3 months runtime to compute on a high end workstation. Although a full 3D domain mesh could in principle be considered, it is not warranted for the aims of this study.

We agree the phrasing  "a natural equilibrium…"  is unclear and we will revise the text. What we meant was that we use hydrometerological forcing as driving conditions for the model, as opposed to cumbersome and often simplified boundary conditions. This enables the thermal-hydrological dynamics of the model to be controlled by weather inputs in a way which more closely resembles nature by precipitation, air temperature, and factors influencing evaporation, etc, as opposed to boundary conditions representing hydraulic pressures or hydraulic gradients, and thermal inputs, etc. Importantly, this is only possible for integrated surface/subsurface models like ATS. We will clarify the phrasing referred to as "a natural equilibrium…" in the revised text.

3. The wide variation in time steps mentioned is awkward and makes interpretation of the influence on results difficult. There is a rate for tracer injection of per second and a time step of monthly for the warming experiments. If it were implicitly daily, could mineralization in the future simulations like the present day simulation? The time step of per second is confusing in light of a monthly step in the future simulations.

Internally, computations are performed with an adaptive time stepping scheme but output frequency is user-controlled. For the multi-decade simulations representing climate change scenarios it is of course not practically possible to output results at a high frequency such as every second. Therefore, output frequency is assigned to monthly snapshots in those cases. However, for present-day conditions and when focusing details such as seasonal variability, it is relevant to consider higher frequency outputs. Although this leads to a range of different time representations in outputs, it is necessary and reflects the complexity and novelty of our study. We will add a sentence explaining that ATS is using adaptive time stepping with a maximum time step of 1 day and a minimum timestep of 1e-10 seconds and that output frequency depends on the scope of the simulation (i.e. seasonal vs. climate change scenario).

4.  The term seasonal variability in the title is odd and unnecessary, as seasonal variability in a study like this is essential. Having the term 'sensitivity study' in the title would be more meaningful.

Seasonal variability stands in contrast to climate change and highlights the fact that we do both in this study. However, we do not conduct a full sensitivity study regarding model parameters or forcing. Hence, we prefer to keep the title as it is in the current version of the manuscript.

Minor comments

5.  Line 6: grammar, "Here, we analyze of solute transport…"

Noted

6.  Line 43: Unclear. What is "they" in the statement "With permafrost acting as a largely impermeable layer between the two, they are…"

Will change "they" to "the shallow and deep aquifers are mostly disconnected systems"

7.  Figure 3 caption: It appears from the graphic that width (y-direction) decreases with distance in the x-direction. As x increases from 0 to 1000, the width gets smaller. Please clarify.

We will add a sentence to the Figure caption to explain this nature of a convergent hillslope setting.

8.  Line 159 states that the model is run for an average yearly cycle. Then at line 172 there is reference to year-to-year differences. How can year-to-year differences in spinup be determined? Further clarify the spinup time period and protocol.

For each day of every year of the spinup, a certain state of the ground and the ground surface is calculated by the model. In the beginning, when the model is not in a cyclic steady state yet, there will be a difference between those days between one year and

the next. Once these differences decrease to a negligible amount, the system is determined to be in "cyclic steady state". We will include a plot in the Appendix showing differences in the surface temperature at a specific location for each day of the 10 years, showing that the difference between each e.g., June 1 becomes negligible. This is common practice in the modeling community (see comment #22 by Reviewer 1). We will add the following sentence to the manuscript:

 "*We confirmed that the spinup process places the system in a cyclic steady state (see Appendix) where conditions in each day of the year are nearly identical to those of the same day of the previous year*".

9. Line 171: typo "a the"

Noted

10. Line 173: What ten years? Is weather data not for an average yearly cycle (climatology)? Be specific about the ten year period.

We will change it to "ten years of average weather conditions"

11. Line 186: Clarify how numerically does a tracer become injected at a rate with a time step of a second. What is the model time step? If it is daily, how can something occur every second? Does the mass enter as a total each day? Intrinsic time step of the model is not clear.

ATS uses an adaptive time stepping scheme with user-assigned limits. This is advantageous as it adjusts the time step automatically using an iterative procedure to ensure numerical convergence depending on the dynamics of the system. For example, in spring and summer, with meltwater infiltration and variable groundwater flow, the time stepping scheme iteratively reduces the computational time step typically to a small value (less than a second), whereas during less dynamic periods, such as winter, the time step is typically increased to a maximum of one day, which helps reduce runtime. Of course the specified quantity is an injection *mass rate*; the actual mass injected during a time step depends on the length of that time step.

12. Line 235: typo, for for

Noted

13. Figure S3: component mass overlay is difficult to see in the graphic. Also add label for elevation for the y axis to make clearer what is being shown.

These will be reworked for clarity in the revised manuscript. Please also see our response to comment #49 by Reviewer 1.

14. Line 18 in supplement: units of %C ¿ 10 are unclear.

Noted

15. Line 337: typo, "have lead to"

Will be changed to "have led to"

16. Line 357: Do the magnitudes (+25 and +60 cm) of the abrupt active layer deepening have any grounding in reality? That is, do any studies report abrupt thaw that is some meaningful fraction of the amount modeled here? Some perspective on the magnitudes would aid in interpretation of the results.

We will add a reference to active layer variability in the Adventdalen valley. Strand et al., 2021 have shown that an abrupt increase of 30 cm is possible between a cold and a warm year. Hence, the modeled abrupt thaw is within the currently observed variability and additionally exaggerates it by a factor of 2. In a rapidly warming Arctic such as in the RCP8.5 scenario, heat extremes will be more common and may lead to abrupt thawing of the active layer, which is why we choose to consider scenarios which may seem extreme today, but may well be feasible within the time frame of the warming scenarios / coming century.

17. Line 368: "...is released from both the injection column…" grammar and unclear meaning.

We will remove "both".

18. Line 391: At the mention of ancient carbon, it would be helpful to remind the reader that this is frozen carbon. For readers that skim the abstract and discussion, the terms ancient and buried will have unclear meanings. Making clear that ancient means ancient currently frozen carbon would help.

We will add this to the text.

References:

Strand, S.M., Christiansen, H.H., Johansson, M., Åkerman, J. and Humlum, O., 2021. Active layer thickening and controls on interannual variability in the Nordic Arctic compared to the circum-Arctic. *Permafrost and Periglacial Processes*, *32*(1), pp.47-58.

---

## Author Response (AR1)

**REVIEWER 1**

Comments by the reviewer are in black text

Our responses are in blue text

*Where applicable, proposed changes to the revised manuscript are in blue and italics*

GENERAL COMMENTS:

The manuscript deals with an important and debated problem, the permafrost carbon feedback. More specifically, it uses a numerical simulation approach for assessing solute transport in a permafrost catchment, in order to study the impacts of climate change on DOC export from the soil of the considered watershed. The mechanistic modelling of dissolved species in permaftrost affected areas is at the forefront of the current research in cryohydrogeology, and the used tool is a state of the art one. Thus this work is both novel and of great interest for cryospheric sciences.

Meanwhile, critical weaknesses affect its reliability. First of all, the study focuses on a specific site, while the claimed aim of the work is to draw general conclusions about lateral transfers in permafrost regions. This is contradictory, and if the authors want to produce a general study then the problems of upscaling and transposing to different biogeoclimatic permafrost contexts must be carefully dealt with. I have also a number of technical concerns about the set up of the performed numerical simulations, for instance the poorly justified pseudo-3D approach, the absence of any convergence study, and some problems in the definition of boundary conditions. Finally the discussion of the obtained simulation results is rather weak and should be strengthened. All these points are detailed in the specific comments below.

Overall I think that the manuscript cannot be published in its present form. A thorough effort is needed for making it more solid and meaningful, including possibly new computations and results depending on the answers to some of my concerns (e.g.: numerical convergence, relevance of top and bottom boundary conditions). So I recommend that a major revision of this work should be undertaken prior to reconsider whether or not it should be published in The Cryosphere.

We thank the reviewer for their thoughtful review. To improve the current manuscript, we made several modifications to address their comments. The main focus of the improvements was on the justification of the meshing strategy, which has been neglected in the original manuscript. Further, we added clarification on the applicability of our results on a larger pan-Arctic scale, improved figure clarity, and added additional descriptions to figures where needed. We are confident that this will significantly improve the manuscript.

SPECIFIC COMMENTS:

1. - Title and l 8-9: The title refers to 'a permafrost catchment', while abstract draws conclusions for 'permafrost regions'. There is no *a priori* reason that conclusions made from the study of a peculiar catchment should be relevant for permafrost regions as a whole.

We agree that the conclusion that our results apply on a larger scale may be misleading as it is currently written. While it is not possible to generalize our results for the entire northern hemisphere permafrost region, the results from this study can be extended to other permafrost hillslope catchments with similar climatic, hydrological and geological conditions. This site considered in our study is representative of gravely footslopes with a gentle inclination (5-15°, Hamm and Frampton 2021). We adjusted the phrasing in the abstract and improved the discussion on how representative these results may be in a broader context of similar permafrost hillslope regions.

See Abstract

L5-10: "*Here, we focus on a sub-catchment in Endalen, Svalbard, as a representative example of a high Arctic hillslope underlain by continuous permafrost. We analyze solute transport in the form of a non-reactive tracer representing dissolved organic carbon (DOC) using a physics- based numerical model with the objective to study governing cryotic and hydrodynamic transport mechanisms relevant for warming permafrost regions*"

and Discussion

L443-451: "*While the Endalen valley represents a single case within the context of high Arctic hillslope systems underlain by continuous permafrost, similar systems are widespread across the Arctic, making this site a valuable example for studying permafrost carbon transport. The slope of approximately 12° in our study area is representative of gently sloped terrain in permafrost landscapes, which accounts for about 20% of the total permafrost region, commonly found in areas such as Greenland and parts of Canada (Hamm and Frampton, 2021). However, our findings may not be directly transferable to other permafrost settings, such as flat terrain, where differences in groundwater flow velocities, saturation levels, and small-scale processes like cryosuction could significantly alter transport dynamics, especially in landscapes influenced by microtopography, such as polygonal tundra. Additional studies across a variety of terrains, supported by reliable field observations, are necessary to gain a more comprehensive understanding of lateral carbon transport in permafrost environments.*"

2. - l 17: Why considering mineralization of organic matter to $CO_2$ only, and not also to $CH_4$? May be due to the peculiarity of the study site. This should be explained.

We included this information in the methods section, rather than in the abstract. To better highlight our decision, we added more explanation as to why we do not consider mineralization into CH4. The main reason is that the study site is a hillslope and compared to wetlands and peatlands is unlikely to contribute significantly to overall permafrost CH4 release. We also added a reference to this statement in the methods section.

L231-233: "*In this study, we do not consider mineralization into methane (CH4 ), as the study site is located on a hillslope with rapid groundwater flow. Therefore, the production of methane is expected to be insignificant compared to those generated by wetlands (Denman et al., 2007).*"

3. - l 27-41: Other questions than PCF maybe related to lateral transfer of dissolved chemical species from the active layer under climate change (e.g.: impacts on river / ocean ecosystems).

We added a sentence on potential impacts on river ecosystems and the ocean, such as rusting rivers, mercury release, and ocean acidification.

L44-50: "*Thawing of permafrost not only releases organic carbon but also has implications for the mobilization of other chemical species, including contaminants such as mercury or trace metals. As permafrost thaws, the previously sequestered contaminants in frozen soil can be released into the environment, potentially increasing pollutant levels in aquatic systems, threatening human health (O'Donnell et al., 2012; Smith et al., 2024). Additionally, the increased input of DOC from thawed permafrost can influence surface water chemistry. The lateral transport of DOC into oceans may interact with ocean acidification processes, as increased CO2 absorption by oceans decreases pH (Semiletov et al., 2016). This interaction could affect marine biogeochemical processes and ecosystem health.*"

4. - l 71-72: The assumption that DOC may be considered as a non reactive tracer should be discussed here, with relevant references.

A passive tracer is used to numerically represent advective solute transport in the simulations; the conceptualisation used is that a given quantity of DOC is transported by the advective flow field but can undergo mineralisation which is dependent on soil thermal and saturation dynamics. Although this does not account for the full complexity of carbon reactions it embodies an important advancement in coupling soil and groundwater dynamics with potential DOC transport and is a novel contribution of our study. The implications are mentioned in the introduction but further discussed in the discussion section. We refrain from adding it to the introduction section.

5. - l 80-82: Most likely this behavior will be site dependent! For instance not the same for a tundra hill slope with low evapotranspiration or a boreal forest hill slope with high evapotranspiration, depending on the slope itself, etc.

The emphasis on the site-specific hypothesis was missing. We changed the text accordingly.

L89-92: "*We hypothesize that for a high-Arctic hillslope as represented by our study site in Endalen, Svalbard (i) buried carbon within the active layer will be transported faster than TOL carbon due to higher saturated soil conditions at the bottom of the active layer; and (ii) ancient permafrost carbon will be exposed to similar highly saturated late season conditions leading to rapid transport upon thaw.*"

6. - l 111: Is Qs sink/source term accounting for ionic exclusion? To say, does the used model take into account the specificity of solute transport with freeze/thaw?

Yes, exclusion of solute from the ice is included, although it is enforced by the nonlinear solver and the formulation, and not in the Qs term. The solute in ATS is represented as a moles of solute per moles of liquid water. This way, when the water in the soil freezes, the solute remains in the unfrozen water fraction of the soil and solute concentration increases to honor mass conservation, which is enforced by the nonlinear solver. Upon melting of the pore ice, the opposite occurs. We added the following text in the methods section to clarify this:

L126-128: "*Note that solute is excluded from the ice phase by the formulation which represents solute as moles of solute per moles of liquid water and by the nonlinear solver which enforces solute mass conservation. Thus freezing of pore water results in an increase of solute concentration and the opposite is true for melting of pore ice.*"

7. - l 113: Why neglecting dispersivity? The errors associated with this simplifying assumption should be discussed.

This study deliberately focuses on lateral transport by advection. Although it would be interesting to also consider dispersion, it is highly substrate-dependent and can not easily be quantified with field observations. Hence, any assumptions of dispersion coefficients would add to parameterisation uncertainty and obfuscate the advection-specific results discovered here. For example, the observation of vertical tracer movement by wetting/drying and cryosuction could have been misinterpreted as dispersivity had it been included. We added an explanation to the text better explaining what the consequences of this decision are.

L128-138: "*In the simulations performed in this study, dispersion is intentionally omitted by setting $D_l = 0$, focusing exclusively on solutes advected by water flow. This approach allows for the identification and analysis of the transient and seasonally variable flow field exhibited in the active layer, as well as the effects of freeze-thaw, cryosuction, wetting-drying in partially saturated soil, and unfrozen water seepage in permafrost. Although it would be valuable to consider dispersion, it is highly substrate-dependent and challenging to quantify with field observations. Consequently, assuming dispersion coefficients would increase parameterization uncertainty and potentially obfuscate the advection-specific transport patterns caused by small-scale water movement such as percolation of unfrozen water at sub-zero temperatures or transport through capillary*

*forces (such as cryosuction).Thus, the omission of dispersion ensures a clearer focus on lateral transport driven by advection, simplifying interpretation of the model's results.*"

And Discussion:

L415-418: "*Note that explicit dispersivity was omitted in the model (see Section 2.2) and is therefore not responsible for this observation. We argue that, had dispersivity been included, this result might have been obscured, potentially leading to a misinterpretation of cryosuction's role in solute transport as substrate-dependent dispersion.*"

8. - l 118-119, « path of highest flow accumulation »: What about lateral transfers from the slopes to the thalweg? For studying the drainage of active layer waters and the associated solute fluxes the choice of focusing on the thalweg line does not seems obvious to me. This should be discussed.

We now discuss these choices more thoroughly in an expanded description of the meshing process (Appendix A and D, please see next comment).

9. - l 121, « pseudo-3D approach »: I cannot understand the usefulness of varying y-axis width of cells along the x direction. According to Fig.2, there is only one mesh along y-axis at every x position. So this seems to me a 2D mesh (as said in l 169). The Figure 3 and the associated explanations are hard to follow and should be thoroughly improved (e.g.: a large blue area appear on Fig.2, but only green and red area are mentioned in the text.) Whatever the pseudo-3D approach really is, the statement made by the authors that it « allows us to account for thermal and hydrological balances across the entire catchment area without the need for a complex and computationally intensive full 3D mesh » should be justified. How does the y-axis width varies along the thalweg? Has any comparative study between results of a full 3D approach and results of this pseudo-3D approach been done? It should have been, prior to use the pseudo 3D approach, or at least if it is not practicable due to computation time the approximation should be discussed, as well as the associated errors.

We acknowledge that the justifications as to why we chose the particular meshing approach were not adequately explained. We revised that section to better provide the justification. The key point here is that the use of a variable-width representative hillslope is a well-established modeling approach in catchment and hillslope hydrology. In this approach, the sides of the hillslope are determined from topography by gradient descent, thus aligning the sides with the surface flow and allowing for no-flow boundary conditions on the sides. The fact that the width is now variable results directly from the 3D topography and preserves flow convergence/divergence, which is well-established to be important. We added some justification to the meshing approach to the main text in the revised manuscript:

L140-154: "*In this study, we employed a pseudo-3D variable-width mesh to capture the hydrological and thermal processes of the Endalen sub-catchment. The mesh extends 1040 m in the direction of highest flow accumulation and is of variable width in the*

*direction perpendicular to flow to enforce the correct contributing area to each mesh cell on the surface. Such variable-width hillslope meshes preserve flow convergence, ensuring hydrological processes are well represented without the expense of a fully three-dimensional model (e.g., Fan and Bras, 1998; Troch et al., 2003; Hazenberg et al., 2015). In the direction of flow, the mesh is divided into three key sections (see Fig. 3):*

*1. The upper slope (0-1000 m) – primarily serves as a water source (green-shaded area).*

*2. The main area near the valley bottom (1000-1020 m) – where transport experiments and tracer observations are conducted (red-shaded area).*

*3. The buffer zone (1020-1040 m) – designed to prevent boundary effects from influencing the simulation results (grey-shaded area).*

*The mesh (blue body in Fig. 3) is coarser in the upper slope and refined in the lower areas to better capture fine-scale processes close to the valley bottom. This strategy, together with variable-width elements that preserve flow convergence, provides an accurate representation of catchment-scale processes without the computational burden of a full 3D model (Appendix A and D).*"

We also expanded on the explanation of the mesh by moving mesh-specific information to the Appendix (Appendix A) to be able to elaborate more thoroughly without increasing the already lengthy main text significantly. We added details to the representation of the catchment hydrology, the boundary conditions, and the importance of accounting for all the water in the form of precipitation within the catchment rather than only accounting for a short transect and the precipitation falling onto its surface. We also added a sentence on the complexity of the processes represented in the model and how this does not allow for a feasible representation in 3D due to computational costs (see Appendix A).

10. - l 126-127, « main area of interest »: Why focusing on such a tiny area of 20 m length, while simulations are done on a 1040 m large domain? This important choice should justified and discussed.

As described in the text, the upper parts of the transect are mainly in place to provide the adequate amount of water to the lower parts of the slope, where organic matter can be expected to exist in the permafrost. We added to this explanation by highlighting that the rocky nature of the upslope part leads to very low carbon abundance and that carbon transport in this section is not of interest. Only in the lower parts of the slope, where the inclination is not as steep anymore, carbon may be present.

See Appendix A

11. - l 129-130, « By directing precipitation from the upper slope to the lower areas, we ensure realistic hydrological conditions with flow accumulation towards the valley bottom. »: What is meant here? The physical equations solved by the numerical model do ensure that gravity exerts a vertical descendant driving force

on water, so that water flows from top to bottom when gravity dominates. This sentence seems pointless, I recommend to delete it.

This is now part of the reworked mesh-section described in comment #9 (Appendix A). The sentence was rephrased for clarity.

12. - l 130-131,« This division of the mesh allows for accurate modeling of the thermal-hydrological processes in the catchment. » : Such a statement should be justified.

This is now part of the reworked mesh-section described in comment #9 (Appendix A).

13. - l 132, « Each column in this mesh area [...] varies in width in the y-direction. »: Why and how? See above the point on the pseudo-3D mesh approach.

This is now part of the reworked mesh-section described in comment #9 (Appendix A). Wel also added information on how the mesh has been created and that its shape has been chosen to represent the approximate catchment area derived from the digital elevation model.

14. - l 132 – 141: Here mesh cells dimensions are described, but not justified. In a modelling study relying on PDE spatio-temporal discretizations (e.g.: finite differences method, finite volume methods, finite elements methods, etc), it is mandatory to assess the truncation errors by dedicated convergence studies, designed for the simulation case under concern. The results of such a convergence study should be given here, by means of a upper bound of the truncation errors for the outputs of interest. Only in this way one can be sure that the variations discussed in the numerical results are not simply due to truncation errors. Please include here the results of such convergence study for the case under consideration.

The ATS model has been used for several years in multiple studies of cold regions hydrology and has undergone thorough testing (Painter et al., 2016, Coon et al., 2019, Jan et al., 2020, Painter et al., 2023). The relative error tolerance in the present simulations is set to 1e-6 for all variables, which ensures conservation of mass and energy with a very fine tolerance.

Our aim in this study is to improve the understanding of carbon export by groundwater flow in the active layer of a permafrost hillslope. For this purpose, we base our mesh discretization approach on previous studies (e.g., Jan et al., 2020, Hamm and Frampton, 2021). In particular, we adopt a very fine vertical discretization of the active layer which resolves freeze-thaw dynamics and groundwater flow in the main area of interest where the analysis of tracer transport is conducted. A new feature of our study is the inclusion of the entire watershed upslope of the main area of interest, where a coarser mesh discretization is adopted, which is necessary in order for the model to be computationally feasible. This in essence corresponds to modeling an entire catchment, where downscaling is applied to a small area of focus which is highly resolved (here, the

downslope area of interest for tracer transport) compared to the entire catchment (here, the upslope area). The combination allows for preserving thermal-hydrological dynamics of the catchment stemming from the hydrometeorological forcing with detailed representation of groundwater flow and solute transport in the specific smaller downslope area, and which is the focus of the study.

To ensure the downscaling approach is suitable, we have investigated the effect of increasing the resolution of the more coarsely resolved upslope area. Two additional cases are considered, where the length of the mesh cells in the upslope area are reduced from the original 23 m to 11 m and 5.8 m, i.e. effectively doubling the resolution in each case. Details of the moisture dynamics change in the upslope region which upon careful analysis is deemed to be physically consistent with the increase of the spatial resolution. A finer resolution leads to more responsive freeze-thaw and moisture release dynamics, causing slightly increased thaw rates in the main area of interest, but importantly, only cause minor changes to the solute tracer breakthrough curves (Figure D4 of Appendix D, also below) and do not alter the main results of the study. As these observed seasonal differences are small, we maintain the simulations representing long term climate change scenarios with the original coarser mesh. For site-specific studies aiming to reproduce field measurements, accurate replication of field-measured thaw rates would be important. However,since the goal of this study was to gain a better qualitative understanding of carbon transport in the active layer, the absolute timing of the breakthrough curves is of secondary importance. Therefore we decided to proceed with the coarsest mesh resolution for the climate change simulations, which are extremely computationally demanding. Given the seasonal differences are small, we are confident that the long term effects are reasonably represented with the configuration. The details of this mesh refinement study are presented in Appendix D of the revised manuscript.

[Figure]

*Fig. D4 of Appendix D: Breakthrough curves like shown in Fig. 5 without the surface transport contribution but including breakthrough curves for all tested mesh resolutions for the upper slope area. The different panels show breakthrough curves in the surface (a and c) and the subsurface (b and d) for each, the TOL carbon (top panel) and buried carbon (lower panel) for each of the three cases and at both observation point distances.*

15. - l 134-137: I do not understand why a buffer zone is needed. This should be explained.

This is now be part of the reworked mesh-section described in comment #9 (Appendix A). The buffer zone is used to avoid effects of the outlet boundary condition; please see response to comment #17 below.

16. - l 142-146: All of these sentences look like unjustified statements. Moreover what is exactly stated is not completely clear. For instance what means « preserving the subsurface volume representation of the catchment » and « a natural equilibrium without artificially imposed boundary conditions »? This paragraph should be either deleted or rewritten in a clearer and justified way.

This is now be part of the reworked mesh-section described in comment #9 (Appendix A). We clarified the text and deleted confusing statements.

17. - l 147-148: « The vertical sides of the model are assigned zero-flux boundaries for water » this is questionable, especially for the outlet vertical boundary. Are there any field observation that can be used to justify this choice? This should be discussed.

It is important to note that the outlet boundary is not closed to flow. As noted in the original manuscript, water expressing on the surface is removed. Thus, by closing the subsurface on the outlet, water flow is forced to the surface where it is removed. This can create an artifact in the immediate vicinity of the outlet, which is why we use a buffer region and do not analyze results in that region. By previous experience (Jan and Painter, 2020; Hamm and Frampton, 2021) and physical arguments we know this boundary condition produces physically correct behavior away from the immediate vicinity of the boundary. We elaborated more extensively on the decision in the Appendix.

L546-550: "*3. Buffer Zone (1020-1040 meters): The horizontal resolution is increased to 1 m in the x-direction in this area, which acts as a buffer zone where water can accumulate in the subsurface toward the end of the transect. The boundary condition prescribed at the surface of the last column allows water to leave the system at the surface of the last mesh column. To avoid misinterpreting these artifacts as hillslope processes, this zone is not central to the analysis of the results and is implemented based on methods established by Jan et al. (2020) and Hamm and Frampton (2021).*"

18. - l 151, « in line with borehole observation in Svalbard »: This is important, since it is likely the reason why a 40 m thickness as been chosen for the modeling domain. Please add a figure with the mentioned soil temperature profile

evolution, as well as a discussion for explaining in which way these data where used for choosing not only the bottom thermal boundary conditions but also the domain thickness.

We added a plot with trumpet curves showing that the seasonal surface temperature signal does not penetrate the ground deep enough to affect the bottom boundary condition to the appendix. The maximum annual temperature variations at depth -15 m are 0.1°C, indicating this is a reasonable approximation of the depth of zero annual amplitude (DZAA). Also, see our reply to comment #23 on why we are confident that a 40 m deep domain is sufficient.

See Appendix D and Figure D1

[Figure]

*Figure D1. "Trumpet curves" showing the seasonal development of the soil temperature profile with depth for four given times. The profiles indicate that the zero annual amplitude depth is located at ˜15 m depth.*

19. - l 159-165: I have serious concerns about the chosen methodology for building 'present day-like' precipitation. Why not using a day-of-the-year average like for other forcings? This is not justified. Stating that « the resulting rainfall distribution resembles the variability of natural rainfall throughout the year» is not enough for making the arbitrary, artificial precipitation forcing data set relevant. The use of arbitrary precipitation data may impair the interpretability of the simulation results, so either the 'resemblence' between the artificial data set and the observation

should be quantitatively demonstrated, either the observation data set itself should be used as forcing data.

This approach was previously presented in Magnússon et al., 2022. For precipitation, a day-of-year-average is not representative of the physical system because it will always create very small values for each day in a predominantly dry place like the Adventdalen area, as opposed to many days with no precipitation and a few days with significant precipitation. These small precipitation values would lead to an unrealistically high fraction being evaporated due to the surface energy balance and the overall windy conditions in the valley. Additionally, there would be no days with absolute 0 precipitation, which is not realistic. As described in Magnússon et al. 2022, the daily rainfall amounts are not arbitrary, but represent the frequency-intensity distribution of the natural rainfall variability. Details can be found in the Supplementary Methods VI in the cited paper by Magnússon et al. 2022 and in the included repository with an annotated script that was used to create this precipitation time-series.

20. - l 165-166, « Soil physical properties are defined to resemble highly conductive material. »: This should be justified. Why not moderately conductive material?

We added a justification to this statement. We assume a highly conductive material in the model because observations in the field site suggest that most of the subsurface material is very gravely or mossy. There are no field observations that describe the subsurface composition in detail.

L159-161: "S*oil physical properties are based on qualitative field observations and are defined to resemble highly conductive material (Table 1). A more detailed description of the mesh setup, boundary conditions, and soil physical properties can be found in the Appendix (Appendix A).*"

See also comment #27 and Appendix B

21. - l 169-170, 'to establish a water table at target depth »: I do not understand. What is the 'target depth'? Are there observational evidences of a water table 'at target depth'?

The spinup process needs to establish an ice saturated subsurface below the active layer. A multistep procedure for doing this with ATS was developed more than a decade ago and is the standard spinup procedure for ATS simulations of continuous permafrost. The first step in that procedure is to establish a hydrostatic water column with the water table close to the surface. We then freeze that water in place with an open boundary condition on top to allow water to be pushed out by the liquid-to-ice volume expansion. That establishes an ice table as close to the surface as possible. Subsequent spinup steps with seasonally varying top boundary conditions then remove any excess ice/water and place the system in the desired ice-saturated state in cyclic equilibrium with the atmosphere. The final step enables the active layer to develop together with a seasonally perched water table at a depth consistent with present day conditions. We replaced the confusing sentence with the following text:

L164-166: *"First, a single column model extending to the full depth of the final 2D mesh is used to establish an ice-saturated subsurface with an ice table near the surface. This was accomplished by freezing an initially hydrostatic and isothermal water column from below keeping an open top boundary to allow the volume expansion of the phase change to push out excess water."*

22. - l 173-174: Why 10 years? Please provide the criterium used for this choice.

10 annual cycles for the second spinup step is a commonly used length for this initialization step in a 2D problem (e.g. Gao and Coon 2022, Jafarov et al. 2022) to reach an annual steady/cyclic state. In the plot below, it can be seen how the mean annual difference in surface temperature at the surface of the first mesh column in the area of interest between 2 adjacent years during the spinup decreases and reaches a minimal temperature variation after 5 years. We included this plot in the Appendix and added the following sentence.

L171-172: *"We confirmed that the temperature is in a cyclic steady state at the end of the 10 year spinup period (Fig. D2)."*

[Figure]

*Fig D2: Plot of difference in surface temperature at the surface of the first mesh column in the area of interest. Mean differences are denoted as mean annual differences between the current year and the prior year. Despite some variability in the temperature difference, the mean difference in the last year is below 0.01°C with an absolute maximum daily difference of 0.2°C. The system is therefore considered to be in a cyclic steady state.*

23. - l 176-178: The targets of this study are simulations under climate change, while the use of a thermal boundary conditions of constant temperature equal to present day temperature at the bottom of the domain (l 150-151) is not compatible with the simulation of climate change scenarios. Or at least, it implies

the assumption that the temperature at 40 m depth is not impacted by surface temperature variations at the considered time scale of 50 years. This is a major concern for the validity of the produced simulation results. In order to demonstrate that this strong approximation does not impair the discussed results, the time variations of temperature in depth, close to the bottom boundary, must be shown. If they are not negligible, then the simulations should be re-ran with an appropriate bottom boundary condition (e.g.: geothermal heat flux).

Geothermal heat flux boundary condition is problematic for these simulations because at 40 m depth the temperature profile is likely to contain a memory of past climates. Rather than deal with spinup in a reconstructed historical climate and all the uncertainties that this would create, it is better to use deep borehole temperature in regions where it is available. We used two simplified 1D column simulations with the same surface forcing as the full 2D transect, and with 1) constant temperature boundary condition and 2) geothermal heat flux boundary condition (which is set to 0) to show that the bottom boundary condition does not significantly affect the temperature regime in the active layer. We included the following figure in the Appendix (Fig. D3) and added the following to the body of the manuscript:

L213-215: "*As we do not change the bottom boundary condition in these simulations, we confirmed with 1D simulations (Fig. D3) that the treatment of the lower boundary has negligible effect on the near-surface region of interest up to year 40 and then only minor effects from year 40-50.*"

[Figure]

Fig. D3: Thaw depth progression in simplified 1D column models using the same forcing as the full 2D transect in the main text for the (a) RCP 4.5 and (b) RCP 8.5 scenario, with each a constant temperature boundary condition (blue) and a heat flux boundary condition (orange) prescribed at the bottom of the

*domain. The simulations show that the choice of bottom boundary condition has little effect on the processes in the active layer throughout years 1-40 and only small effects (never more than 1 mesh cell difference between the two versions) in the years 40-50. Hence, we are confident that the bottom boundary condition is adequate and the domain depth is deep enough.*

24. - l 184-187: Additional explanations should be added here: why this rate and total amount of injection, why this moment?

We added more information on the injection rate. The moment of injection is, as described, meant to represent a day at which the domain is at fully frozen conditions before the onset of thaw. The rate of injection is determined by the target length of injection (1 daily cycle) and the total amount of tracer injected is arbitrary since the transport is conservative. We set it to 100 mol simply to represent 100 units of solute.

L183-186: "*Both tracers are injected during fully frozen conditions from May 16 to 17 (24 hours). This ensures that they stay frozen and in place before the onset of thaw. The injection rate is a constant rate of 0.0012 mol s-1 . This yields a total of 100 mol in the model, which is an arbitrary number to represent 100 units of solute. Since transport is conservative and non-reactive, the total mass of solute in the model will not change.*"

25. - l 204-205, «air temperature (T avg ), which is the only variable that changes over time in the respective scenarios. »: Why? Precipitation does also change along time in IPCC climate change scenario. This choice should be discussed.

Precipitation changes are difficult to predict and model predictions for high latitudes often diverge. Given that precipitation changes are not the focus of the study, we believe it is reasonable to adopt a stylized scenario that neglects precipitation changes to focus on temperature effects. We added the following sentence to explain our motivation for this scenario:

L211-213: "*This scenario is stylized and only aims to capture the effects of air temperature warming without considering complex, difficult-to-predict, and highly uncertain changes in precipitation.*"

26. - Figure 4: Why the BTCs are multimodal? For instance, 2 modes for TOL carbon in Fig4.b and 2 modes for buried carbon in Fig4.c?

The release of tracer in the model is determined by the thawing rate of the active layer, which in turn is determined by air temperature changes. In our model, we use site-specific average air temperature and precipitation, which can vary from day to day. In a more idealized setting (e.g., annual temperature representation in the shape of a sine-curve or constant precipitation/infiltration rates) may cause a more homogenized image. In our case, daily variations are causing non-linear thaw rates and can hence cause non-linear BTCs as tracer gets mobilized according to the thaw rates and groundwater flow through infiltration. In the updated results, the non-linear behavior is less pronounced. Therefore, we did not make changes to the text.

27. - l 271-272: Is this temporal partitioning between surface and subsurface transport in agreement with field observation? At least has the ponded water before mid-June has been observed in the field? Whenever it is possible numerical results should be discussed at the light of field observations.

We added a section to the appendix describing the qualitative field observations made in September 2022. We were not able to make any quantitative assessments of the landscape, hence we initially did not include those observations. However, it is helpful for the interpretation of the model results. We refer to the added section in the main text whenever suitable. As for the inundation of the surface, the catchment is indeed characterized by inundated conditions in the beginning of the thaw season (mid-June) due to snowmelt and shallow thaw depths.

L286-288: "*This aligns with field observations, where snowmelt in early June led to fully inundated conditions across portions of the sub-catchment surface, followed by drying as thaw depth increased and melt water can infiltrate into the subsurface (see Appendix B).*"

And Appendix B

28. - l 279-281: In Figure 4.d the peak of the 13 August seems higher than the one of the 14 June, although the opposite is stated in the text. This bimodality should be better explained.

The bimodality is caused by two main modes of transport at two different times of the active layer development. The first peak in the subsurface at 20m can be explained by the initial surface overland flow which causes the spike in Fig (former)4 (now 5)b as it transports solute mass with it, which then infiltrates into the subsurface and produces a subsurface BTC peak. Later, tracer that has not been transported on the surface initially, will be transported in the subsurface by slower groundwater flow. However, this happens later in the season, leading to the second peak. We agree that the "less pronounced" peak is misleading and deleted the sentence, while adding some more explanation about the bimodality to the description of the figure.

L294-300: "*An initial subsurface pulse peak at 20 m can be seen just after the shift from surface- to subsurface dominated transport on 7 June and can be attributed to the early active layer development, where the shallow active layer is saturated. Most of the transport during this time happens at the surface, but as soon as surface water starts infiltrating in mid-June, it creates the first subsurface BTC peak. A second peak at the 20 m observation point on 26 July indicates the arrival of the remaining mass mobilized in the warm season. The abrupt end of the BTC can be attributed to the increasing liquid saturation in the end of July that enables rapid transport at fully saturated conditions. All in all, the BTCs suggest long travel times in the subsurface upon initial release.*"

29. - Figure 8: Vertical peak on 30th of August, dual peaks in the 1st of September … I think these features are strange. May be due to convergence problems? The convergence study must be done for assessing it.

These features are not caused by convergence issues, rather by the fact that we are representing a realistic topography instead of an idealized transect. Changes in topography combined with the undulating permafrost table can lead to vertical and sub-vertical groundwater flow and cause the observed vertical increase in solute mass. Further, we mask values smaller than 0.06 mol for better visualization, which does not imply that there is no tracer in the remaining cells, but the quantity is below the defined threshold. This has been described briefly in the text (Line 315). We extended this explanation and adjusted the figure legend to reflect the lower solute mass limit.

"*Figure 9*. 2D representation of the buried carbon plume dispersion between the point of tracer injection (x = 1000 m) and the last observation point (x = 1020 m) for selected dates during summer after tracer injection. Elevation is given in relative elevation to the conceptual valley bottom (surface of the valley bottom = 0 m). Tracer mass is given in mol and values < 0.06 mol are masked for better visualization. The blue solid line marks the water table above the surface (ponded depth), the black dashed line indicates the thaw depth at each date.."

And see our response to comment #47

L335-338: "*Here, the plume movement is largely uniform in the subsurface. Due to the unevenness of the terrain and highly saturated conditions, some tracer mass is transported upwards towards the surface (e.g., at 1005 m on 23, 25, and 27 August in Fig. 9), but the amount of buried carbon tracer being transported on surface is small (max. 8% on 26 July, Fig. 5b and d, green shaded area).*"

30. - l 323-324: « A substantial fraction of the initially injected tracer mass (~ 40%) moves vertically (both upwards and downwards) within the same mesh column in which it was injected (see Fig. 9a and d and Fig. 10). » What phenomena are responsible for this vertical redistribution ? Diffusion, freeze/thaw cycles related effects? This should be explicated.

The vertical movement is a combination of drying/wetting and cryosuction. This is explained using Figure (former) 10 (now 11). The upwards movement (cryosuction) was previously only discussed in the discussion section, and we now also added clarifications on this to the results section.

L347-350: "*This observation can be attributed to percolation (during thaw) and cryosuction (during freeze-up). When the active layer is developing in the late thawing season, some tracer moves vertically downwards into partially frozen layers while the opposite occurs during the freeze-up when the freezing front from above draws water towards it due to capillary forces referred to as cryosuction.*"

31. - Figure 10.h: Numerical instability ? Should be corrected, or explained if it is not an artifact.

The blue cell downslope of the injection point in Fig. 10h is not an artifact nor numerical instability, but rather indicates that some tracer has moved downslope already. It only

appears in this specific time instance plot due to the high temporal (daily) variability in weather conditions and output frequency. In the original figure, we masked values smaller than 1 to improve visibility. Since this is not essential to the interpretation of the results and may confuse the reader, we increased the threshold to 1.5, which will further improve clarity by removing the distracting downslope tracer movement. We included the information about the masking in the figure caption.

"*Figure 11. Representation of the vertical redistribution of tracer from the initial injection cell (red cross) throughout the depth profile of the injection column. (a-c) indicate the distribution throughout a full freeze-thaw cycle, (d-f) represent vertical redistribution from the third to the fourth freeze-up including the mid-winter vertical movement of the second freeze-up, and (g-i) represent vertical distribution by the end of the first modeled decade. Mass is depicted in molar mass and only values larger than 1.5 mol are shown, the remaining mass is masked for better visualization.*"

32. - l 346 : « This observation highlights the importance of mesh resolution in lateral transport simulations. ». I fully agree. A convergence study must be done.

Convergence issues are not causing this observation. The mesh resolution in the present-day climate simulations is highest (1 cm per cell) in the contemporary active layer and decreases slowly with depth. However, energy- and mass conservation apply to these cells in the same way that it does in the contemporary active layer cells. The mesh resolution is therefore adequate.

However, after careful analysis, we found that the abrupt release is more likely caused by the beginning of the formation of a talik around the injection cell. We have adjusted the text accordingly.

L367-369: "*This is likely due to the formation of a talik beginning in the winter between years 48 and 49. The initial talik encompasses the ancient carbon injection cell at a depth of 1.15 m, allowing for extended transport periods, including during winter.*"

33. - l 352, « vertical mobilization »: once again the involved mechanism must be explicated.

This is now mentioned earlier on (see comment #30) and is also part of the discussion.

34. - l 407-411 : « We partly address this by representing a converging slope model setup, where the cell width in transverse direction varies depending on the distance in longitudinal direction. This way, the surface area of the catchment is preserved, and it is possible to accurately represent water and energy balances as well as infiltration and evaporation rates throughout the catchment. This approach has previously been applied by Gao and Coon (2022). » I do not think that using a one cell-thick discretization in the transversal direction may allow to simulate the effect of the watershed geometry, either convergent or divergent

(using the terminology of Gao and Coon 2022). Nor in the present manuscript or in Gao and Coon 2022 are presented arguments for supporting the validity/usefulness of such a 'pseudo-3D' meshing methodology. A proper comparative study should be done for this, between results obtained with« pseudo-3D » meshes and with full 3D meshes. Of course with only one cell no lateral fluxes may be computed.The only interest I would see would be to weight the inward fluxes through the top cell face according to the area of the cell, but then why not simply apply a spatial weighting on the incoming fluxes prescribed by the boundary conditions? Including this in the meshing seems to me inappropriate and confusing. Anyway in this case the methodology used for computing the cells widths must be explicated.

See our reply to comment #9 and the proposed changes to the text. We aligned the mesh with the topographic gradient and thus the flow; by construction there is no tangential lateral flow, only in the vertical and along the topographic gradient. Flow-aligned meshes are a standard approach for reducing the computational complexity of hillslope simulations, as noted in our reply to comment #9.

35. - l 465, « Under the simulated environmental and soil physical conditions in this study, »: This precision hold true not only for this point, but for all the listed conclusions. The writing of the manuscript should better reflects the fact that this is a numerical study of transport in a specific site, with no possibility of automatic generalization for permafrost regions as a whole.

This very important point is now discussed more thoroughly in the discussion. While the results of this study may apply for foothill systems, they may unfold very differently for permafrost locations in e.g., plains with micro-topography.

L443-451: "*While the Endalen valley represents a single case within the context of high Arctic hillslope systems underlain by continuous permafrost, similar systems are widespread across the Arctic, making this site a valuable example for studying permafrost carbon transport. The slope of approximately 12° in our study area is representative of gently sloped terrain in permafrost landscapes, which accounts for about 20% of the total permafrost region, commonly found in areas such as Greenland and parts of Canada (Hamm and Frampton, 2021). However, our findings may not be directly transferable to other permafrost settings, such as flat terrain, where differences in groundwater flow velocities, saturation levels, and small-scale processes like cryosuction could significantly alter transport dynamics, especially in landscapes influenced by microtopography, such as polygonal tundra. Additional studies across a variety of terrains, supported by reliable field observations, are necessary to gain a more comprehensive understanding of lateral carbon transport in permafrost environments.*"

TECHNICAL CORRECTIONS :

36. - l 25: Missing ) at the end of the line.

Noted

37. - l 29, l30, and elsewhere: I would recommend to systematically use 'organic carbon' instead of just 'carbon' for naming the C part of the organic matter stored in permafrost.

We prefer to keep using the word "carbon" as it is mentioned nearly 180 times in the current manuscript. We will clarify in the introduction that we refer to organic carbon throughout the manuscript.

38. - l 32: Missing s at 'question'.

Noted

39. - l71-84: Part of this paragraph should be in the Methods section (e.g.: choice of distinguishing four carbon pools and using labelled tracer for identifying them).

This final paragraph in the introduction section summarizes our approach to address the questions that arise from the literature summary in the rest of the introduction and hence is essential to the formulation of the hypothesis. A thorough explanation of the carbon pools and labels follows in the methods section.

40. - l 151, «bottom horizontal boundary »: According to Figure 3, the bottom boundary is not horizontal.

We will remove "horizontal"

41. - l 251-260: These information should be included in the Methods section, along with a figure for quantitative locations of the injection points and of the measurement points within the modelling domain.

We agree that a zoomed-in figure with the injection points is beneficial for the interpretation of the results and we added such an illustration to the Appendix (Fig. A1). References to this figure have been added in section 2.4. We prefer to keep the introductory sentences in the results section as they currently are to preserve the flow of reading and introduce the way in which the results are presented.

[Figure]

*Fig A1: Location of the injection points for the seasonal carbon transport simulations (TOL carbon (blue cross) and buried carbon (orange cross) as well as for the climate change scenarios. The shallower ancient carbon (Ancient 1, red cross) is located just below the current active layer extent and the deep ancient carbon (Ancient 2, green cross) is located about half a meter below the present day active layer. Tracer mass just after injection is given in blue to yellow.*

42. - l 258, « explicit », « continuous in space » : Odd vocabulary. A BTC represents the temporal evolution (not exactly continuous, since there is a time discretization) of concentration at a given location in space, while a plume vizualisation represents the spatial distribution of concentration at a given moment.

The sentence has been change for clarity:

L272-275: "*Hence, the BTC represents the temporal evolution of tracer concentration at a given location, showing how the concentration changes over time, though discretized due to time steps in the model. The plume distribution, on the other hand, represents the spatial distribution of tracer concentration at specific moments in time, showing how the tracer is distributed in space at a particular point in time.*"

43. - Legend of table 2: The use of different concentration thresholds for TOL and buried carbon should be mentioned in the body of the text, in the Methods section.

We added this information to section 2.5 "Model forcing" under "Present-day weather conditions".

L199-202: "*In the model output, a BTC is defined when the concentration reaches a minimum threshold of 0.01 mol at a given observation point. We distinguish between surface and subsurface BTCs, with arrival times marked by the concentration exceeding and subsequently falling below this threshold. For surface transport of the buried tracer, however, the threshold is set to 0.0005 mol due to the small amount of mass transported along the surface.*"

44. - l 273: « when all runoff is occurring in the subsurface »: Then it is not run off, but groundwater flow.

We prefer to keep our original phrasing as it encompasses both percolation and groundwater flow, as well as highlighting the surface-subsurface nature of runoff and flow interactions. Subsurface runoff is a defined hydrological term (e.g., Pilgrim et al. 1978).

45. - l 285: Do not refer specifically to a Figure in Supplementary material (Fig. S3) in the body of the text. Instead, refer to the supplementary material as a whole.

Supplementary information will be moved to the Appendix

46. - l 286 – 287 : « However, given the specific solute transport patterns in this model, » What is meant here ? Unclear.

We reworded this to:

L306-307: "*However, due to the transport behavior of the solute in the model, the tracer is mostly located in the highly saturated zone, where mineralization is very low or absent (e.g., on 14 June, Fig. 7a).*"

47. - l 299, « vertically transported upward »: Oddly said. Strictly speaking, the topographical effect mentioned here does not generate ascendant flow.

In this part of the domain, the local topography combined with the undulating permafrost table, which acts as an impermeable boundary for flow, causes the active layer to be locally saturated with liquid water at the time of tracer arrival, which in turn enables the observed vertical/sub-vertical tracer movement. The manuscript text will be revised as follows:

L318-321: "*A small fraction of buried carbon tracer (tracer mass < 0.005 mol) experiences surface transport (also visible in Fig. 9) because it gets transported upwards by groundwater upwelling. This is caused by a combination of the terrain unevenness and the undulating impermeable permafrost table, causing local downslope water accumulation and a fully saturated active layer.*"

48. - l 302: Do not refer specifically to a Figure in Supplementary material (Fig. S4) in the body of the text.

Supplementary information has been moved to the Appendix

49. - Caption of Fig. 6, « Note that the tracer mass is restricted to the uppermost subsurface cell in this snapshot and is difficult to visualize in this illustration » : True, then this figure has to be improved. May be that plotting the two variables separately would be an option? Besides, Fig. 6 is mentioned only very briefly once in the text. Either it should be deleted or more extensively commented.

We added an inset to the figure, zooming into the area where there is tracer and also changed the color scheme so that it becomes more visible (see Fig. 7 below). Further, we added a snapshot for buried carbon to reduce the amount of Figures and deleted a the supplementary Figures S3 and S4.

[Figure]

*Fig 7.: Temporal snapshot of potential normalized mineralization rate and component mass plume of TOL carbon spreading throughout the transect on 14 Jun. Light-orange areas mark comparably low potential mineralization rates, dark-orange areas indicate the highest potential for mineralization. Component mass for the selected date (14 Jun) is added as a blue to yellow overlay. Note that the tracer mass is restricted to the uppermost subsurface cell. The inset at the top right shows a zoomed-in version of the area in which tracer is present.*

50. - l 315, « some tracer is moved upwards »: see comment on l 299.

See comment #47

L336-338: "*Due to the unevenness of the terrain and highly saturated conditions, some tracer mass is transported upwards towards the surface (e.g., at 1005 m on 23, 25, and 27 August in Fig. 9), but the amount of buried carbon tracer being transported on surface is small (max. 8% on 26 July, Fig. 5b and d, green shaded area).*"

51. - l 317: 12 % does not obviously look negligible.

Changed to "small"

52. - Figure 7: I can't see anything regarding the solute mass. This Figure should be thoroughly reworked so that it becomes informative. Besides, what is the « ponded depth » ?

We added the information about the ponded depth and also the thaw depth into the figure caption. Apologies for the oversight. However, we argue that the tracer mass is as visible as possible while still showing the entire transect of interest. For better visualization, we will decrease the opaqueness of the mesh and the thaw depth line.

"*Figure 8*. 2D representation of the top organic layer (TOL) carbon plume dispersion between the point of tracer injection (x = 1000 m) and the last observation point (x = 1020 m) for selected dates during summer after tracer injection. Elevation is given in relative elevation to the conceptual valley bottom (surface of the valley bottom = 0 m). Tracer mass is given in mol and is masked to only represent values > 0.01 mol. The blue solid line marks the water table above the surface (ponded depth), the black dashed lines indicates the thaw depth at each date."

53. - Figure 9: The annual cycles/peaks should be discussed. « The visual increase in mass above 100 mol in the injection columns in (a) and (d) is not a physical phenomenon but a result of aggregating and rounding errors during post-processing of the model output. » Then the post-processing should be improved.

Please see response to comment #54 below.

54. - l 350-355: If the post-processing method does not allow to conclude, then it should be improved.

The issue here is that a limited precision is used in output files causing very slight round-off errors. A limited precision is used due the large quantity of output data. To improve this, an increased precision in the output files could be implemented, but then the simulations would need to be re-run, which is not practical as the runtime is over 3 months wall clock time. As this is a minor issue which has no bearing on the results of the study we prefer to keep the data processing as is, but propose to improve the figures (Fig 10 and Fig 12) by truncating to the 100 molar upper limit where applicable. This correction should help clarify the presentation of results without distracting from cumbersome explanations of data post-processing. The explanation of data post-processing will be provided in the AppendixC.

[Figure]

Fig. 10: Ancient carbon mobilization upon gradual warming in the RCP4.5 (a-d) and RCP8.5 (d-f) scenario simulations. Tracer injection mass (a and d) indicate when the tracer gets mobilized at the initial injection point in the respective cell (dashed line) and integrated vertically over the entire injection column (solid line). Red and orange lines represent ancient carbon in 1.15 m and 1.55 m depth, respectively. When molar mass reaches 0, all tracer has been laterally transported away from the injection point. Tracer BTCs at 10 m (b and e) and 20 m (c and f) show the arrival times of tracer mass released at the injection column. Note that rounding errors of small amounts of mass have led to a tracer mass > 100 mol when integrated over the entire column. Hence, values > 100 mol were truncated to 100 mol. For details see Appendix C.

And Appendix C

55. - Caption of Figure 11: « The visual increase in mass above 100 mol in the injection column in (b) is not a physical phenomenon but a result of aggregating and rounding errors during post-processing of the model output. » Once again the post-processing method should be improved.

Please see response to comment #54 above and the revised Figure 12.

[Figure]

Fig. 12: Mobilization of ancient carbon upon abrupt active layer deepening. The injected tracer mass (a and b) indicates when the tracer gets mobilized at the initial injection point in the respective vertically integrated injection column. Solid and dashed lines represent injection mass mobilization under the +5 ∘C and the +3 ∘C scenario, respectively. Tracer BTCs at 10 (c) and 20 m (d) indicate when and how fast the tracer passes through the observation column. Red and orange lines represent the tracer representing ancient carbon at 1.15 m and 1.55 m depth, respectively. Note that rounding errors of small amounts of mass have lead to a

*tracer mass > 100 mol when integrated over the entire column. Hence, values > 100 mol were truncated to 100 mol. For details see Appendix C.*

56. - l 414-420: this should be part of the introduction, not the discussion.

This part places the work in the context of what has been done on permafrost carbon transport modeling previously and how it can be improved in the future. We therefore think it is important to mention this in the discussion to highlight future development of this work and prefer to keep it as is.

57. - l 420-425: this should be part of the Methods section, not of the Discussion.

This section is discussing limitations of using a non-reactive representation of the tracer and we feel it is therefore important to elaborate on in the discussion section. We wish to refrain from discussing alternatives in the methods section.

58. - l 432-447: This should be part of a perspective section, not of the Discussion.

We feel it is useful and customary to include perspectives such as this in the discussion section. A dedicated section for this part would likely hinder the flow of the discussion. We therefore prefer to keep this part in the discussion.

59. - Section 4 Discussion: Given the parts that should not be included in the discussion (see the three comments above), the Discussion section is rather short (a bit more than one page), and lack of in depth analyses of the produced results. I recommend to strengthen this section, including for instance elements for linking the simulated behavior and the site specificity.

We extended the discussion by a discussion about the site specificity, its role in a wider context in the Arctic, and added a discussion point on wetness conditions. We believe this will strengthen the discussion. See our response to comment #35.

60. - l 452, « mechanical transport »: Sounds weird. Passive transport would be more relevant.

We removed "mechanical"

**References**

Gao, B. and Coon, E.T., 2022. Evaluating simplifications of subsurface process representations for field-scale permafrost hydrology models. *The Cryosphere*, *16*(10), pp.4141-4162.

Hamm, A. and Frampton, A., 2021. Impact of lateral groundwater flow on hydrothermal conditions of the active layer in a high-Arctic hillslope setting. *The Cryosphere*, *15*(10), pp.4853-4871.

Jan, A. and Painter, S.L., 2020. Permafrost thermal conditions are sensitive to shifts in snow timing. *Environmental Research Letters*, *15*(8), p.084026.

Magnússon, R.Í., Hamm, A., Karsanaev, S.V., Limpens, J., Kleijn, D., Frampton, A., Maximov, T.C. and Heijmans, M.M., 2022. Extremely wet summer events enhance permafrost thaw for multiple years in Siberian tundra. *Nature Communications*, *13*(1), p.1556.

Pilgrim, D.H., Huff, D.D. and Steele, T.D., 1978. A field evaluation of subsurface and surface runoff: II. Runoff processes. *Journal of Hydrology*, *38*(3-4), pp.319-341.

**REVIEWER 2**

Comments by the reviewer are in black text

Our responses are in blue text

*Where applicable, proposed changes to the revised manuscript are in blue and italics*

**Major comments**

The manuscript describe a study using a numerical model to investigate the factors that influence mobilization and transport of carbon in soils in permafrost regions. Warming and permafrost thaw are likely to cause the transport of carbon from soils through aquatic systems, so a better understanding of physical mechanisms is crucial to the development of more accurate models and climate simulations.

We thank the reviewer for their thoughtful review. To improve the current manuscript, we have made several modifications to address their comments. The main focus of the improvements is on the justification of the meshing strategy, which has been neglected in the original manuscript.While it is not feasible to simulate several different sites with this model, we now better clarify how generalizable our results are in the wider context. Further, we have added several clarifications to the text where needed to improve the readability of the manuscript. We are confident that the changes have contributed to significantly improving the manuscript.

1. The study is designed as a highly idealized case for a small area in Svalbard. The grid setup is not standard. The time periods noted in the text are variable and confusing. The short time period of the abrupt that complicates interpretation. This application of a single model in this fashion makes extension of the study findings to broad regions of the Arctic questionable. The results would be more meaningful by applying the model for at least one or two other configurations based on observations of soil texture and organic carbon content, weather data, elevation gradients, which are available at select sites in Europe, Siberia, Canada, and northern Alaska. Doing so would allow the authors to have more confidence that what they are seeing represents a robust response and not an artifact of a unique model setup. The authors should also state the choice of application to the Endalen valley on Svalbard where observations are sparse. Is there a rich history of field study there?

We realize the description of the grid setup was not adequately explained and we will revise the text to better provide the justification. The key point is that the use of a flow-aligned, variable-width representative hillslope is a well-established modeling approach in catchment and hillslope hydrology. Please see our response to comment #9 by Reviewer 1. And while we agree that a study of multiple sites would be of interest in a future project, that's clearly out of scope for this project, which addresses our study site and takes advantage of available field observations. Importantly, we are confident that the understanding gained in this study is transferable to other similar locations and sites,

although of course not generally to any permafrost region. We clarified and elaborated on the potential applicability to different locations in the revised manuscript. Further, we addressed the issue of only modeling this specific site by better highlighting the available field observations, which include groundwater flow velocity observations with the help of tracer experiments. This is in line with comment #27 by Reviewer 1.

2. The 0.25 km2 catchment is relatively small. Why not use a traditional uniform horizontal grid and with vertical soil layers? How many grid cells would be needed at resolution of 1 m? Would the computational expense be prohibitive? How does the setup result in "a natural equilibrium without artificially imposed boundary conditions", and what does that phrase mean? Boundary conditions and zero-flux boundaries in a typical 3-D model setup could be the same as those in this study. In other words, I don't see the advantages of this current grid mesh, nor the implications for the interpretation of results. While the authors refer to computational expense, more justification should be presented.

A variable resolution mesh with high refinement around a region of interest is very standard practice in groundwater and surface/subsurface integrated hydrologic modeling. This is especially true for combined flow and transport models because finer meshes are typically required to resolve transport. In our case, we refined our mesh in the region of our numerical transport experiments and kept a coarse mesh away from that region to set the appropriate flow conditions for transport. We stand by this choice of mesh. However, we tested several different mesh resolutions for the upper slope area and came to the conclusion that a mesh resolution of 5.8 m per column in the upper slope area is giving us the most robust results. Hence we changed the description of the seasonal carbon transport results to the new mesh (see also comment #14 by Reviewer 1)

The advantage of the domain setup used in this study is it allows for a feasible computational runtime and simplifies analysis of results in a way which is suitable for the aims of the study. For reference, the climate change scenarios alone required over 3 months runtime to compute on a high end workstation. Although a full 3D domain mesh could in principle be considered, it is not warranted for the aims of this study.

We agree the phrasing "a natural equilibrium…" is unclear and revised the text. What we meant was that we use hydrometerological forcing as driving conditions for the model, as opposed to cumbersome and often simplified boundary conditions. This enables the thermal-hydrological dynamics of the model to be controlled by weather inputs in a way which more closely resembles nature by precipitation, air temperature, and factors influencing evaporation, etc, as opposed to boundary conditions representing hydraulic pressures or hydraulic gradients, and thermal inputs, etc. Importantly, this is only possible for integrated surface/subsurface models like ATS.

3. The wide variation in time steps mentioned is awkward and makes interpretation of the influence on results difficult. There is a rate for tracer injection of per second and a time step of monthly for the warming experiments. If it were implicitly daily, could mineralization in the future simulations like the present day

simulation? The time step of per second is confusing in light of a monthly step in the future simulations.

Internally, computations are performed with an adaptive time stepping scheme but output frequency is user-controlled. For the multi-decade simulations representing climate change scenarios it is of course not practically possible to output results at a high frequency such as every second. Therefore, output frequency is assigned to monthly snapshots in those cases. However, for present-day conditions and when focusing details such as seasonal variability, it is relevant to consider higher frequency outputs. Although this leads to a range of different time representations in outputs, it is necessary and reflects the complexity and novelty of our study. We added a sentence explaining that ATS is using adaptive time stepping with a maximum time step of 1 day and a minimum timestep of 1e-10 seconds and that output frequency depends on the scope of the simulation (i.e. seasonal vs. climate change scenario).

L118-121: *"ATS employs an adaptive time-stepping scheme with user-defined minimum and maximum time steps. In this study, we set the minimum time step to 1e-10 days and the maximum to 1 day, allowing for efficient and accurate simulations across different seasons, which require higher or lower resolution time stepping, as well as for a user defined output interval."*

4. The term seasonal variability in the title is odd and unnecessary, as seasonal variability in a study like this is essential. Having the term 'sensitivity study' in the title would be more meaningful.

Seasonal variability stands in contrast to climate change and highlights the fact that we do both in this study. However, we do not conduct a full sensitivity study regarding model parameters or forcing. Hence, we prefer to keep the title as it is in the current version of the manuscript.

Minor comments

5. Line 6: grammar, "Here, we analyze of solute transport…"

Noted

6. Line 43: Unclear. What is "they" in the statement "With permafrost acting as a largely impermeable layer between the two, they are…"

Changed "they" to "the shallow and deep aquifers are mostly disconnected systems"

7. Figure 3 caption: It appears from the graphic that width (y-direction) decreases with distance in the x-direction. As x increases from 0 to 1000, the width gets smaller. Please clarify.

We added a sentence to the Figure caption to explain this nature of a convergent hillslope setting.

"***Figure 3****. Representation of the mesh (blue shape) used for model simulations based on the catchment illustrated in Fig. 2. Dimensions of mesh elements vary in x-, y- as well as in the z-direction. The mesh (blue shape) represents a converging hillslope system with a wider width at the catchment boundary and a small width at the outlet. The green shaded area indicates the upper slope in which runoff is generated, the red area highlights the main area of interest later used in the presentation of the results, and the gray shaded area indicates the buffer zone. While the width (y-direction) increases uniformly with distance in the x-direction, column length in the x-direction is large in the upper part of the catchment (~23 m) and smaller in the last 40 m of the catchment (0.5 m between 1000 m and 1020 m and 1 m between 1020 m and 1040 m). In the z-direction, cell resolution is higher (2 cm cell-1) in the active layer (upper 1.2 m) and lower in the permafrost (up to 2 m cell-1). The total depth of the domain is 40 m and the elevation is defined relative to the surface at the valley bottom, which is set to 0 m elevation.*"

8. Line 159 states that the model is run for an average yearly cycle. Then at line 172 there is reference to year-to-year differences. How can year-to-year differences in spinup be determined? Further clarify the spinup time period and protocol.

For each day of every year of the spinup, a certain state of the ground and the ground surface is calculated by the model. In the beginning, when the model is not in a cyclic steady state yet, there will be a difference between those days between one year and the next. Once these differences decrease to a negligible amount, the system is determined to be in "cyclic steady state". We included a plot in the Appendix showing differences in the surface temperature at a specific location for each day of the 10 years, showing that the difference between each e.g., June 1 becomes negligible. This is common practice in the modeling community (see comment #22 by Reviewer 1). We added the following sentence to the manuscript:

L171-172:  "*We confirmed that the temperature is in a cyclic steady state at the end of the 10 year spinup period (Fig. D2).*"

9. Line 171: typo "a the"

Noted

10. Line 173: What ten years? Is weather data not for an average yearly cycle (climatology)? Be specific about the ten year period.

We changed it to "ten years of average weather conditions"

11. Line 186: Clarify how numerically does a tracer become injected at a rate with a time step of a second. What is the model time step? If it is daily, how can something occur every second? Does the mass enter as a total each day? Intrinsic time step of the model is not clear.

ATS uses an adaptive time stepping scheme with user-assigned limits. This is advantageous as it adjusts the time step automatically using an iterative procedure to ensure numerical convergence depending on the dynamics of the system. For example, in spring and summer, with meltwater infiltration and variable groundwater flow, the time stepping scheme iteratively reduces the computational time step typically to a small value (less than a second), whereas during less dynamic periods, such as winter, the time step is typically increased to a maximum of one day, which helps reduce runtime. Of course the specified quantity is an injection *mass rate*; the actual mass injected during a time step depends on the length of that time step (see comment #3).

12. Line 235: typo, for for

Noted

13. Figure S3: component mass overlay is difficult to see in the graphic. Also add label for elevation for the y axis to make clearer what is being shown.

We reworked Figures S3 and S4 to better visualize component mass and added labels to the x- and y-axis. However, we decided to remove the figures with 9 panels each and replace them with the snapshot Figures like the one in Fig. 6 in the original manuscript. The new Figure 7 is now showing snapshots for both, TOL and buried carbon. Please also see our response to comment #49 by Reviewer 1.

14. Line 18 in supplement: units of %C ¿ 10 are unclear.

Noted

15. Line 337: typo, "have lead to"

Changed to "have led to"

16. Line 357: Do the magnitudes (+25 and +60 cm) of the abrupt active layer deepening have any grounding in reality? That is, do any studies report abrupt thaw that is some meaningful fraction of the amount modeled here? Some perspective on the magnitudes would aid in interpretation of the results.

We added a reference to active layer variability in the Adventdalen valley. Strand et al., 2021 have shown that an abrupt increase of 30 cm is possible between a cold and a warm year. Hence, the modeled abrupt thaw is within the currently observed variability and additionally exaggerates it by a factor of 2. In a rapidly warming Arctic such as in the RCP8.5 scenario, heat extremes will be more common and may lead to abrupt thawing of the active layer, which is why we choose to consider scenarios which may seem extreme today, but may well be feasible within the time frame of the warming scenarios / coming century.

L382-386: "*Such variability is not unprecedented. Strand et al. (2020) showed that variations in active layer depth in Adventdalen between cold and warm years can reach up to 30 cm. Our simulations fall within this range, but intentionally amplify the effects of*

*an exceptionally warm year on active layer dynamics, resulting in an exaggerated active layer deepening by a factor of two. The abrupt deepening causes the entire BTC of the ancient carbon tracer at 1.15 m depth to start and finish within the same year and initiates the release of the tracer at 1.55 m in the +5°C scenario (Fig. 12).*"

17. Line 368: "...is released from both the injection column…" grammar and unclear meaning.

Removed "both".

18. Line 391: At the mention of ancient carbon, it would be helpful to remind the reader that this is frozen carbon. For readers that skim the abstract and discussion, the terms ancient and buried will have unclear meanings. Making clear that ancient means ancient currently frozen carbon would help.

We added this to the text.

L419-420: "*Under the assumption of an abrupt increase in ALT due to one very warm year ($T_{avg}$ +3, +5), the transport of ancient, currently frozen carbon tracer becomes more similar to the transport of buried carbon.*"

References:

Strand, S.M., Christiansen, H.H., Johansson, M., Åkerman, J. and Humlum, O., 2021. Active layer thickening and controls on interannual variability in the Nordic Arctic compared to the circum‑Arctic. *Permafrost and Periglacial Processes*, *32*(1), pp.47-58.

---

## Author Response (AR2)

**Response letter to reviewer comments**

The authors have significantly enhanced their manuscript, providing additional information, reorganizing the paper a bit and clarifying the aims and limits of their work. The newly included Appendixes are of high interest.

Meanwhile, some of my methodological concerns are not fully alleviated. First, the novelty of the proposed pseudo-3D approach for the simulation of coupled water, heat and solute transfer in a real watershed is still not acknowledged, and consequently the need of testing it is still overlooked. Second, the proposed convergence study is only partial, and especially it does not include the main area of interest. Additional numerical experiments must be undertaken for consolidating the methodological aspect of this work.

So I recommend a supplementary revision step for this manuscript.

Response: We thank the reviewer for their careful evaluation of our study, which has provided many useful insights leading to several significant improvements and clarifications. Our intention is in fact not to conduct a site-specific study, instead, we use available site data to design a realistic and reasonable semi-generic model broadly representative of convergent hillslopes, with the aim to investigate the relative differences of transport of solutes and carbon released at different depths in the active layer and permafrost. The question whether a variably width hillslope approximates a 3D catchment is irrelevant in the context of our study and is the result of a miscommunication about what we are trying to accomplish. Our model domain is inspired by the site and captures key physiographic characteristics of the site, but it is intended to be a synthetic domain that is broadly representative of hillslopes in continuous permafrost regions. We are not attempting a case study that would require 3D representation of heterogeneity. The convergent variable-width hillslope is uncontroversial and based on fundamental physics of flow (model sides must correspond to flow streamlines and thus lines of steepest descent of the surface elevation to be no-flow boundaries), and is necessary to preserve the contributing area to a stream segment for a given travel distance to the stream, where distance is defined along the flowpath. Importantly, it ensures the model is consistent with a convergent hillslope conceptualization.

Convergent hillslopes are well understood to be much more common than uniform or divergent hillslopes, so the choice of a convergent hillslope to study is an obvious one. The site which we base the model conceptualisation on corresponds to a small region along one side of a hillslope of Endalen valley in Svalbard. We use this as the basis for design and analysis because it corresponds to a well-defined convergent flow system representative for hillslopes in many valley systems throughout Svalbard and the wider Arctic. To eliminate potential misunderstanding between the hillslope that we conceptualize, versus the much larger Endalen valley catchment, which we do not consider, we have refrained from the use of the term catchment in favor of the term hillslope in our revised manuscript. This change

will align better with the terminology commonly adopted in hillslope hydrology and clarify the context of the model intention.

We have clarified the text throughout the manuscript, and especially emphasized the aims of the study, i.e., as a semi-generic representation of a convergent hillslope flow system, in the Introduction section, and clarified the approach in the Methods sections, as well as in Appendix A. Several figures have been improved to help clarify these points and new ones have been added. We have also conducted yet another mesh discretization study, focusing on the transport region (i.e., the main area of interest of the model analysis), demonstrating its robustness, included in the updated Appendix D. Specifics and details are provided in our responses below.

COMMENTS :
1. L 515-523 (problem with lines numbering, I take the manuscript version with apparent modifications as reference): "similar systems are widespread across the Arctic" Please quantify the % of surface coverage of 'high Arctic hillslope systems underlain by continuous permafrost' over the total permafrost area, over the continuous permafrost area and over the Arctic region. Please also include the ecotype (bare soil, tundra …?) in the site characteristics.

Response: This prevalence of similar hillslopes in permafrost regions in the Arctic is reported in the reference cited in the text (Hamm and Frampton, 2019). Continuous permafrost is generally accepted to be landscapes underlain by permafrost with 90% or more in extent, and a reference to Brown et al. (2002) has been added to the text which highlights that Svalbard is underlain by continuous permafrost (line 459 of the revised version). While detailed quantification of soil types may be interesting it is unfortunately not realistically possible to provide with reasonable accuracy, but the qualitative description suffices for the purpose of the discussion in our study.

2. L 76: In my opinion it should be at least stated here that considering non-reactive tracer is a simplifying assumption.
Response: This part of the text is referring to Jafarov et al. (2022).

3. L 126, Qs in Eq. (1) : OK, then to which physical process is associated Qs? Is Qs set to 0 in your simulations? The reader should be able to understand this after this paragraph.
Response: Qs is a source/sink term. This part of the text has been clarified in the revised manuscript (lines 127-135). The details of solute injection during the transport simulation phase are provided in Section 2.4.

4. L 188-190: "This strategy, together with variable-width elements that preserve flow convergence, provides an accurate representation of catchment-scale processes without

the computational burden of a full 3D model (Appendix A and D)." Please find hereafter my comments on Appendix A and D. Although things are a bit clearer now, I think that there is still some work to be done. Stated here that the representation of catchment-scale processes is accurate still look like an unjustified statement, see my comments on lines 602-606.

Response: Our aim is not to conduct a site-specific study which reproduces a particular hillslope or catchment; rather, we use available site data to design a realistic semi-generic/stylized model representation of the hillslope system, with the objective of investigating solute transport in the active layer. Semi-generic approaches are commonly adopted and often necessary for remote cold regions where data availability is limited (e.g., Lemieux et al. (2024), Lamontagne-Hallé et al. (2020), Walvoord and Kurylyk (2016)). The strategy of studying water and solute movement along hillslopes separately from movement along stream channels is well-established in catchment science and global change research (see, for example, the extensive discussion in the recent review by Fan et al. 2019). We selected this location as the basis for the model design as a representative hillslope for many valley systems throughout Svalbard and the wider Arctic. We have clarified the aims and intentions with the approach throughout the revised text, in particular in the Introduction section (lines 80-89) and Methods section (Section 2.2 lines 113-120, Section 2.3 lines 145-176) as well as Appendix A (lines 536-555) and corresponding figures Fig. 2, 3, and A1. We have also conducted yet another mesh discretization study, focusing on the transport region (i.e., the main area of interest of the model analysis), demonstrating its robustness, included in Appendix D (lines 671-676) and Figs D5, D6.

Lemieux, J., Frampton, A., Fortier, P., 2024. Recent Advances (2018–2023) and Research Opportunities in the Study of Groundwater in Cold Regions. Permafrost & Periglacial ppp.2255. https://doi.org/10.1002/ppp.2255

Lamontagne-Hallé, P., McKenzie, J.M., Kurylyk, B.L., Molson, J., Lyon, L.N., 2020. Guidelines for cold-regions groundwater numerical modeling. WIREs Water 7, e1467. https://doi.org/10.1002/wat2.1467

Walvoord, M.A., Kurylyk, B.L., 2016. Hydrologic Impacts of Thawing Permafrost—A Review. Vadose Zone Journal 15, vzj2016.01.0010. https://doi.org/10.2136/vzj2016.01.0010

5. L 592: Why 1040 m?

Response: This is the length of the model domain representation of the hillslope from the upper boundary to outlet. We have clarified the model design procedure in the Methods section (Section 2.3 lines 145-176) and Appendix A (lines 536-555) and corresponding figures Fig 3 and A1.

6. L 600 : "dominant topographic controls on flow (Dunne and Black, 1970; Anderson and Burt, 1978) " Why citing experimental studies made in peculiar hillslopes in nonpermafrost areas, as relevant for characterising the flow in this peculiar permafrost watershed? Obviously topography is important for flow in continental surfaces, but why using these references here for justifying this is not clear to me.

Response: Solute transport is driven by surface overland flow and shallow groundwater flow in the suprapermafrost flow system of the active layer during the unfrozen period. The governing physical principles of surface overland flow and groundwater flow during this period are the same as for non-permafrost regions. Indeed, the suprapermafrost flow system in continuous permafrost regions is even simpler than typical non-permafrost locations. In particular, for continuous permafrost environments, topography exerts a strong control on flow in the relatively shallow active layer as permafrost acts as an essentially impermeable boundary inhibiting deep groundwater recharge. The permafrost-specific processes of relevance, heat transport with freeze/thaw and cryosuction, are largely one-dimensional in the vertical direction, and are thus not affected by the spatial structure adopted (column models are adequate in the absence of flow).

7. L 602-606 : Fan and Bras 1998 proposed 1D analytical solutions for flow in and over a hillside with a Darcy-type equation for either divergent, convergent or uniform hillslopes. Then, for applying these analytical solutions at the catchment scale, they suggested to divide the complex topographical surface of the catchment in elementary hillslopes of one of these three basic types. Troch et al. 2003 then extended this approach to nine basic plan shape / Curvature profile types hillslopes, and by using a numerical resolution of Boussinesq equation for flow on the obtained 1D domains. Paniconi et al., 2003, made a comparative study of the results of this approach with the one obtained with fully 3D Richards equation-based flow simulation, once again only for basic hillslope types. Finally Hazenberg et al., 2015 apply this approach for LSM simulations. On the other hand If I understand correctly what you do is to approximate the whole catchment as a kind of convergent hillslope with the thalweg topographical line as longitudinal profile, then to approximate this 3D convergent hillslope as a 2D transect with variable width of the unique y-axis cell, and then to apply a numerical resolution of variably saturated flow and transport with freeze/thaw to the obtained pseudo-3D representation of the watershed. This would be significantly different from the developments cited above, although directly inspired by them. I would recommend this new approach to be explained and tested. If I correctly understood, in Gao and Coon 2022, a basic geometry in the style of Troch et al. 2003 was dealt with, without studying whether or not such a simplified geometry allows to catch the dynamics in real, complex watersheds, but rather for giving a theoretical test case for evaluating relative importance of various processes in the numerical simulation of permafrost dynamics. And no comparison with full 3D results was proposed in this later paper. So I think that this new approach is potentially promising, but should be carefully assessed, from at least two points of view :

- Is the simplified approach developed in Fan and Bras 1998, Troch et al., 2003, Paniconi 2003 and Hazenberg et al., 2003 for Boussinesq equation applicable to the modelling of permafrost dynamics + solute transport, i.e. for 3D simulations of coupled flow, heat transfer and solute transfer?
- Is this simplified approach applicable to the considered watershed, given its complex morphological structure ? I Guess that this approach could not always be successfully applied, for peculiar watershed topologies for instance. Furthermore, I wonder whether or not the proposed methodology for building the y-axis width along slope does conserve the properties of the hypsometric curve of the watershed? I think this point could be important for properly handle the altitudinal distribution of contributing areas, and thus for the time of concentration.

Such an assessment would likely request large computational means, which should not be a problem given the High Performance Computing capabilities of ATS.

Response: It is important to make the distinction between a catchment with a "complex morphological structure" and a hillslope. The strategy of studying hillslope processes separately from stream processes is well-established in catchment science and global change research (see, for example, the highly cited review paper of Fan et al. 2019); we are taking this standard approach. Granted, our convergent hillslope has some internal structure, but variation in elevation within grid cells is small compared with variation across the grid cells, as is obvious from our revised Fig 2. More to the point, we emphasise again that our aim is not to create a site-specific model that reproduces the "complex morphological structure" of that site; this would be outside of the scope and aims of our study and would also require field measurements that are not available. In our opinion, such place-specific case studies like that would be less interesting scientifically because it would tell us much about the single site but would be less generalizable because of uniqueness of place.

Instead, we use information from the hillslope site in Endalen as a representative site for design of a semi-generic model, with the aim to study the relative impact of solute transport released from different depths in the active layer and permafrost. For this, it is not necessary to capture the precise or absolute timing of seasonal events as our findings are not dependent on those specifics. Further, the hillslope that we use as the basis for model design is a topographically well-defined convergent flow system. Even though our aim is to construct a representative semi-generic mode, we adopt a variable width approach as it preserves the contributing area to a stream segment for a given travel distance to the stream, where distance is defined along the flowpath, which ensures the model design is consistent with convergent hillslopes. As noted in our previous responses, the plan shape and profile (and thus the hypsometric curve) are preserved by construction. This greatly simplifies the model representation in terms of its applicability in allowing for topography-controlled runoff and adopting lateral no-flow boundary conditions. The approach using a surface energy balance model also simplifies

the implementation of surface boundary conditions, which can be based on readily available hydro-meteorological data from weather stations, avoiding the uncertainty of imposing surface temperature-based boundary conditions.

Several studies consider variable-width hillslopes in addition to the ones we have already cited. Although the specific goals and methods differ, the conceptualization of variable width hillslopes to preserve the area-distance relationships is the same for all. Studies on hillslope representation using the Boussinesq approximation for different hillslope width functions include Paniconi et al. (2003), Troch et al. (2004), Hilberts et al. (2004), Hsieh and Huang (2023). Indeed, although Hazenberg et al. (2015) apply variable-width hillslopes to Earth System Models, Hazenberg et al. (2016) evaluates the approach with site measurements from a well-characterised catchment, albeit for uniform width conceptualizations. Fan et al. (2019) again highlights the use of a variable width concept, and Chaney et al. (2018) perform large-scale analyses of variable width hillslopes at a relatively coarse scale, collapsing those into a few canonical simulations using statistical analyses of the resulting hillslopes. Loritz et al. (2017) conceptualize and parameterize representative 2D hillslope models against two monitored catchments of widely differing sizes (~300 km^2 and ~20 km^2), showing good agreement with discharge, but highlight some variability depending on seasonal variability. Another useful review summarising hillslope catchment research which includes these and many other studies is provided by Paniconi and Putti (2015).

We have clarified the text in the Introduction section (lines 80-89) and Methods section (Section 2.2 lines 113-120, Section 2.3 lines 145-176) and also revised Fig 2 to include topography elevation contours, which now helps clarify the convergent nature of the hillslope, and updated Fig 3 with additional panels to show the domain and mesh discretization more clearly. Also, the more detailed description of the model design in Appendix A (lines 536-555) has been rewritten with further clarification both in text and with an additional figure Fig A1 which shows the convergent nature of the hillslope and model domain representation.

Paniconi, C., Troch, P.A., van Loon, E.E., Hilberts, A.G.J., 2003. Hillslope-storage Boussinesq model for subsurface flow and variable source areas along complex hillslopes: 2. Intercomparison with a three-dimensional Richards equation model. Water Resources Research 39. https://doi.org/10.1029/2002WR001730

Troch, P.A., Van Loon, A.H., Hilberts, A.G.J., 2004. Analytical solution of the linearized hillslope-storage Boussinesq equation for exponential hillslope width functions. Water Resources Research 40, 2003WR002850. https://doi.org/10.1029/2003WR002850

Hilberts, A.G.J., Van Loon, E.E., Troch, P.A., Paniconi, C., 2004. The hillslope-storage Boussinesq model for non-constant bedrock slope. Journal of Hydrology 291, 160–173. https://doi.org/10.1016/j.jhydrol.2003.12.043

Hsieh, P.-C., Huang, T.-T., 2023. Modelling of hillslope storage under temporally varied rainfall recharge. Math. Model. Nat. Phenom. 18, 9. https://doi.org/10.1051/mmnp/2023009

Hazenberg, P., Fang, Y., Broxton, P., Gochis, D., Niu, G.-Y., Pelletier, J.D., Troch, P.A., Zeng, X., 2015. A hybrid-3D hillslope hydrological model for use in Earth system models. Water Resources Research 51, 8218–8239. https://doi.org/10.1002/2014WR016842

Hazenberg, P., Broxton, P., Gochis, D., Niu, G.-Y., Pangle, L.A., Pelletier, J.D., Troch, P.A., Zeng, X., 2016. Testing the hybrid-3-D hillslope hydrological model in a controlled environment. Water Resources Research 52, 1089–1107. https://doi.org/10.1002/2015WR018106

Chaney, N.W., Van Huijgevoort, M.H.J., Shevliakova, E., Malyshev, S., Milly, P.C.D., Gauthier, P.P.G., Sulman, B.N., 2018. Harnessing big data to rethink land heterogeneity in Earth system models. Hydrology and Earth System Sciences 22, 3311–3330. https://doi.org/10.5194/hess-22-3311-2018

Loritz, R., Hassler, S.K., Jackisch, C., Allroggen, N., van Schaik, L., Wienhöfer, J., Zehe, E., 2017. Picturing and modeling catchments by representative hillslopes. Hydrology and Earth System Sciences 21, 1225–1249. https://doi.org/10.5194/hess-21-1225-2017

Paniconi, C., Putti, M., 2015. Physically based modeling in catchment hydrology at 50: Survey and outlook. Water Resources Research 51, 7090–7129. https://doi.org/10.1002/2015WR017780

8. L 630-643: Hard to follow. A schematic figure presenting which slope is dealt with etc would probably be helpful.
   Response: The model design in Appendix A has been rewritten and the part this comment refers to was deemed unnecessary and has been removed. A new figure Fig A1 has been introduced to clarify the site location with a map view and view of the model domain. Fig 2 and A1a show elevation bands, which map directly to grid cells in our semi-generic hillslope model shown in Fig 3 and A1b.

9. L 646-647: "At the top boundary (surface), a surface energy balance, which serves as a source and sink for water and energy in the subsurface, is derived from site-specific weather data." How is this SEB derived? This should be explained.
   Response: The SEB refers to the surface energy balance model used, which has been described in detail in previous publications. This has been clarified in the main text in the Methods section 2.3 (lines 168-173) and in Appendix A (lines 583-592), and relevant references have been added.

10. L 659-661: "By using a frequency-density function, the resulting distribution more closely resembles natural rainfall variability, ensuring a realistic representation of precipitation events. This precipitation model plays a critical role in shaping the hydrological and thermal balance within the soil." Interesting. If I understand correctly the used precipitation distribution is a realization of a random process with prescribed statistical moments? Then I think that it would be interesting to test the variability of the

output results of this study when using different precipitation distributions obtained by different realization of this random process.

Response: While exploring different precipitation patterns would be an interesting avenue for future research, it falls outside the scope of this study and does not align with our current aims. The use of the variable frequency-density function is intended to represent realistic precipitation variability based on the available 10-year hydro-meteorological weather dataset, with an approach based on our previous research.

11. L 662-665: Table 1 should be inserted here, or this paragraph moved in the body of the text as a comment of this table. This should be also more precise : why these peculiar values has been chosen ? Saying 'to resemble highly water-conductive material' is not specific enough. You did not invent these value I guess, the abaqus from where they were extracted or whatever should be mentioned.

Response: These values are consistent with qualitative observations of soil textures on site. The revised version has a reference to Table 1 (page 9), which is updated with relevant sources.

12. L 703-709: Figure D3 should be cited here.

Response: Done.

13. L 716: Figure D4 is not displayed in the Appendix.

Response: Figure D4 is included in Appendix D.

14. L 716-728: The convergence study, showing dependency of the results on mesh refinement, is done only for the upper part of the domain. It should be done for the whole domain, especially in the Main Area of Interest. Another comment related to the answer of the authors regarding my comments 32 : "energy- and mass conservation apply to these cells in the same way that it does in the contemporary active layer cells. The mesh resolution is therefore adequate." Please do not forget that energy and mass conservation is necessary but not sufficient for obtaining converged numerical results ; truncation errors must also be assessed.

Response: The text in the Appendix and corresponding text in the Methods section has been improved to clarify the modeling approach, please see our previous responses. We undertook another mesh convergence study in response to this comment, increasing the mesh resolution in the transport region, i.e., the main area of interest for the analysis. Numerical differences were negligible and further confirms the model robustness. We note the original mesh with spacing of 0.5 m was already overly refined based on our experience with these types of simulations, so it is no surprise that results from a superfine mesh with 0.25 m spacing was virtually indistinguishable. We have added new figures Fig D5 and D6 showing temperature and Darcy velocity for comparing the original

fine mesh of the seasonal simulations and our new superfine mesh. We also added text to in Appendix D describing the result (lines 671-676).

---

## Author Response (AR3)

**Response letter for Model-based analysis of solute transport and potential carbon mineralization in the active layer of a hillslope underlain by permafrost with seasonal variability and climate change, by Hamm, A., Mannerfelt, E.S., Mohammed, A., Painter, S.L., Coon, E., Frampton, A.**

**23 May 2025**

**Message from Editor**

Based on the last reviewer's report, it appears that the main issue, namely the clear definition of the study's focus, has been addressed adequately, leaving only minor concerns regarding the description of the methodology. If these can also be addressed, the manuscript should be in a good shape for publication.

Response: We thank the Editor for their careful assessment and positive feedback and appreciate the recognition that the main concerns have been addressed. We have revised the manuscript to clarify the remaining points raised by the Reviewer, as outlined in our responses below. We are grateful for the Reviewer's recommendation for acceptance with minor revisions and for their support in moving the manuscript toward publication.

**Comments from Reviewer**

The additional work done by the authors enables to better defined the scope and the limits of their study. The authors propose a modelling work somewhere in between the study of an idealized, generic case and the study of a site-specific case, which in my opinion makes difficult to draw the line between the conclusions that are generic and those that are site-specific. Nevertheless that is the explicit aim of the authors, and as such I think that it can be presented to the community. I have still some minor concerns regarding the presentation of the study site and of the simulation methodology that should be dealt with prior to publication, see my specific comments below. But overall I think that now the manuscript is in a sufficiently good shape for publication, so at this stage I recommend acceptance with minor revisions.

Response: We thank the Reviewer for their constructive feedback and positive assessment. We have carefully addressed the remaining minor concerns regarding the presentation of the study site and simulation methodology, as detailed in our responses below. We believe these revisions further clarify the objectives and methods and further strengthen the manuscript presentation.

Specific comments:

l99-111 (line numbering from the final manuscript version): in this paragraph the ecotype of the site (tundra, bare soil, ... ?) must be given.

Response: The site description has been updated with a specification of high-arctic mountain tundra, as follows (L103): "The landscape in the area is characterized by high-arctic mountain tundra with gentle slopes towards the valley bottom …"

l115-116: "a synthetic domain that is broadly representative of hillslopes in continuous permafrost regions."
I am still not convinced that the chosen hillslope maybe considered as 'broadly representative' of continuous permafrost region. This is a N-W slope in (probably) high arctic tundra, most likely it is not representative of S-E slopes, or of boreal forest environments. This sentence could be rephrased as follow : "a synthetic domain that is representative of the site conditions, in a high arctic tundra environment (?) of the continuous permafrost area."

Response: This specification has been included and the text has been rephrased, as follows (L115): "The model design aims to capture key physiographic characteristics of the site and is intended to be a synthetic domain that is broadly representative of hillslopes in a high-arctic mountain tundra environment underlain by continuous permafrost."

l 154-156: "The variable-width mesh approach preserves flow convergence, enables a reasonable representation of the surface-energy balance, and allows hydrological processes to be well represented without the expense of a fully three-dimensional model (e.g., Fan and Bras, 1998; Troch et al., 2003; Hazenberg et al., 2015)."
Concerning the 'reasonable representation of surface-energy balance' and the 'well represented hydrological processes', this sentence presents assumptions as demonstrated facts, at least in permafrost contexts. Moreover, the cited literature does not deal with coupled water and heat transfers with freeze/thaw, nor with solute transport. The sentence could be rephrased in the following way: "The variable-width mesh approach preserves flow convergence, and enables a reasonable representation of the hydrological processes in non-permafrost contexts without the expense of a fully three-dimensional model (e.g., Fan and Bras, 1998; Troch et al., 2003; Hazenberg et al., 2015). In this study we assume that this modelling approach is also suitable for the simulation of coupled water and heat transfers with freeze/thaw, and to waterborne solute transport in such a permafrost context."

Response: This has been included and the text has been rephrased, as follows (L154): "The variable-width mesh approach preserves flow convergence, enables a reasonable representation of the surface-energy balance, and allows hydrological processes to be well represented without the expense of a fully three-dimensional model (e.g., Fan and Bras, 1998; Troch et al., 2003; Hazenberg et al., 2015). In this study we assume that this approach is also

suitable for simulation of coupled water and heat and solute transport with active layer freeze-thaw in a permafrost environment."

l 675-676: "Simulated temperature and x-velocity using the fine and superfine meshes are nearly indistinguishable (Fig. D5 and D6)." Indistinguishable by the human eye on a figure is not a relevant criterium of comparison for a convergence study. Here a quantitative comparison must be done, for instance by giving the L1 and L2 norms of the values of the differences between the temperature fields obtained with the two meshes.

Response: A quantitative comparison between the temperature fields and velocity fields for the mesh cases further demonstrates that the differences between them are negligible. We updated the text in Appendix D showing the result of the differences, as follows (L678): "The mean absolute temperature difference between the fine and superfine meshes in the thawed part of the highly refined transport region is 6.55 $10^{-3}$ C. The mean absolute difference in x-direction Darcy velocity between the fine and superfine meshes in the thawed part of the highly refined transport region is 5.91 $10^{-7}$ m/s."